# The Implicit Bias of Adam and Muon on Smooth Homogeneous Neural Networks

Eitan Gronich [1]   Gal Vardi [1]

## Abstract

We study the implicit bias of momentum-based optimizers on smooth homogeneous models. We show that *momentum steepest descent* algorithms like Muon (spectral norm), MomentumGD ($\ell_2$ norm), and Signum ($\ell_\infty$ norm) are *approximate* steepest descent trajectories under a decaying learning rate schedule, proving that these algorithms have a bias towards KKT points of the corresponding margin maximization problem. We extend the analysis to Adam (without the stability constant), which maximizes the $\ell_\infty$ margin, and to Muon-Signum and Muon-Adam, which maximize a hybrid norm. Our experiments corroborate the theory and show that the identity of the margin maximized depends on the choice of optimizer. Overall, our results extend earlier lines of work on steepest descent in homogeneous models and momentum-based optimizers in linear models.

## 1. Introduction

Deep neural networks show remarkable generalization performance despite often being overparameterized, and even when trained with no explicit regularization. A well-established line of work attempts to explain this phenomenon with the notion of the *implicit* bias (tendency) of gradient-based optimization algorithms to converge to well-generalizing solutions. This bias is often realized in the form of maximizing a certain *margin* for the training points (cf. Vardi (2023)).

While earlier works studied mostly gradient descent and showed its implicit bias towards maximizing the $\ell_2$ margin in increasingly complex models, recent years have witnessed an interest in the study of the implicit bias of other optimizers, such as Adam (Kingma and Ba, 2015), AdamW (Loshchilov and Hutter, 2019), and recently Muon (Jor-

dan et al., 2024), hand-in-hand with their rising popularity. Indeed, as these algorithms are used near-universally for training large language models (LLMs) and vision transformers, there is a growing imperative to understand their inner workings.

In this work, we study smooth homogeneous models and show a margin-maximization bias of Adam and Muon. Previous work analyzed the implicit bias of Adam and Muon on linear predictors (Zhang et al., 2024; Fan et al., 2025), and we extend these results to the substantially broader class of smooth homogeneous models. Moreover, our analysis of Muon is a special case of a more general framework that we develop, which is applicable to all momentum-based optimizers built on top of steepest descent algorithms. All of our results hold for a family of exponentially tailed losses that includes the logistic and exponential losses.

Our main contribution is showing that when the parameter direction $\frac{\boldsymbol{\theta}_t}{\|\boldsymbol{\theta}_t\|}$ converges, it converges to the direction of a KKT point of the $\|\cdot\|$-max-margin problem even for trajectories which *approximate* steepest descent in an appropriately defined way. This allows us to focus on momentum-based optimizers on smooth homogeneous models and show:

1. Muon has an implicit bias towards margin maximization with respect to a norm defined using spectral norms of the weight matrices, under a decaying learning rate regime. In fact, the bias towards margin maximization holds for any normalized Momentum Steepest Descent (MSD) algorithm, for the appropriate norm. We show this includes composite MSD algorithms such as Muon-Signum. In addition, we prove an implicit bias of Muon-Adam.

2. Adam (without the stability constant) has an implicit bias towards $\ell_\infty$ margin maximization under a decaying learning rate regime.

**Related Work**

Soudry et al. (2018) first showed that gradient descent in linear models maximizes the $\ell_2$ margin. This result was followed by several works on margin maximization in linear fully-connected, convolutional and diagonal networks (e.g., Ji and Telgarsky (2018); Gunasekar et al. (2018b); Yun et al. (2020); Moroshko et al. (2020)). Going beyond

---

[1]Department of Computer Science, Weizmann Institute, Rehovot, Israel. Correspondence to: Eitan Gronich <eitan.gronich@weizmann.ac.il>.

*Proceedings of the 43rd International Conference on Machine Learning*, Seoul, South Korea. PMLR 306, 2026.

linear networks, Chizat and Bach (2020) studied the implicit bias in infinitely-wide two-layer smooth homogeneous networks, and proved margin maximization w.r.t. a certain function norm, known as the variation norm. Lyu and Li (2019) studied homogeneous models under gradient descent, demonstrating that any limit point of the direction of the vector of parameters $\frac{\boldsymbol{\theta}_t}{\|\boldsymbol{\theta}_t\|}$ is the direction of a KKT point of the max-margin problem. In a complementary result, Ji and Telgarsky (2020) showed that directional convergence of the parameters is indeed guaranteed when optimizing homogeneous models definable in an o-minimal structure with gradient descent. The implicit bias of gradient descent in certain non-homogeneous neural networks was studied in Nacson et al. (2019a); Kunin et al. (2022); Cai et al. (2025). For a more comprehensive survey on the implicit bias of gradient descent, see Vardi (2023).

A general treatment of the implicit bias of the family of steepest descent algorithms was given for linear models by Gunasekar et al. (2018a), who proved maximization of the appropriate norm-dependent margin. Tsilivis et al. (2025) generalized that result and the result by Lyu and Li (2019) and proved for homogeneous models under steepest descent that any limit point of $\frac{\boldsymbol{\theta}_t}{\|\boldsymbol{\theta}_t\|}$ is a KKT point of the max-margin problem.

The implicit bias of Adam in the context of homogeneous models was studied by Wang et al. (2021), who showed a bias towards $\ell_2$-margin maximization. Notably, this work studied Adam without momentum in the numerator and with a stability constant $\varepsilon$ which asymptotically dominates the denominator, driving behavior to be similar to gradient descent. Follow-up works have argued that the analysis of the implicit bias with the stability constant is less faithful to the characteristics of Adam in practice, as the stability constant is typically negligible throughout the trajectory. Such works on Adam without the stability constant have so far focused on linear models and include Zhang et al. (2024) and Fan et al. (2025), in the binary and multiclass settings respectively, who showed $\ell_\infty$ margin maximization, and Baek et al. (2025) who showed that the implicit bias of Adam under a deterministic batching routine can deviate from the full-batch case. We generalize the result by Zhang et al. (2024) to smooth homogeneous models. AdamW, contrasting with other algorithms by its utilization of explicit weight decay, was studied for smooth models and losses by Xie and Li (2024), who showed that limit points of the trajectory are KKT points of the loss under the constraint that the $\ell_\infty$ norm of the parameters is bounded.

Fan et al. (2025) studied normalized steepest descent and its momentum counterparts on linear models in the multiclass setting, including spectral descent and Muon respectively, and showed maximization of the appropriate margins. We generalize their result, albeit in the binary classification

setting.

Studies of Adam include dynamical analyses and proofs of convergence in some settings (e.g. Bock and Weiß (2019b); Défossez et al. (2020); Zou et al. (2019); Zhang et al. (2022); Barakat and Bianchi (2021)), as well as examples of non-convergence (Reddi et al. (2019); Bock and Weiß (2019a)), and other aspects such as generalization and loss-landscape geometry (Wilson et al., 2017; Zhou et al., 2020). The Muon optimizer was inspired by path-SGD (Neyshabur et al., 2015).

The significance of "tame" geometry, including stratifiability, in non-smooth dynamical systems was studied in foundational works such as Bolte et al. (2007a;b), and built upon by Davis et al. (2020) in the context of optimization methods.

## 2. Preliminaries

### 2.1. Setting and Notations

Throughout this work, we consider a fixed binary classification dataset $\{(\mathbf{x}_i, y_i)\}_{i=1}^m \subseteq \mathbb{R}^d \times \{\pm 1\}$, a parameterized model $f(\mathbf{x}; \boldsymbol{\theta})$ for parameters $\boldsymbol{\theta} \in \mathbb{R}^p$, and a *log-concave, exponentially tailed* loss of the form:

$$\mathcal{L}(\boldsymbol{\theta}) = \sum_{i=1}^m \ell\left(y_i f(\mathbf{x}_i; \boldsymbol{\theta})\right) = \sum_{i=1}^m e^{-\varphi(y_i f(\mathbf{x}_i; \boldsymbol{\theta}))}, \quad (1)$$

where $\varphi$ is twice continuously differentiable, strictly monotone increasing and convex, with bounded first and second derivatives (see Appendix C.1), notably allowing for the exponential ($\ell(u) = e^{-u}$) and logistic ($\ell(u) = \log(1 + e^{-u})$) losses. We denote for brevity $z_i(\boldsymbol{\theta}) = y_i f(\mathbf{x}_i; \boldsymbol{\theta})$ and $q_{\min}(\boldsymbol{\theta}) = \min_{i \in [m]} z_i(\boldsymbol{\theta})$. For a trajectory $\boldsymbol{\theta}_t$ we often write $z_i^t = z_i(\boldsymbol{\theta}_t), q_{\min}^t = q_{\min}(\boldsymbol{\theta}_t)$ or $z_i, q_{\min}$ when $t$ is clear from context.

For a vector $\mathbf{v} \in \mathbb{R}^n$ we denote by $\mathbf{v}[j]$ the $j$'th coordinate of $\mathbf{v}$. For $n \in \mathbb{N}$, we denote $[n] = \{1, \ldots, n\}$. We denote by $\|\cdot\|_p$ the $\ell_p$ norm for $p \in [1, \infty]$. For an arbitrary norm $\|\cdot\|$ we denote by $\|\cdot\|_\star$ the dual norm, defined by $\|\mathbf{x}\|_\star = \max_{\|\mathbf{u}\|=1} \langle \mathbf{u}, \mathbf{x} \rangle$. We denote by $\|W\|_{\mathrm{sp}}$ the standard spectral norm of a matrix $W$, and $\|(W_1, ..., W_K)\|_{\mathrm{msp}} := \max_{k \in [K]} \|W_k\|_{\mathrm{sp}}$ (short for max-spectral). We use the standard asymptotic notations $\mathcal{O}, \Omega, \Theta, \mathrm{o}, \omega$. We denote by $C^k(X)$ for $X \subseteq \mathbb{R}^n, n \in \mathbb{N}$ the class of $k$-times continuously differentiable functions from $X$ to $\mathbb{R}$. By $\log u$ we refer to the natural logarithm. By $\mathrm{ess\,lim}, \mathrm{ess\,liminf}, \mathrm{ess\,limsup}$ we refer to *essential* limits holding up to sets of measure 0.

### 2.2. Optimizers

The optimization algorithms we study are derivatives of the *steepest descent* family, a generalization of gradient

descent defined with respect to a norm $\|\cdot\|$ (and its dual norm $\|\cdot\|_\star$). We study the infinitesimal step size (flow) limit of the optimization trajectories. We define steepest descent and its normalized variant in the general case of a subdifferentiable model $f$, allowing for a learning rate schedule $\eta(t) > 0$, as follows:

**Steepest Descent:**

$$\frac{d\boldsymbol{\theta}_t}{dt} \in \left\{ \eta(t) \cdot \arg\min_{\|\mathbf{u}\|=\|\mathbf{g}_t\|_\star} \langle \mathbf{u}, \mathbf{g}_t \rangle \mid \mathbf{g}_t \in \partial\mathcal{L}(\boldsymbol{\theta}_t) \right\}, \tag{2}$$

**Normalized Steepest Descent:**

$$\frac{d\boldsymbol{\theta}_t}{dt} \in \left\{ \eta(t) \cdot \arg\min_{\|\mathbf{u}\|=1} \langle \mathbf{u}, \mathbf{g}_t \rangle \mid \mathbf{g}_t \in \partial\mathcal{L}(\boldsymbol{\theta}_t) \right\}, \tag{3}$$

for almost every $t \geq 0$, where $\partial\mathcal{L}$ is the Clarke subdifferential of $\mathcal{L}$ (see Appendix A), which reduces to $\nabla\mathcal{L}$ wherever $f$ is differentiable. Notably, *gradient descent* and *coordinate descent* are recovered with $\|\cdot\| = \|\cdot\|_2, \|\cdot\|_1$ respectively from Equation (2), and *sign gradient descent* is recovered with $\|\cdot\| = \|\cdot\|_\infty$ from Equation (3). We note that in the flow regime, normalization of the update and the addition of the LR schedule $\eta(t)$ do not affect the trajectories in parameter space (but rather the traversal speed only); the above presentation of Equations (2), (3) is brought for completeness and comparison to Equations (5), (6) below.

Introducing momentum-based optimizers, we consider a choice of subgradients $\mathbf{g}_t \in \partial\mathcal{L}(\boldsymbol{\theta}_t)$ along the trajectory, and denote the momentum estimate by the following ODE with the given explicit solution:

$$\frac{d\mathbf{m}_t}{dt} = c_1(\mathbf{g} - \mathbf{m}_t), \quad \mathbf{m}_0 = \mathbf{0}$$
$$\left( \text{Explicitly:} \quad \mathbf{m}_t = \int_0^t c_1 e^{-c_1(t-s)} \mathbf{g}_s ds \right). \tag{4}$$

For the full derivation of the above continuous analogue of momentum, see Appendix B. Here, $c_1 > 0$ is the *momentum smoothing parameter*, and $\frac{1}{c_1}$ is the characteristic time frame in which past gradients are accumulated ($c_1$ is analogous to $-\log(\beta_1)$ for a discrete momentum parameter $\beta_1 \in (0, 1)$, and is roughly $1 - \beta_1$ when $\beta_1$ is close to 1). We now define *momentum steepest descent* and its normalized counterpart:

**Momentum Steepest Descent:**

$$\frac{d\boldsymbol{\theta}_t}{dt} \in \left\{ \eta(t) \cdot \arg\min_{\|\mathbf{u}\|=\|\mathbf{m}_t\|_\star} \langle \mathbf{u}, \mathbf{m}_t \rangle \right\}, \tag{5}$$

**Normalized Momentum Steepest Descent:**

$$\frac{d\boldsymbol{\theta}_t}{dt} \in \left\{ \eta(t) \cdot \arg\min_{\|\mathbf{u}\|=1} \langle \mathbf{u}, \mathbf{m}_t \rangle \right\}. \tag{6}$$

Classical gradient descent with momentum, for example, is obtained from Equation (5) with $\|\cdot\| = \|\cdot\|_2$. *Muon*, the recently proposed weight-matrix optimizer (Jordan et al., 2024), applies Newton-Schulz orthogonalization iterations on a momentum estimate of the matrix. In our work (as in Fan et al. (2025)), Muon refers to the exact orthogonalization setting rather than the Netwon-Schulz approximation (i.e., $U\Sigma V^T \mapsto UV^T$ where $U\Sigma V^T$ is the SVD of the weight matrix $W$ – defined precisely in C.6). Muon with exact orthogonalization is recovered from Equation (6) with $\|\cdot\| = \|\cdot\|_{\text{sp}}$ for a single-layer network. When running Muon simultaneously on each weight matrix of a multi-layer network, the resulting trajectory follows Equation (6) with $\|\cdot\| = \|\cdot\|_{\text{msp}}$ (see notations in Subsection 2.1). Bernstein and Newhouse (2024) noted that Shampoo with accumulation disabled is, too, spectral descent, although accumulation in Shampoo is not identical to momentum. Another algorithm adhering exactly to Equation (6) is *Signum* (Bernstein et al., 2018), i.e. momentum sign gradient descent ($\|\cdot\| = \|\cdot\|_\infty$).

The final optimizer we discuss is *Adam*, for which we define similarly to the above

$$\frac{d\mathbf{v}_t}{dt} = c_2(\mathbf{g}_t^2 - \mathbf{v}_t), \quad \mathbf{v}_0 = \mathbf{0}$$
$$\left( \text{Explicitly:} \quad \mathbf{v}_t = \int_0^t c_2 e^{-c_2(t-s)} \mathbf{g}_s^2 ds \right), \tag{7}$$

where the square is taken element-wise. Following Zhang et al. (2024); Fan et al. (2025); Baek et al. (2025); Xie and Li (2024), we consider Adam without the stability constant, since such a constant dominates $\mathbf{v}_t$ in asymptotic analysis, contrary to the typical situation in practice where the stability constant is negligible throughout the trajectory. Therefore we define Adam as the following:

$$\textbf{Adam:} \quad \frac{d\boldsymbol{\theta}_t}{dt} = -\eta(t) \cdot \frac{\hat{\mathbf{m}}_t}{\sqrt{\hat{\mathbf{v}}_t}}, \tag{8}$$

where $\hat{\mathbf{m}}_t = (1 - e^{-c_1 t})^{-1}\mathbf{m}_t$ and $\hat{\mathbf{v}}_t = (1 - e^{-c_2 t})^{-1}\mathbf{v}_t$ are bias-corrected terms, and division and square root are taken element-wise. Adam in the discrete case (Kingma and Ba (2015)) is defined using parameters $\beta_1, \beta_2 \in [0, 1)$, where $\beta_1 \leq \beta_2$ (see definition for the discrete case in Appendix B).[1] Due to the inverse relation between $\beta_i$ and $c_i$, this is analogous to $c_1 \geq c_2$, which is the setting we focus on.

## 2.3. Assumptions

We now introduce the assumptions made in this work. Note that some assumptions overlap, and not all assumptions are used in all sections of the work.

---

[1] Pytorch (Paszke et al., 2019) defaults are $\beta_1 = 0.9, \beta_2 = 0.999$.

**Model Assumptions.** Our main contributions include the following assumptions on $f$:

(M1) $f$ is smooth in $\boldsymbol{\theta}$, i.e. $\forall \mathbf{x} \in \mathbb{R}^d : f(\mathbf{x}; \cdot) \in C^1(\mathbb{R}^p)$.

(M2) $f$ is $L$-homogeneous for some $L \geq 1$, i.e. $\forall \mathbf{x} \in \mathbb{R}^d, \boldsymbol{\theta} \in \mathbb{R}^p, \alpha > 0 : f(\mathbf{x}; \alpha\boldsymbol{\theta}) = \alpha^L f(\mathbf{x}; \boldsymbol{\theta})$.

This includes (deep) linear networks, for which implicit bias has been extensively studied (e.g., Ji and Telgarsky (2018); Gunasekar et al. (2018b); Yun et al. (2020); Moroshko et al. (2020)), but notably also models with smooth non-linear activations such as assumed in Chizat and Bach (2020). One example for an activation function that induces non-linear smooth homogeneous networks is $\mathrm{ReLU}^q(z) := \max\{0, z\}^q$, for any constant $q > 1$ (networks with this activation function have been studied in, e.g., Cao et al. (2022); Min and Vidal (2024; 2025); Chizat and Bach (2020)). Another example for a smooth homogeneous activation is the quadratic activation $z \mapsto z^2$, which has been studied in many prior works (e.g., Soltanolkotabi et al. (2018); Du and Lee (2018); Gamarnik et al. (2019); Sarao Mannelli et al. (2020); Mohamadi et al. (2024); Martin et al. (2024); Arous et al. (2025); Martin et al. (2026)).

We also introduce a weakened version of (M1):

(M1-Weak) $f$ is locally Lipschitz and Whitney $C^1$-stratifiable (thereby admitting a chain rule), see Appendix A.

**Learning Rate Assumptions.** We detail the different sets of assumptions on the learning rate schedule $\eta(t)$, according to the setting (momentum steepest descent and Adam/Muon-Adam).

(LR-MSD) $\eta(t)$ satisfies $\int_0^\infty \eta(t)dt = \infty$ and $\eta(t) \leq o\left(t^{\frac{1}{L}-1}\right)$, where $L \geq 1$ is from (M2).

(LR-Adam) $\eta(t)$ satisfies $\int_0^\infty \eta(t)dt = \infty$ and $\eta(t) \leq o\left(t^{\frac{1}{L}-1}\right)$, where $L \geq 1$ is from (M2), and is non-increasing.

We note that existing works on Adam and momentum steepest descent in linear models (Zhang et al., 2024; Fan et al., 2025; Baek et al., 2025) assumed a non-increasing learning rate $\eta_t$ with $\sum_{t=1}^\infty \eta_t = \infty$ and $\eta_t = o(1)$, as well as additional technical assumptions; our Assumption (LR-Adam) in the linear predictor case ($L = 1$) is somewhat weaker than theirs. We also point out that the extensively studied harmonic learning rate schedule $\eta(t) = \frac{1}{t}$ satisfies the assumptions for any $L > 1$.

**Realizability and Trajectory Assumptions.** Our results hold given the following assumptions on the trajectory:

(T1) Nontrivial trajectory: $\exists N_{\min} > 0, t_0 \geq 0 : \forall t \geq t_0 : \|\boldsymbol{\theta}_t\| \geq N_{\min}$.

(T2) Directional Convergence: $\frac{\boldsymbol{\theta}_t}{\|\boldsymbol{\theta}_t\|}$ converges to some $\bar{\boldsymbol{\theta}}$ with a positive margin $\min_{i \in [m]} y_i f(\mathbf{x}_i; \bar{\boldsymbol{\theta}}) > 0$.

Assumption (T1) guarantees only that $\boldsymbol{\theta}_t$ is eventually bounded away from the origin. In particular this assumption holds if eventually $\mathcal{L}(\boldsymbol{\theta}_t) < m \cdot \ell(0) - \delta$ for some $\delta > 0$. This is therefore a very mild assumption. (T1) follows from realizability when analyzing (normalized) steepest descent, as in (Tsilivis et al., 2025), since the loss is proved to decay once $\mathcal{L}(\boldsymbol{\theta}_t) < \ell(0)$; momentum steepest descent and Adam, however, only asymptotically approximate steepest descent, and such a decay is only proved for them in this work under Assumptions (T1) and (T2).

Assumption (T2) is common in the implicit bias literature for linear and homogeneous networks (see, for instance, Gunasekar et al. (2018a;b); Chizat and Bach (2020); Nacson et al. (2019b)).[2] We note the question of convergence is typically decoupled from that of implicit bias, and results on implicit bias often assume convergence. As the direction of parameters in homogeneous models captures all model behavior up to scale, assuming directional convergence is akin to assuming convergence of the model. Regarding the positive margin, we note that given that the model interpolates the training data (i.e. realizability), the margin is positive throughout the whole late phase of the trajectory, and the assumption only rules out the possibility of the margin decaying to 0 asymptotically. This assumption can also be found in Gunasekar et al. (2018b); Nacson et al. (2019b). Finally, in Figure 1 we show empirical evidence of directional convergence and strictly positive margins in our experiments.

**Adam Well-Definability Assumption.** To discuss Adam without the stability constant and guarantee that $\mathbf{v}_t[j] > 0$ for all $j \in [p]$, which is required to prevent division by zero, we introduce the following technical assumption regarding the initialization. This assumption appears in similar form also in Zhang et al. (2024) and Fan et al. (2025), while Baek et al. (2025) assume nonzero coordinates of the input in the iterative batching regime, leading to a similar conclusion.

(A1) There exist $\tau > 0, \rho > 0$ such that for all $j \in [p]$ and for almost any $t \in [0, \tau]$, we have $\mathbf{g}_t[j]^2 > \rho$. These $\tau$ and $\rho$ may be arbitrarily small.

---

[2]These works were published with no result known about directional convergence at the time (directional convergence of gradient descent was proved by (Ji and Telgarsky, 2020)).

## 2.4. Margin Maximization and KKT Conditions

An important notion when discussing implicit bias is that of the *(hard) margin*, defined for homogeneous models with respect to a norm $\|\cdot\|$ by

$$\gamma(\boldsymbol{\theta}) = \min_{i \in [m]} y_i f\left(\mathbf{x}_i; \frac{\boldsymbol{\theta}}{\|\boldsymbol{\theta}\|}\right) . \quad (9)$$

Under Assumption (T2), $\gamma(\boldsymbol{\theta}_t)$ converges to $\gamma(\bar{\boldsymbol{\theta}}) > 0$. We also consider the following *soft margin*

$$\widetilde{\gamma}(\boldsymbol{\theta}) = \frac{\varphi^{-1}\left(\log \frac{1}{\mathcal{L}(\boldsymbol{\theta})}\right)}{\|\boldsymbol{\theta}\|^L} , \quad (10)$$

as a convenient substitute for $\gamma(\boldsymbol{\theta})$. The soft margin approximates the hard margin with a $\mathcal{O}\left(\frac{\log m}{\|\boldsymbol{\theta}\|^L}\right)$ error that vanishes whenever $\|\boldsymbol{\theta}\| \to \infty$ (as is indeed proved in all of our results). The examination of quantities relating to $\widetilde{\gamma}$ prove important in the analysis.

Our results pertain to the following objective, known as margin maximization, which is not directly optimized by any of the aforementioned algorithms:

$$\min_{\boldsymbol{\theta} \in \mathbb{R}^p} \frac{1}{2}\|\boldsymbol{\theta}\|^2 \quad \text{s.t.} \quad \forall i \in [m] : y_i f(\mathbf{x}_i; \boldsymbol{\theta}) \geq 1 . \quad (11)$$

Minimizing the norm $\|\boldsymbol{\theta}\|$ while preserving feasibility ($\forall i \in [m] : y_i f(\mathbf{x}_i; \boldsymbol{\theta}) \geq 1$) is known to be equivalent to maximizing the margin $\gamma(\boldsymbol{\theta})$. For general homogeneous models, Problem (11) is non-convex, and the implicit bias of algorithms towards minimizing it is shown in light of the *KKT (Karush-Kuhn-Tucker) conditions*, which are local stationarity conditions:

**Definition 2.1.** A point $\boldsymbol{\theta} \in \mathbb{R}^p$ with $y_i f(\mathbf{x}_i; \boldsymbol{\theta}) \geq 1$ for all $i \in [m]$ is said to satisfy the KKT conditions of Problem (11) if there exist $\mathbf{k} \in \partial \frac{1}{2}\|\boldsymbol{\theta}\|^2$, coefficients $\lambda_1, ..., \lambda_m \geq 0$ and subgradients $\mathbf{h}_i \in \partial f(\mathbf{x}_i; \boldsymbol{\theta})$ with:

1. $\sum_{i=1}^m \lambda_i y_i \mathbf{h}_i - \mathbf{k} = \mathbf{0}$;

2. $\sum_{i=1}^m \lambda_i (y_i f(\mathbf{x}_i; \boldsymbol{\theta}) - 1) = 0$.

## 3. Results

In this section, we present our main results. We will discuss the proof ideas in Section 5, with all formal proofs deferred to the appendix.

### 3.1. Normalized Steepest Descent

Our first theorem extends the analysis of Tsilivis et al. (2025) to the setting of normalized steepest descent. We note that Theorem 3.1 can be derived from Tsilivis et al. (2025) with a time reparameterization argument, due to the equivalence of trajectories in the flow regime. However, we present the result with a detailed proof for completeness, and since the proof also reveals previously unknown convergence rates (see Lemma C.10).

The assumptions of the theorem follow Tsilivis et al. (2025), requiring (M1-Weak) and only momentary realizability.

**Theorem 3.1.** *Let $\boldsymbol{\theta}_t$ be a trajectory of normalized steepest descent with respect to a norm $\|\cdot\|$ (Equation (3)). Assume (M1-Weak), (M2). Additionally assume that $\int_0^\infty \eta(t)dt = \infty$ and that there exists $t_0 \geq 0$ such that $\mathcal{L}(\boldsymbol{\theta}_{t_0}) < 1$. Then, any limit point $\bar{\boldsymbol{\theta}}$ of $\frac{\boldsymbol{\theta}_t}{\|\boldsymbol{\theta}_t\|}$ is the direction of a KKT point of Problem (11) with the same norm $\|\cdot\|$.*

### 3.2. Momentum Steepest Descent, Muon and Muon-Signum

We consider margin maximization in momentum steepest descent. The following result is based on the general insight that convergence of $\frac{\boldsymbol{\theta}_t}{\|\boldsymbol{\theta}_t\|}$ to a KKT point of Problem (11) holds even when the trajectory is only an *approximation* of steepest descent. We elaborate on the appropriate definition and characteristics of approximate steepest descent in Section 5 and Appendix C.4.

**Theorem 3.2.** *Let $\boldsymbol{\theta}_t$ be a trajectory of normalized or unnormalized momentum steepest descent with respect to a norm $\|\cdot\|$ (Equation (6) or (5)). Under Assumptions (M1), (M2), (LR-MSD), (T1), (T2), the limit point $\bar{\boldsymbol{\theta}}$ of $\frac{\boldsymbol{\theta}_t}{\|\boldsymbol{\theta}_t\|}$ is the direction of a KKT point of Problem (11) with the norm $\|\cdot\|$.*

Moreover, we show that running multiple normalized (momentum) steepest descent algorithms in parallel on different parts of the parameter vector with respect to different norms is equivalent to a single run of normalized (momentum) steepest descent algorithm relative to the maximal norm among them (see Appendix C.6). As a result we obtain the following corollary on the implicit bias of Muon:

**Corollary 3.3.** *If $\boldsymbol{\theta} = (W_1, ..., W_K)$ is a collection of matrices and Muon is run on each matrix simultaneously with the same schedule $\eta(t)$, then Muon is a case of normalized momentum steepest descent with $\|\cdot\| = \|\cdot\|_{\mathrm{msp}}$, and the statement of Theorem 3.2 holds.*

When running Muon in practice, often the non-matrix parameters are optimized using Adam (Jordan et al. (2024), Liu et al. (2025)). Adam has been compared to sign gradient descent and Signum (see Orvieto and Gower (2025)) as possible simplifications. Recently, Scion (Pethick et al., 2025) has been proposed, which uses Muon side-by-side with sign gradient descent. This motivates understanding the implicit bias of these "composite" algorithms, to which we contribute the following corollary regarding Muon-Signum, and a theorem for Muon-Adam in Subsection 3.3.

**Corollary 3.4.** *If $\boldsymbol{\theta} = (W_1, ..., W_K, \mathbf{u})$ is a collection*

*of matrices and additional parameters* $\mathbf{u}$, *Muon is run on each matrix independently and Signum is run on* $\mathbf{u}$ *with the same schedule* $\eta(t)$, *then Muon-Signum is a case of normalized momentum steepest descent with* $\|\boldsymbol{\theta}\| = \max\{\|(W_1, ..., W_K)\|_{\mathrm{msp}}, \|\mathbf{u}\|_\infty\}$, *and the statement of Theorem 3.2 holds.*

### 3.3. Adam and Muon-Adam

Notably, Adam is *not* a normalized momentum steepest descent algorithm, as its updates are ratio terms of two momentum estimates of different rates. This makes the case of Adam (and hence also Muon-Adam) especially challenging. Yet, we show that results of the same flavor hold for Adam in the decaying learning rate regime. As discussed after Equation (8), we focus on the parameter regime $c_1 \geq c_2 > 0$, which is the one more common in practice, as this is equivalent to $\beta_1 \leq \beta_2$. Here, we will require that $\eta(t)$ is non-increasing; we stress that $\eta(t)$ is chosen externally to the algorithm, and in all practical cases of a decaying learning rate, $\eta(t)$ is chosen to be eventually monotonically decreasing.

**Theorem 3.5.** *Let* $\boldsymbol{\theta}_t$ *be a trajectory of Adam with* $c_1 \geq c_2$ *(Equation* (8)*). Under Assumptions (M1), (M2), (LR-Adam), (T1), (T2), (A1), the limit point* $\bar{\boldsymbol{\theta}}$ *of* $\frac{\boldsymbol{\theta}_t}{\|\boldsymbol{\theta}_t\|}$ *is the direction of a KKT point of Problem* (11) *with* $\|\cdot\| = \|\cdot\|_\infty$.

Next, we consider Muon-Adam. Here, we allow for different momentum parameters and different *base* learning rates for the Muon and Adam algorithms,[3] and show the following:

**Theorem 3.6.** *Assume* $\boldsymbol{\theta} = (W_1, ..., W_K, \mathbf{u}) \in \mathbb{R}^p$ *is a parameter vector representing a collection of matrices and additional parameters* $\mathbf{u}$. *Assume* $W_1, ..., W_K$ *follow a trajectory of Muon and* $\mathbf{u}$ *follows a trajectory of Adam, with respective learning rates of the form* $\eta_0^M \eta(t), \eta_0^A \eta(t)$ *for* $\eta_0^M, \eta_0^A > 0$ *and momentum parameters* $c_M$ *for Muon and* $c_1 \geq c_2$ *for Adam. Assume (M1), (M2), (LR-Adam), (T1), (T2), (A1). Then, the limit point* $\bar{\boldsymbol{\theta}}$ *of* $\frac{\boldsymbol{\theta}_t}{\|\boldsymbol{\theta}_t\|}$ *is the direction of a KKT point of Problem* (11) *with respect to*

$$\|\boldsymbol{\theta}\| = \max\left\{\frac{\eta_0^A}{\eta_0^M}\|(W_1, ..., W_K)\|_{\mathrm{msp}}, \|\mathbf{u}\|_\infty\right\}.$$

## 4. Non-Smooth Models

Our results for momentum steepest descent and Adam are stated under the assumption of smooth models (M1). However, as our proofs (Appendix C.5, C.7) show, this may be weakened to (M1-Weak), if the normalized model subgradients converge. More specifically, denote for all $t$ subgradients $\mathbf{h}(\mathbf{x}_i; \boldsymbol{\theta}_t) \in \partial f(\mathbf{x}_i; \boldsymbol{\theta}_t)$ for which $\mathbf{g}_t = -\sum_{i=1}^m \ell'(z_i^t)\varphi'(z_i^t)y_i\mathbf{h}(\mathbf{x}_i; \boldsymbol{\theta}_t)$. The condition is:

---
[3]The same generalization can be applied to Muon-Signum, and indeed for the distinct matrices in Muon, in Corollaries 3.4, 3.3.

(T3) $\forall i \in [m] : \frac{\mathbf{h}(\mathbf{x}_i; \boldsymbol{\theta}_t)}{\|\boldsymbol{\theta}_t\|^{L-1}}$ converges.

Note first that this condition is trivially satisfied for smooth models under (T2): by Theorem B.2(a) in Lyu and Li (2019), it holds that $\frac{\mathbf{h}(\mathbf{x}_i; \boldsymbol{\theta}_t)}{\|\boldsymbol{\theta}_t\|^{L-1}} \in \partial f(\mathbf{x}_i; \frac{\boldsymbol{\theta}_t}{\|\boldsymbol{\theta}_t\|})$. Therefore, for smooth models, i.e., $f \in C^1$, convergence of $\frac{\mathbf{h}(\mathbf{x}_i; \boldsymbol{\theta}_t)}{\|\boldsymbol{\theta}_t\|^{L-1}}$ is guaranteed from convergence of $\frac{\boldsymbol{\theta}_t}{\|\boldsymbol{\theta}_t\|}$ by continuity of $\nabla f(\mathbf{x}_i; \boldsymbol{\theta}_t)$.

For non-smooth models under (M1-Weak), $\frac{\mathbf{h}(\mathbf{x}_i; \boldsymbol{\theta}_t)}{\|\boldsymbol{\theta}_t\|^{L-1}}$ converges whenever $\frac{\boldsymbol{\theta}_t}{\|\boldsymbol{\theta}_t\|}$ eventually stays in the same $C^1$ stratum of $f(\mathbf{x}_i; \cdot)$ (if $\frac{\boldsymbol{\theta}_t}{\|\boldsymbol{\theta}_t\|}$ is exactly on a stratum boundary, $\mathbf{h}$ can be chosen to conform to any of the bordering strata; the choice of $\mathbf{h}$ should be continuous to ensure convergence). In particular, under (T2), this holds whenever the limiting direction $\bar{\boldsymbol{\theta}}$ is an inner point of a stratum. In homogeneous ReLU networks, stratum boundaries are the parameters $\boldsymbol{\theta}_t$ for which a neuron preactivation is exactly 0. Therefore, when using a consistent choice of ReLU subgradient at 0 (which is always the case in practice), (T3) follows from (T2) for every trajectory in which signs of neuron preactivations eventually stabilize. It is unclear whether this is satisfied in practice or under what conditions; it appears to be violated in our experiments on two-layer ReLU networks with the MNIST dataset, but we leave open the possibility that some settings comply with this condition.

## 5. Main Proof Ideas

First, as noted by Tsilivis et al. (2025) and Ji and Telgarsky (2020), KKT stationarity of limit points of $\frac{\boldsymbol{\theta}_t}{\|\boldsymbol{\theta}_t\|}$ is closely related to alignment of parameters and gradients $\left\langle \frac{\boldsymbol{\theta}_t}{\|\boldsymbol{\theta}_t\|}, -\frac{\mathbf{g}_t}{\|\mathbf{g}_t\|_\star} \right\rangle$. We extract this insight into a general blueprint that serves us to prove implicit bias results on homogeneous models satisfying (M1-Weak); namely, Theorem C.8 states that regardless of the optimization algorithm, any limit point $\bar{\boldsymbol{\theta}}$ of $\frac{\boldsymbol{\theta}_t}{\|\boldsymbol{\theta}_t\|}$ with $\gamma(\bar{\boldsymbol{\theta}}) > 0$ is guaranteed to be a KKT point of Problem (11), if $\mathcal{L}(\boldsymbol{\theta}_{t_n}) \overset{n\to\infty}{\longrightarrow} 0$ and $\left\langle \frac{\boldsymbol{\theta}_{t_n}}{\|\boldsymbol{\theta}_{t_n}\|}, -\frac{\mathbf{g}_{t_n}}{\|\mathbf{g}_{t_n}\|_\star} \right\rangle \overset{n\to\infty}{\longrightarrow} 1$ on a subsequence $t_n$ for which $\frac{\boldsymbol{\theta}_{t_n}}{\|\boldsymbol{\theta}_{t_n}\|} \overset{n\to\infty}{\longrightarrow} \bar{\boldsymbol{\theta}}$.

### 5.1. Approximate Steepest Descent

Our main technical contribution is the extension of the KKT stationarity results to *approximate steepest descent algorithms* and specifically momentum-based algorithms, which we describe here. Steepest descent (normalized or unnormalized) with respect to a norm $\|\cdot\|$ may be described succinctly with the following equation for almost any $t \geq 0$:

$$\left\langle \frac{\frac{d\boldsymbol{\theta}_t}{dt}}{\left\|\frac{d\boldsymbol{\theta}_t}{dt}\right\|}, -\frac{\mathbf{g}_t}{\|\mathbf{g}_t\|_\star} \right\rangle = 1. \tag{12}$$

Equation (12) is the linchpin of analyses of steepest descent, as it allows to prove eventual alignment of the (negative) gradients with the parameters themselves. When analyzing momentum-based algorithms, Equation (12) will not be exactly satisfied, but the hope is that a similar relation will hold asymptotically. Hence we define:

**Definition 5.1** (Approximate Steepest Descent). We say that an arc $\boldsymbol{\theta}_t$ is a trajectory of Approximate Steepest Descent with respect to $\|\cdot\|$ if there exist $\nu(t) > 0, R_{\max} > 0$ with:

1. $\lim_{t\to\infty} N(t) := \lim_{t\to\infty} \int_0^t \nu = \infty$;

2. $\limsup_{t\to\infty} \frac{\|\boldsymbol{\theta}_t\|}{N(t)} \leq R_{\max}$;

3. $\operatorname{ess\,liminf}_{t\to\infty} r(t) \geq 1$, where

$$r(t) \overset{a.e.}{=} \sup_{\mathbf{g}_t \in \partial \mathcal{L}(\boldsymbol{\theta}_t)} \left\langle \frac{1}{\nu(t)} \frac{d\boldsymbol{\theta}_t}{dt}, -\frac{\mathbf{g}_t}{\|\mathbf{g}_t\|_\star} \right\rangle .$$

The quantity $\nu(t)$ can be chosen in a flexible manner; for momentum steepest descent (and indeed exact steepest descent) $\nu(t) = \left\| \frac{d\boldsymbol{\theta}_t}{dt} \right\|$ is chosen, but for Adam we choose $\nu(t) = \eta(t)$ (the learning rate). Lemma C.15 shows that the properties in Definition 5.1, taken together with a positive lower bound on the margin, suffice to prove that $\mathcal{L}(\boldsymbol{\theta}_t) \overset{t\to\infty}{\longrightarrow} 0, \|\boldsymbol{\theta}_t\| \overset{t\to\infty}{\longrightarrow} \infty$. Building on this result and on Theorem C.8, in Theorem C.17 we prove that under (T2) and provided that $R_{\max} \leq 1$, the limiting direction $\bar{\boldsymbol{\theta}}$ is a KKT point of Problem (11).

### 5.2. Asymptotic Momentum-Gradient Relations

Our results for momentum steepest descent (MSD) and Adam rely on the analysis of Approximate Steepest Descent. To show that MSD and Adam indeed satisfy Definition 5.1, we analyze the properties of the momentum operator in Appendix B.

In particular, Corollary B.8 offers a key insight, namely that the ratio $\frac{m(t)}{g(t)}$ for a real-valued function $g(t)$ and its momentum estimator $m(t)$ tends to a well-defined limit whenever $\frac{d\log g}{dt}$ converges. This is applied in Lemma C.19, which demonstrates, that in our setting of optimization trajectories, when $\left\| \frac{d\boldsymbol{\theta}_t}{dt} \right\| \leq o\left(t^{\frac{1}{L}-1}\right)$, it holds that $\mathbf{m}_t[j] = \mathbf{g}_t[j](1 \pm o(1))$ for any coordinate $j \in [p]$ which is momentarily of "significant" magnitude at time $t$. This is formalized by observing the set $J_\varepsilon(t) = \left\{ j \in [p] \mid \frac{|\mathbf{g}_t[j]|}{\|\mathbf{g}_t\|_\star} > \varepsilon \right\}$ for an arbitrary $\varepsilon > 0$. Lemma C.19 also shows that $\frac{\mathbf{m}_t}{\|\mathbf{m}_t\|_\star} - \frac{\mathbf{g}_t}{\|\mathbf{g}_t\|_\star} \overset{t\to\infty}{\longrightarrow} 0$, which allows proving that MSD is indeed an Approximate Steepest Descent algorithm.

In the analysis of Adam, Lemma C.19 is again vital, as it implies that $\frac{\hat{\mathbf{m}}_t[j]}{\sqrt{\hat{\mathbf{v}}_t[j]}} = \operatorname{sign}(\mathbf{g}_t[j])(1 \pm o(1))$ whenever

$\mathbf{g}_t[j]$ is momentarily significant (as above). The approximation of Adam to sign gradient descent is in fact the essence of showing $\ell_\infty$ margin maximization, as sign gradient descent is normalized steepest descent with $\|\cdot\| = \|\cdot\|_\infty$. However, it is not necessarily true for Adam that $\left\langle \frac{\frac{d\boldsymbol{\theta}_t}{dt}}{\left\|\frac{d\boldsymbol{\theta}_t}{dt}\right\|_\infty}, -\frac{\mathbf{g}_t}{\|\mathbf{g}_t\|_1} \right\rangle \overset{t\to\infty}{\longrightarrow} 1$. Instead, we rely on the flexibility of Definition 5.1, choosing $\nu(t) = \eta(t)$. We adapt to our setting an important result proved by Zhang et al. (2024) in the discrete case for linear models, which shows that even if momentarily $\frac{\hat{\mathbf{m}}_t[j]}{\sqrt{\hat{\mathbf{v}}_t[j]}} > 1$, the opposite holds on average. Namely, we prove in Lemma B.10 that for any $j \in [p]$

$$\limsup_{t\to\infty} \frac{|\boldsymbol{\theta}_t[j]|}{\int_0^t \eta} = \limsup_{t\to\infty} \frac{\left| \int_0^t \eta(s) \frac{\hat{\mathbf{m}}_s[j]}{\sqrt{\hat{\mathbf{v}}_s[j]}} ds \right|}{\int_0^t \eta} \leq 1 ,$$

allowing us to choose $R_{\max} \leq 1$ for Definition 5.1 and finish the proof.

## 6. Experiments

To validate our findings we train two-layer (one hidden layer) homogeneous networks to classify $m = 2048$ MNIST digits (LeCun et al., 2002) as even or odd, using the logistic loss. Since our results hold for smooth activations, we use squared ReLU (i.e., $z \mapsto \max\{0, z\}^2$), and also run ReLU for empirical comparison. We compare the following optimizers: Normalized Gradient Descent (NGD) with and without momentum, Signum, Adam, Muon (treating the output layer as a matrix with a single row) and Muon-Adam. Training proceeds until the loss reaches a small target value ($10^{-8}$). The stability constant for Adam is chosen to be negligible with respect to gradient norm values ($\varepsilon = 10^{-20}$). A decaying learning rate $\eta(t) = \eta_0 t^{-0.8}$ is chosen to comply with Assumptions (LR-MSD) and (LR-Adam) (as $t^{-0.8} = o\left(t^{-1/2}\right)$). See Appendix D for additional details and for a similar experiment on the CIFAR10 dataset (Krizhevsky et al., 2009) with a 4-layer network.

Results are shown in Figure 1. As expected, NGD (with and without momentum) achieves the largest $\ell_2$ margin, while Signum and Adam do so for the $\ell_\infty$ margin and Muon for $\|\cdot\|_{\mathrm{msp}}$. These findings seem to hold empirically for ReLU as well as squared ReLU, although the latter tends to achieve a higher margin value for $\ell_\infty$-maximizing algorithms. Signum appears to outperform Adam in terms of $\ell_\infty$ margin, which is expected considering that the $\ell_\infty$-margin-maximization properties of Adam may hinge on its similarity to sign gradient descent, of which Signum is a closer approximation. Also, we observe that NGD is second-best to Muon when maximizing $\|\cdot\|_{\mathrm{msp}}$, a phenomenon perhaps explained by the fact that the spectral norm of the output layer is its $\ell_2$ norm. In Appendix D (Figure 2) we compare Muon-Adam with Muon and Adam, and show that it

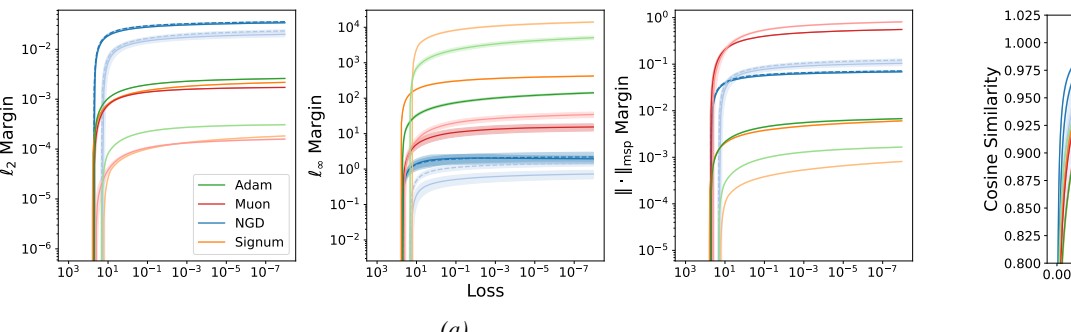

*Figure 1.* (a) Margin values vs. loss for different optimizers. A lighter/darker color signifies the squared-ReLU / ReLU activations respectively. Dotted lines represent optimizers with momentum disabled. Lines are mean values over 10 random seeds, while filled areas represent one standard deviation. (b) Cosine similarity to last iterate $\left\langle \frac{\boldsymbol{\theta}_t}{\|\boldsymbol{\theta}_t\|_2}, \frac{\boldsymbol{\theta}_{\text{last}}}{\|\boldsymbol{\theta}_{\text{last}}\|_2} \right\rangle$, plotted on a normalized linear time scale.

maximizes the appropriate margin.

To test the assumption of directional convergence (T2), Figure 1b shows the cosine similarity of the iterates to the last iterate, $\left\langle \frac{\boldsymbol{\theta}_t}{\|\boldsymbol{\theta}_t\|_2}, \frac{\boldsymbol{\theta}_{\text{last}}}{\|\boldsymbol{\theta}_{\text{last}}\|_2} \right\rangle$. Note that here the x-axis is a linear time scale normalized with respect to the total training time. We observe, for example, that for all algorithms, alignment is above 0.99 for the entire second half of the trajectory, suggesting that directional convergence indeed holds. All trajectories have the margin $\gamma(\boldsymbol{\theta}_t)$ bounded away from 0 for the entire late phase of training, validating $\gamma(\bar{\boldsymbol{\theta}}) > 0$. Also, Assumption (T1) holds in all experiments.

## 7. Conclusion

In this work we examined the properties of the popular momentum-based optimizers Adam and Muon on smooth homogeneous models. The study is conducted through the unifying perspective of approximate steepest descent, a framework we believe to be general and widely applicable to first-order optimization methods related to the steepest descent family. We crucially show that the momentum mechanism is asymptotically faithful to the significant gradient coordinates, when the learning rate decays. Our treatment of Muon and Muon-Signum is only a special case of compositions of normalized momentum steepest descent algorithms, while results for Adam and Muon-Adam rely directly on the framework of approximate steepest descent.

Several important questions remain open. First, our results for momentum-based optimizers hold also for *non-smooth* models under a strong trajectory Assumption (T3), as discussed in Section 4. It is unclear whether these algorithms have a provable margin-maximization bias for non-smooth models, notably ReLU networks, with no such assumptions, or whether Assumption (T3) can be formally proved in certain settings. Second, our results assume directional convergence of the parameters. For gradient descent, the implicit bias in homogeneous models was analyzed by Lyu and Li (2019) before Ji and Telgarsky (2020) formally proved

directional convergence; a natural question is whether a directional-convergence guarantee can also be proved for Adam and Muon. Third, the implicit bias of gradient descent was analyzed also for certain non-homogeneous models (Nacson et al., 2019a; Kunin et al., 2022; Cai et al., 2025), and it would be interesting to show such results for other optimizers.

Finally, exploring the theoretical and practical implications of our results is an intriguing research direction. In which settings can generalization of models be deliberately improved with an informed choice of optimizer? Are privacy attacks based on satisfaction of KKT conditions, as shown in Haim et al. (2022); Buzaglo et al. (2023); Oz et al. (2024); Golbari et al. (2025) for gradient descent, also feasible for Adam and Muon? What are the implications of the implicit bias in these optimizers for adversarial robustness (Vardi et al., 2022; Frei et al., 2023)? We hope that our results will help advance understanding of the above questions.

## Impact Statement

This paper presents work whose goal is to advance the field of machine learning. There are many potential societal consequences of our work, none of which we feel must be specifically highlighted here.

## Acknowledgments

This work was supported by the Israel Science Foundation (grant No. 2574/25), a research grant from Mortimer Zuckerman (the Zuckerman STEM Leadership Program), and research grants from the Center for New Scientists at the Weizmann Institute of Science, and the Shimon and Golde Picker – Weizmann Annual Grant.

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

# A. Clarke Subgradients and Chain Rule

Our notion of the subgradient of a function $f = f(\boldsymbol{\theta}) : \mathbb{R}^p \to \mathbb{R}$ is that of Clarke (1975):

$$\partial f(\boldsymbol{\theta}) := \mathrm{conv} \left\{ \lim_{k \to \infty} \nabla f(\boldsymbol{\theta}_k) \mid \lim_{k \to \infty} \boldsymbol{\theta}_k = \boldsymbol{\theta}, f \text{ is differentiable at } \boldsymbol{\theta}_k \right\}, \tag{13}$$

where $\mathrm{conv}(\cdot)$ is the convex hull of a set (the set of finite convex combinations of points from that set).

The following basic chain rule holds for all locally Lipschitz functions.

**Theorem A.1** (Theorem 2.3.9 and 2.3.10 in (Clarke, 1990)). *Let $f_1, \ldots, f_m : \mathbb{R}^p \to \mathbb{R}, \mathcal{L} : \mathbb{R}^m \to \mathbb{R}$ be locally Lipschitz functions and define $\mathbf{f} = (f_1, \ldots, f_m)$. Let $(\mathcal{L} \circ \mathbf{f})(\boldsymbol{\theta}) = \mathcal{L}(f_1(\boldsymbol{\theta}), \ldots, f_m(\boldsymbol{\theta})) : \mathbb{R}^p \to \mathbb{R}$ be the composition of $\mathcal{L}$ with $f$. Then, it holds:*

$$\partial(\mathcal{L} \circ \mathbf{f})(\boldsymbol{\theta}) \subseteq \mathrm{conv} \left\{ \sum_{i=1}^m \alpha_i \mathbf{h}_i : \boldsymbol{\alpha} \in \partial\mathcal{L}(f_1(\boldsymbol{\theta}), \ldots, f_m(\boldsymbol{\theta})), \mathbf{h}_i \in \partial f_i(\boldsymbol{\theta}) \right\} . \tag{14}$$

The following corollary holds for a smooth $\mathcal{L}$, and implies in our setting for the loss $\mathcal{L}(\boldsymbol{\theta}) = \mathcal{L}(f(\mathbf{x}_1; \boldsymbol{\theta}), \ldots, f(\mathbf{x}_m; \boldsymbol{\theta}))$ that any $\mathbf{g} \in \partial\mathcal{L}(\boldsymbol{\theta})$ is given as a sum $\mathbf{g} = \sum_{i=1}^m \ell'(y_i f(\mathbf{x}_i; \boldsymbol{\theta})) y_i \mathbf{h}_i$ where $\ell'$ is the derivative of the per-sample loss, and $\mathbf{h}_i \in \partial f(\mathbf{x}_i; \boldsymbol{\theta})$.

**Corollary A.2** (Chain Rule with an Outer Smooth Function). *Let $f_1, \ldots, f_m : \mathbb{R}^p \to \mathbb{R}$ be locally Lipschitz and $\mathcal{L} : \mathbb{R}^m \to \mathbb{R}$ be $C^1$. Then*

$$\partial(\mathcal{L} \circ \mathbf{f})(\boldsymbol{\theta}) \subseteq \left\{ \sum_{i=1}^m \alpha_i \mathbf{h}_i : \boldsymbol{\alpha} \in \partial\mathcal{L}(f_1(\boldsymbol{\theta}), \ldots, f_m(\boldsymbol{\theta})), \mathbf{h}_i \in \partial f_i(\boldsymbol{\theta}) \right\} . \tag{15}$$

*Proof.* Denote

$$A = \left\{ \sum_{i=1}^m \alpha_i \mathbf{h}_i : \boldsymbol{\alpha} \in \partial\mathcal{L}(f_1(\boldsymbol{\theta}), \ldots, f_m(\boldsymbol{\theta})), \mathbf{h}_i \in \partial f_i(\boldsymbol{\theta}) \right\} .$$

Using Theorem A.1, it suffices to show that $\mathrm{conv} A \subseteq A$. Indeed, let $n \in \mathbb{N}$, $\boldsymbol{\alpha}^{(j)} \in \partial\mathcal{L}(f_1(\boldsymbol{\theta}), \ldots, f_m(\boldsymbol{\theta})), \mathbf{h}_i^{(j)} \in \partial f_i(\boldsymbol{\theta})$ for $i \in [m], j \in [n]$ and let $\lambda_j > 0, j \in [n]$ with $\sum_{j=1}^n \lambda_j = 1$. Since $\mathcal{L} \in C^1$, $\partial\mathcal{L} = \{\nabla\mathcal{L}\}$ is unique at any point, so in fact $\forall j \in [n] : \boldsymbol{\alpha}^{(j)} = \nabla\mathcal{L}(f_1(\boldsymbol{\theta}), \ldots, f_m(\boldsymbol{\theta})) =: \boldsymbol{\alpha}$, and

$$\sum_{j=1}^n \lambda_j \sum_{i=1}^m \alpha_i^{(j)} \mathbf{h}_i^{(j)} = \sum_{j=1}^n \lambda_j \sum_{i=1}^m \alpha_i \mathbf{h}_i^{(j)} = \sum_{i=1}^m \alpha_i \sum_{j=1}^n \lambda_j \mathbf{h}_i^{(j)} .$$

Since $\partial f_i(\boldsymbol{\theta})$ is convex by definition, $\sum_{j=1}^n \lambda_j \mathbf{h}_i^{(j)} \in \partial f_i(\boldsymbol{\theta})$, so we are finished. $\qquad\square$

We also consider the notion of Whitney-$C^1$ stratifiability. We refer the reader to Section 5.2 in Davis et al. (2020) for a technical introduction to the topic. For our purposes, it suffices to know that if $f : \mathbb{R}^p \to \mathbb{R}$ is Whitney-$C^1$ stratifiable, its graph is a finite union of $C^1$ manifolds (implying that (T3) follows from (T2) for some trajectories), with conditions along the boundaries allowing us to assume that $f$ admits a chain rule.

**Definition A.3.** $\boldsymbol{\theta}_t : [0, \infty) \to \mathbb{R}^p$ is an *arc* if it is absolutely continuous on every compact interval, or equivalently if there exists $\boldsymbol{\theta}_t' : [0, \infty) \to \mathbb{R}^p$ which is Lebesgue integrable on every interval $[0, t]$ so that

$$\forall t \geq 0 : \boldsymbol{\theta}_t = \boldsymbol{\theta}_0 + \int_0^t \boldsymbol{\theta}_s' ds .$$

In all the trajectories we discuss, $\boldsymbol{\theta}_t$ is defined as an integral over $\frac{d\boldsymbol{\theta}_t}{dt}$, so $\boldsymbol{\theta}_t$ is an arc.

**Theorem A.4** (Chain Rule for Arcs and Stratifiable Functions - Theorem 5.8 in (Davis et al., 2020)). *If $f : \mathbb{R}^p \to \mathbb{R}$ is locally Lipschitz and Whitney $C^1$-stratifiable, then for any arc $\boldsymbol{\theta}_t : [0, \infty) \to \mathbb{R}^p$, almost all $t \geq 0$, and all $\mathbf{g} \in \partial f(\boldsymbol{\theta}_t)$, it holds:*

$$\frac{df(\boldsymbol{\theta}_t)}{dt} = \left\langle \mathbf{g}, \frac{d\boldsymbol{\theta}_t}{dt} \right\rangle . \tag{16}$$

Finally, we also need a chain rule for the norm $\|\boldsymbol{\theta}_t\|$. We may circumvent the requirement that $\|\cdot\|$ be $C^1$-stratifiable with the following definition and theorem:

**Definition A.5** (Subdifferentially Regular Functions - Definition 5.3 in (Davis et al., 2020)). $f : \mathbb{R}^p \to \mathbb{R}$ is said to be subdifferentially regular if $\forall \boldsymbol{\theta} \in \mathbb{R}^p, \mathbf{g} \in \partial f(\boldsymbol{\theta})$,

$$f(\boldsymbol{\theta}') \geq f(\boldsymbol{\theta}) + \langle \mathbf{g}, \boldsymbol{\theta}' - \boldsymbol{\theta} \rangle + \mathrm{o}\left(\|\boldsymbol{\theta}' - \boldsymbol{\theta}\|\right) \quad \text{as } \boldsymbol{\theta}' \to \boldsymbol{\theta} .$$

In particular, any convex function, including any norm, satisfies the above inequality without an error term $\mathrm{o}\left(\|\boldsymbol{\theta}' - \boldsymbol{\theta}\|\right)$ and thus is subdifferentially regular.

**Theorem A.6** (Chain Rule for Arcs and Subdifferentially Regular Functions - Lemma 5.4 in (Davis et al., 2020)). *If* $f : \mathbb{R}^p \to \mathbb{R}$ *is locally Lipschitz and subdifferentially regular, then for any arc* $\boldsymbol{\theta}_t : [0, \infty) \to \mathbb{R}^p$, *almost all* $t \geq 0$, *and all* $\mathbf{g} \in \partial f(\boldsymbol{\theta}_t)$, *it holds:*

$$\frac{df(\boldsymbol{\theta}_t)}{dt} = \left\langle \mathbf{g}, \frac{d\boldsymbol{\theta}_t}{dt} \right\rangle . \tag{17}$$

For reference, the following is a standard characterization of the subdifferential of a norm:

$$\partial \|\boldsymbol{\theta}\| = \{\mathbf{v} \in \mathbb{R}^p \mid \langle \mathbf{v}, \boldsymbol{\theta} \rangle = \|\boldsymbol{\theta}\|, \|\mathbf{v}\|_\star \leq 1\} . \tag{18}$$

Note that by definition of the dual norm it holds that $\langle \mathbf{v}, \boldsymbol{\theta} \rangle \leq \|\boldsymbol{\theta}\| \|\mathbf{v}\|_\star$, so in fact when $\boldsymbol{\theta} \neq \mathbf{0}$, $\|\mathbf{v}\|_\star = 1$ for any $\mathbf{v} \in \partial \|\boldsymbol{\theta}\|$.

# B. Momentum (EMA)

## B.1. Discrete Momentum

For a parameter $\beta \in (0, 1)$ and a real-valued sequence $g_n, n \geq 1$, discrete momentum $m_n$ is defined as follows:

$$m_n = \beta m_{n-1} + (1 - \beta)g_n, \quad m_0 = 0 , \tag{19}$$

with the explicit solution:

$$m_n = (1 - \beta) \sum_{k=1}^{n} \beta^{n-k} g_k .$$

We give for reference the definition of Adam in the discrete case for a sequence of subgradients $\mathbf{g}_n$, parameters $\beta_1, \beta_2 \in (0, 1)$, a learning rate $\eta_n$ and a constant $\varepsilon \geq 0$ (in our analysis $\varepsilon = 0$). Square, division and square root are taken elementwise.

**Adam (Discrete):**

$$\begin{aligned}
\mathbf{m}_n &= \beta_1 \mathbf{m}_{n-1} + (1 - \beta_1)\mathbf{g}_n, \quad \mathbf{m}_0 = \mathbf{0} \\
\mathbf{v}_n &= \beta_2 \mathbf{v}_{n-1} + (1 - \beta_2)\mathbf{g}_n^2, \quad \mathbf{v}_0 = \mathbf{0} \\
\hat{\mathbf{m}}_n &= (1 - \beta_1^n)^{-1}\mathbf{m}_n, \quad \hat{\mathbf{v}}_n = (1 - \beta_2^n)^{-1}\mathbf{v}_n \\
\Delta\boldsymbol{\theta}_n &= -\eta_n \frac{\hat{\mathbf{m}}_n}{\sqrt{\hat{\mathbf{v}}_n} + \varepsilon} .
\end{aligned} \tag{20}$$

## B.2. Continuous Momentum

Notice that Equation (19) shows that the direction of update of $m_n$ is $g_n - m_{n-1}$:

$$m_n - m_{n-1} = (1 - \beta)(g_n - m_{n-1}) .$$

The natural analogue for a continuous time variable $t$ and a parameter $c > 0$ is therefore:

$$\frac{dm_t}{dt} = c(g_t - m_t), \quad m_0 = 0 .$$

This equation has the following explicit solution, clearly of a similar form to the discrete version:

$$m_t = \int_0^t c e^{-c(t-s)} g_s ds .$$

This form is identical to (Wang et al., 2021), with the caveat that they assumed $c = 1 - \beta$, a form restricting $c$ to small values (note that taking $c \to \infty$ induces the regime $m_t = g_t$). To clearly uncover the connection between $\beta$ and $c$, assume that a discrete momentum iteration takes a unit time $\Delta t = 1$, and that steps are small enough so that $g_s$ is roughly constant on $[t, t+1]$. Then

$$m_{t+1} - m_t = \int_0^{t+1} ce^{-c(t+1-s)}g_s ds - \int_0^t ce^{-c(t-s)}g_s ds = (e^{-c} - 1)m_t + \int_t^{t+1} ce^{-c(t+1-s)}g_s ds$$

$$\approx (e^{-c} - 1)m_t + (1 - e^{-c})g_t = (1 - e^{-c})(g_t - m_t) .$$

So $c$ is analogous to $-\log \beta$, hence spanning the entire range $(0, \infty)$. When $\beta$ is close to 1 we obtain $c \approx (1 - \beta)$.

In the definition of discrete Adam, a multiplicative factor of $(1 - \beta^n)^{-1}$ is used to correct the initial bias accrued when initializing $m_0 = 0$. The continuous analogue for the bias correction is $(1 - e^{-ct})^{-1}$, since for small $t$, $m_t \approx g_0 \int_0^t ce^{-c(t-s)}ds = g_0(1 - e^{-ct})$.

**Definition B.1.** We denote

$$\mathcal{F} = L_{\text{loc}}^\infty([0, \infty)), \quad \mathcal{G} = \left\{ g : [0, \infty) \to \mathbb{R} \mid \exists \tau > 0, \rho > 0 : g^2(t) > \rho \text{ a.e. on } [0, \tau] \right\} ,$$

where $L_{\text{loc}}^\infty([0, \infty))$ is the space of Lebesgue measurable functions $g(t) : [0, \infty) \to \mathbb{R}$ so that $g(t)$ is essentially bounded (bounded except on a measure zero set) on every compact interval $[a, b]$. We denote also for convenience

$$M_{[a,b]}(g) := \operatorname*{ess\,sup}_{[a,b]} |g| .$$

It is a standard fact that $L_{\text{loc}}^\infty([0, \infty)) \subseteq L_{\text{loc}}^p([0, \infty))$ for any $p \geq 1$, the latter being the space of functions $g$ with $\int_{[a,b]} |g|^p < \infty$ on every compact interval $[a, b]$. This allows us to define the momentum expression for $g$. The class $\mathcal{G}$ encapsulates Assumption (A1), facilitating a discussion of the Adam ratio.

**Definition B.2** (Momentum/EMA). Let $g(t) \in \mathcal{F}$. For a parameter $c > 0$ denote the following by EMA $(g, c)(t)$ or $A(g, c)(t)$ for short:

$$\text{EMA}(g, c)(t) = A(g, c)(t) := \int_0^t ce^{-c(t-s)}g(s)ds .$$

Also denote for convenience:

$$I(g, c)(t) = \int_0^t e^{cs}g(s)ds, \quad I_\infty(g, c) = \lim_{t \to \infty} I(g, c)(t) ,$$

so $A(g, c)(t) = ce^{-ct}I(g, c)(t)$.

We prove a series of useful lemmas about the properties of the EMA.

**Lemma B.3** (Momentum ODE). *Let $g(t) \in \mathcal{F}$ and $c > 0$. Then for almost any $t \geq 0$,*

$$\frac{dA(g, c)}{dt} = c(g(t) - A(g, c)(t)) .$$

*Proof.* Since $g$ is Lebesgue integrable, by differentiation rules and the fundamental theorem of calculus we have

$$\frac{dA(g, c)}{dt} = -c^2 e^{-ct}I(g, c)(t) + ce^{-ct}\frac{dI(g, c)}{dt} = -cA(g, c)(t) + ce^{-ct}e^{ct}g(t) = c(g(t) - A(g, c)(t))$$

$\square$

**Lemma B.4** (Uniform Bound on Adam Ratio). *Let $g(t) \in \mathcal{F}$ and $c_1 > \frac{c_2}{2} > 0$. Then,*

$$|A(g, c_1)(t)| \leq A(|g|, c_1)(t) \leq \frac{c_1}{\sqrt{c_2(2c_1 - c_2)}}\sqrt{A(g^2, c_2)(t)} .$$

*In particular,*

$$A(|g|, c_1)(t) \leq \mathcal{O}\left(\sqrt{A(g^2, c_2)(t)}\right) ,$$

*and*

$$A(|g|, c_1)(t) \leq \sqrt{A(g^2, c_1)(t)} .$$

*Proof.* By the triangle inequality and Cauchy-Schwarz on the inner product space of $L^2$-integrable functions on $[0, t]$, we get

$$|A(g, c_1)(t)| \leq \int_0^t c_1 e^{-c_1(t-s)} |g(s)| \, ds = \int_0^t c_1 e^{-(c_1 - \frac{c_2}{2})(t-s)} e^{-\frac{c_2}{2}(t-s)} |g(s)| \, ds$$

$$\leq c_1 \left( \int_0^t e^{-c_2(t-s)} g^2(s) ds \right)^{\frac{1}{2}} \left( \int_0^t e^{-(2c_1 - c_2)(t-s)} ds \right)^{\frac{1}{2}}$$

$$\leq \frac{c_1}{\sqrt{c_2}} \sqrt{A(g^2, c_2)(t)} \left( \frac{1 - e^{-(2c_1 - c_2)t}}{2c_1 - c_2} \right)^{\frac{1}{2}}$$

$$\leq \frac{c_1}{\sqrt{c_2(2c_1 - c_2)}} \sqrt{A(g^2, c_2)(t)}$$

$\square$

**Lemma B.5** (Asymptotic Relations). *Let $c > 0$ and $0 \leq g(t) \in \mathcal{F}$. Assume $g$ is not a.e. 0.*

1. *If $I_\infty(g, c) < \infty$ then $\frac{A(g,c)(t)}{e^{-ct}} \overset{t \to \infty}{\Longrightarrow} cI_\infty(g, c) > 0$ and*

$$\forall F \in \mathcal{F}, I_\infty(F, c) < \infty : \frac{A(F, c)}{A(g, c)} \longrightarrow \frac{I_\infty(F, c)}{I_\infty(g, c)} .$$

2. *If $I_\infty(g, c) = \infty$ then $\frac{A(g,c)(t)}{e^{-ct}} \to \infty$ and*

   (a)
$$\forall t_0 \geq 0 : \left| \frac{\int_{t_0}^t ce^{-c(t-s)} g(s) ds}{A(g, c)(t)} - 1 \right| \leq \mathcal{O} \left( \frac{e^{-ct}}{A(g, c)(t)} \right) \overset{t \to \infty}{\to} 0 .$$

   (b) *For any $F(t) \in \mathcal{F}$ and $C > 0, t_0 \geq 0$:*
$$(\forall t \geq t_0 : F(t) \leq Cg(t)) \Rightarrow A(F, c)(t) \leq CA(g, c)(t)(1 + \text{o}(1)) ,$$
$$(\forall t \geq t_0 : F(t) \geq Cg(t)) \Rightarrow A(F, c)(t) \geq CA(g, c)(t)(1 - \text{o}(1)) .$$

   (c) *If eventually $g(t) > 0$, for any $F(t) \in \mathcal{F}$ with $\frac{F(t)}{g(t)} \overset{t \to \infty}{\longrightarrow} C \in [-\infty, \infty]$ it holds that*
$$\frac{A(F, c)}{A(g, c)} \overset{t \to \infty}{\longrightarrow} C .$$

3. *In both cases ($I_\infty(g, c) = \infty, I_\infty(g, c) < \infty$), for any $0 \leq F(t) \in \mathcal{F}$:*
$$F(t) \leq \mathcal{O}(g(t)) \Rightarrow A(F, c)(t) \leq \mathcal{O}(A(g, c)(t)) ,$$
$$F(t) \geq \Omega(g(t)) \Rightarrow A(F, c)(t) \geq \Omega(A(g, c)(t)) .$$

*Proof.* 1. By definition $A(g, c)(t) = ce^{-ct} \int_0^t e^{cs} g(s) ds$, so

$$\lim_{t \to \infty} \frac{A(g, c)(t)}{e^{-ct}} = \lim_{t \to \infty} cI(g, c)(t) = cI_\infty(g, c) .$$

And for any $g, F$ with $I_\infty(F, c) < \infty$,

$$\lim_{t \to \infty} \frac{A(F, c)(t)}{A(g, c)(t)} = \lim_{t \to \infty} \frac{A(F, c)(t)}{ce^{-ct}} \frac{ce^{-ct}}{A(g, c)(t)} = \frac{I_\infty(F, c)}{I_\infty(g, c)} .$$

2. (a) Note that since $g \geq 0$,

$$\int_{t_0}^t ce^{-c(t-s)} g(s) ds \leq A(g, c)(t) \leq M_{[0, t_0]}(g) \cdot (e^{-c(t-t_0)} - e^{-ct}) + \int_{t_0}^t ce^{-c(t-s)} g(s) ds .$$

Since $I_\infty(g, c) = \infty$, in particular $A(g, c)(t) > 0$ for large enough $t$. Dividing by $A(g, c)(t)$ we get

$$1 - \frac{M_{[0,t_0]}(g) \cdot (e^{-c(t-t_0)} - e^{-ct})}{A(g, c)(t)} \le \frac{\int_{t_0}^t ce^{-c(t-s)}g(s)ds}{A(g, c)(t)} \le 1 \,,$$

$$\left| \frac{\int_{t_0}^t ce^{-c(t-s)}g(s)ds}{A(g, c)(t)} - 1 \right| \le \frac{M_{[0,t_0]}(g) \cdot (e^{-c(t-t_0)} - e^{-ct})}{A(g, c)(t)} = \mathcal{O}\left( \frac{e^{-ct}}{A(g, c)(t)} \right) = \mathrm{o}\,(1) \,.$$

(b) Let $C > 0, t_0 \ge 0$ with $\forall t \ge t_0 : F(t) \le Cg(t)$. Then,

$$A(F, c)(t) \le M_{[0,t_0]}(F) \cdot (e^{-c(t-t_0)} - e^{-ct}) + C \int_{t_0}^t ce^{-c(t-s)}g(s)ds \,.$$

Therefore by item 2(a),

$$A(F, c)(t) \le \mathcal{O}\left(e^{-ct}\right) + CA(g, c)(t)(1 + \mathrm{o}\,(1)) \le \mathrm{o}\,(A(g, c)) + CA(g, c)(t)(1 + \mathrm{o}\,(1)) = CA(g, c)(t)(1 + \mathrm{o}\,(1)) \,.$$

The other direction is completely symmetrical.

(c) First address the case $C \in (0, \infty)$. Since $C > 0$, by the previous item,

$$\frac{A(F, c)(t)}{A(g, c)(t)} \le C(1 + \mathrm{o}\,(1)) \overset{t \to \infty}{\Longrightarrow} C \,,$$

so

$$\limsup_{t \to \infty} \frac{A(F, c)(t)}{A(g, c)(t)} \le C \,.$$

In the same fashion

$$\liminf_{t \to \infty} \frac{A(F, c)(t)}{A(g, c)(t)} \ge C \,,$$

showing the limit. If $C = 0$, then fixing $\varepsilon > 0$, $\frac{|F| + \varepsilon g}{g} \overset{t \to \infty}{\Longrightarrow} \varepsilon$, implying by the previous case

$$\frac{A(|F|, c)}{A(g, c)} + \varepsilon = \frac{A(|F|, c) + \varepsilon A(g, c)}{A(g, c)} = \frac{A(|F| + \varepsilon g, c)}{A(g, c)} \overset{t \to \infty}{\Longrightarrow} \varepsilon \,,$$

and so

$$\left| \frac{A(F, c)}{A(g, c)} \right| \le \frac{A(|F|, c)}{A(g, c)} \overset{t \to \infty}{\Longrightarrow} 0 \,.$$

For $C \in (-\infty, 0)$, applying the case $C > 0$ with $-F$ suffices, since $\frac{-F}{g} \to -C > 0$ and $A(-F, c) = -A(F, c)$. This finishes for a finite $C$. If $C = \infty$ then the above shows $\liminf_{t \to \infty} \frac{A(F,c)(t)}{A(g,c)(t)} \ge C'$ for any $C' > 0$, showing $\liminf_{t \to \infty} \frac{A(F,c)(t)}{A(g,c)(t)} = \infty$, and symmetrically for $C = -\infty$.

3. If $I_\infty(g, c) = \infty$ then the result follows from item 2(b). Assume $I_\infty(g, c) < \infty$. If $F(t) \le \mathcal{O}\,(g(t))$ then $I_\infty(F, c) < \infty$, so the result follows from item 1. If $F(t) \ge \Omega(g(t))$, consider two cases: if $I_\infty(F, c) < \infty$ then the result again follows from item 1, and if $I_\infty(F, c) = \infty$ then $\frac{A(F,c)}{A(g,c)} = \frac{I(F,c)(t)}{I(g,c)(t)} \to \infty$ and in particular $A(F, c) \ge \Omega(A(g, c))$.

$\square$

**Corollary B.6.** *Let $c > 0$ and $g \in \mathcal{F}$.*

1. *If eventually $g(t) \le M$ for some $M \in \mathbb{R}$ then $A(g, c)(t) \le M + \mathrm{o}\,(1)$, and if eventually $g(t) \ge M$ then $A(g, c)(t) \ge M - \mathrm{o}\,(1)$.*

2. *If $\lim_{t \to \infty} g(t) = C \in [-\infty, \infty]$ then $\lim_{t \to \infty} A(g, c)(t) = C$.*

*Proof.* Let $M \in \mathbb{R}$, and $t_0$ with $\forall t \ge t_0 : g(t) \le M$. Then

$$A(g, c)(t) \le M_{[0,t_0]}(g)(e^{-c(t-t_0)} - e^{-ct}) + M(1 - e^{-c(t-t_0)}) \le M + \mathrm{o}\,(1) \,.$$

The lower bound is symmetrical.

For the limit, choose $h \equiv 1 \in \mathcal{F}$. Clearly $I_\infty(h, c) = \infty$, and $A(h, c) = 1 - e^{-ct}$. If $\lim_{t \to \infty} g(t) = \lim_{t \to \infty} \frac{g(t)}{h(t)} = C$ then $\lim_{t \to \infty} \frac{A(g,c)}{A(h,c)} = C$ by Lemma B.5, implying $\lim_{t \to \infty} A(g, c)(t) = C$. $\qquad\square$

The following lemma gives sufficient conditions for $\frac{A(g,c)}{g}$ converging to a constant ratio; see the following corollary for a simpler condition for differentiable functions.

**Lemma B.7.** *Let $c > 0$ and $0 < g(t) \in \mathcal{F}$. Assume that*

1. $\lim_{t \to \infty} \frac{e^{-ct}}{g(t)} = 0$.

2. *There exists $k \in [0, c)$ so that for every fixed $u > 0$, $\frac{g(t-u)}{g(t)} \xrightarrow{t \to \infty} e^{ku}$.*

3. *There exists $M(u) \geq 0$ with $\int_0^\infty c e^{-cu} M(u) du < \infty$, and $t_0 \geq 0$ so that $\forall t \geq t_0, 0 < u < t - t_0 : \frac{g(t-u)}{g(t)} \leq M(u)$.*

*Then $\frac{A(g,c)(t)}{g(t)} \to \frac{c}{c-k}$.*

*Proof.* Denote $H_t(u) = \mathbf{1}_{\{u \leq t - t_0\}} \cdot c e^{-cu} \frac{g(t-u)}{g(t)}$. For any $t \geq t_0$,

$$
\begin{aligned}
\frac{A(g, c)(t)}{g(t)} &= \frac{1}{g(t)} \int_0^{t_0} c e^{-c(t-s)} g(s) ds + \int_{t_0}^t c e^{-c(t-s)} \frac{g(s)}{g(t)} ds = \\
&\leq \frac{M_{[0,t_0]}(g)}{g(t)} \left( e^{-c(t-t_0)} - e^{-ct} \right) + \int_0^{t-t_0} c e^{-cu} \frac{g(t-u)}{g(t)} du = \\
&= \frac{M_{[0,t_0]}(g)}{g(t)} \left( e^{-c(t-t_0)} - e^{-ct} \right) + \int_0^\infty H_t(u) du .
\end{aligned}
$$

And since $g(t) > 0, A(g, c)(t) > 0$,

$$
\frac{A(g, c)(t)}{g(t)} \geq \int_0^{t-t_0} c e^{-cu} \frac{g(t-u)}{g(t)} du = \int_0^\infty H_t(u) du .
$$

Recall that by hypothesis $\frac{e^{-ct}}{g(t)} \xrightarrow{t \to \infty} 0$. Also, by hypothesis, $H_t(u) \leq c e^{-cu} M(u)$ with $\int_0^\infty c e^{-cu} M(u) du < \infty$. Finally note $H_t(u) \xrightarrow{t \to \infty} c e^{-(c-k)u}$ pointwise. Therefore by the dominated convergence theorem on $[0, \infty)$:

$$
\lim_{t \to \infty} \frac{A(g, c)(t)}{g(t)} = \lim_{t \to \infty} \int_0^\infty H_t(u) du = \int_0^\infty \lim_{t \to \infty} H_t(u) du = \int_0^\infty c e^{-(c-k)u} du = \frac{c}{c-k}
$$

$\qquad\square$

The following useful corollary is stated for $g(t) > 0$, but if $g(t) < 0$ it applies to $-g$ and $A(-g, c) = -A(g, c)$. Therefore for $g(t) \neq 0$ with a constant sign, the condition $-\frac{g'}{g} = -\frac{d \log|g|}{dt} \to k$ implies items 1-3 as written.

**Corollary B.8** (Asymptotic Momentum-Function Ratio ). *Let $c > 0$ and $0 < g(t) \in \mathcal{F}$. Assume $g(t)$ is differentiable almost everywhere, locally absolutely continuous and $\operatorname{ess} \lim_{t \to \infty} -\frac{g'(t)}{g(t)} = \operatorname{ess} \lim_{t \to \infty} -\frac{d \log g(t)}{dt} = k \in [0, \infty)$. Then:*

1. *If $k > c$ then $\frac{A(g,c)(t)}{e^{-ct}} \xrightarrow{t \to \infty} c I_\infty(g, c) < \infty$ and in particular $\frac{A(g,c)(t)}{g(t)} \to \infty$.*

2. *If $k = c$ then $\frac{A(g,c)(t)}{g(t)} = c \cdot \frac{\int_0^t G}{G(t)} \xrightarrow{t \to \infty} \infty$ for $G(t) = e^{ct} g(t)$.*

3. *If $k < c$ then $\frac{A(g,c)(t)}{g(t)} \xrightarrow{t \to \infty} \frac{c}{c-k}$.*

*Proof.* For any $k \in [0, \infty)$, if $-\frac{d \log g}{dt} \xrightarrow{t \to \infty} k$ then, since $g$ is locally absolutely continuous, we have by integration on $[0, t]$

$$
\log g(t) = -kt + o(t) \quad \Rightarrow \quad g(t) = e^{-kt + o(t)}
$$

This implies $\int_0^\infty e^{ct} g(t) dt < \infty$, thus the case $k > c$ is simply a reiteration of Lemma B.5 item 1.

For the case $k = c$, denote $G(t) = e^{ct} g(t)$, so it holds that $\frac{d \log G}{dt} \overset{t \to \infty}{\longrightarrow} c - c = 0$ and:

$$A(g,c)(t) = ce^{-ct} \int_0^t G(s) ds$$

$$\frac{A(g,c)(t)}{g(t)} = c \cdot \frac{\int_0^t G(s) ds}{G(t)}$$

We claim this expression tends to $\infty$. Indeed, for any $\varepsilon > 0$ there exists $t_0 \geq 0$ with for almost all $t \geq t_0 : \left| \frac{d \log G}{dt} \right| \leq \varepsilon$, hence it holds by integration on any interval $[s,t], t \geq s \geq t_0$ that $\log \frac{G(t)}{G(s)} \leq \varepsilon(t-s)$, hence $G(t) \leq G(s) e^{\varepsilon(t-s)}$ and

$$\int_0^t \frac{G(s)}{G(t)} ds \geq \int_{t_0}^t \frac{G(s)}{G(t)} ds \geq \int_{t_0}^t e^{-\varepsilon(t-s)} ds = \frac{1}{\varepsilon} \left( 1 - e^{-\varepsilon(t-t_0)} \right) \overset{t \to \infty}{\longrightarrow} \frac{1}{\varepsilon} .$$

Since $\varepsilon$ was arbitrary this proves $\int_0^t \frac{G(s)}{G(t)} ds \overset{t \to \infty}{\longrightarrow} \infty$.

For the case $k < c$ we use Lemma B.7. First note that $\frac{e^{-ct}}{g(t)} = e^{-(c-k)t + o(t)} \to 0$. Also, for any fixed $u$,

$$\log \frac{g(t-u)}{g(t)} = - \int_{t-u}^t \frac{d \log g(s)}{ds} ds \overset{t \to \infty}{\longrightarrow} ku ,$$

$$\frac{g(t-u)}{g(t)} \overset{t \to \infty}{\longrightarrow} e^{ku} .$$

And there exists $t_0$ and $k < c' < c$ for which $\forall t \geq t_0 : \frac{-d \log g}{dt} < c'$, so $\forall t \geq t_0, 0 < u < t - t_0$:

$$\log \frac{g(t-u)}{g(t)} \leq c'u ,$$

$$\frac{g(t-u)}{g(t)} \leq e^{c'u} =: M(u) .$$

With $\int_0^\infty ce^{-cu} M(u) du < \infty$. Thus the conditions of Lemma B.7 hold, implying $\frac{A(g,c)(t)}{g(t)} \overset{t \to \infty}{\longrightarrow} \frac{c}{c-k}$. $\qquad \square$

**Lemma B.9** (Adam Ratio is Bounded at Initialization). *Let $g(t) \in \mathcal{F} \cap \mathcal{G}$, and let $\rho, \tau > 0$ for which $g^2(t) > \rho$ a.e. on $[0, \tau]$. Denote $m_t = A(g, c_1)(t), v_t = A(g^2, c_2)(t)$ for $c_1 \geq c_2 > 0$, and $\hat{m}_t = (1 - e^{-c_1 t})^{-1} m_t, \hat{v}_t = (1 - e^{-c_2 t})^{-1} v_t$. Then $\frac{\hat{m}_t}{\sqrt{\hat{v}_t}}$ is bounded on $(0, \tau]$.*

*Proof.* For any $s \in (0, \tau]$ it holds that

$$|\hat{m}_s| = \frac{|m_s|}{1 - e^{-c_1 s}} \leq \frac{\int_0^s c_1 e^{-c_1(s-r)} |g_r| dr}{1 - e^{-c_1 s}} \leq \frac{M_{[0,\tau]}(g) \int_0^s c_1 e^{-c_1(s-r)} dr}{1 - e^{-c_1 s}} = M_{[0,\tau]}(g) < \infty .$$

And that

$$\hat{v}_s = \frac{v_s}{1 - e^{-c_2 s}} = \frac{\int_0^s c_2 e^{-c_2(s-r)} g^2(r) dr}{1 - e^{-c_2 s}} \geq \frac{\rho \int_0^s c_2 e^{-c_2(s-r)} dr}{1 - e^{-c_2 s}} = \rho > 0 .$$

Therefore

$$\forall s \in (0, \tau] : \left| \frac{\hat{m}_s}{\sqrt{\hat{v}_s}} \right| \leq \frac{M_{[0,\tau]}(g)}{\sqrt{\rho}} < \infty .$$

$\qquad \square$

**Lemma B.10** (Adaptation of Lemma A.4 in Zhang et al. (2024)). *Let $c_1 \geq c_2 > 0$ and $0 < \eta(t) \in \mathcal{F}$ non-increasing with $\int_0^\infty \eta(t) dt = \infty$. Let $g_t = g(t) \in \mathcal{F} \cap \mathcal{G}$ with $g^2(t) < 1 - \delta$ eventually for some $\delta > 0$. Denote $m_t = A(g, c_1)(t), v_t =$*

$A(g^2, c_2)(t)$ *for* $c_1 \geq c_2 > 0$, *and* $\hat{m}_t = (1 - e^{-c_1 t})^{-1} m_t$, $\hat{v}_t = (1 - e^{-c_2 t})^{-1} v_t$. *Then*

$$\left| \int_0^t \eta(s) \frac{\hat{m}_s}{\sqrt{\hat{v}_s}} ds \right| \leq \int_0^t \eta(s) ds + \mathcal{O}\left( \left( \int_0^t \eta(s) ds \right)^{\frac{1}{2}} \right).$$

*And therefore,*

$$\frac{\left| \int_0^t \eta(s) \frac{\hat{m}_s}{\sqrt{\hat{v}_s}} ds \right|}{\int_0^t \eta(s) ds} \leq 1 + o(1).$$

*Proof.* From the hypothesis $g \in \mathcal{G}$ let $\tau > 0, \rho > 0$ with $g^2(t) > \rho$ a.e. on $[0, \tau]$, and choose $t_0 < \min\{\tau, \frac{\log 2}{c_1}\}$. By Lemma B.9 and since $\eta \in \mathcal{F}$, we have that $\eta(s) \frac{\hat{m}_s}{\sqrt{\hat{v}_s}}$ is bounded on $(0, t_0]$, so

$$\left| \int_0^{t_0} \eta(s) \frac{\hat{m}_s}{\sqrt{\hat{v}_s}} ds \right| < \infty.$$

Thus we focus now on $[t_0, t]$. Denote $q(s) = \frac{\sqrt{1 - e^{-c_2 s}}}{1 - e^{-c_1 s}}$. By Cauchy-Schwarz,

$$\left| \int_{t_0}^t \eta(s) q(s) \frac{m_s}{\sqrt{v_s}} ds \right| \leq \underbrace{\left( \int_{t_0}^t \eta(s) q^2(s) ds \right)^{\frac{1}{2}}}_{\star} \underbrace{\left( \int_{t_0}^t \eta(s) \frac{m_s^2}{v_s} ds \right)^{\frac{1}{2}}}_{\star\star}.$$

It suffices to show that $\star, \star\star$ are each at most $\int_0^t \eta(s) ds + \mathcal{O}(1)$. Since $\int_0^\infty \eta(s) ds = \infty$, this is equivalent to showing that $\star, \star\star$ are each at most $\int_{t_a}^t \eta(s) ds + \mathcal{O}(1)$ for a fixed $t_a$.

For $\star$, using $\forall x \in [0, \frac{1}{2}] : \frac{1}{(1-x)^2} \leq 1 + 6x$, notice that $q^2(t) \leq \frac{1}{(1 - e^{-c_1 t})^2} \leq 1 + 6e^{-c_1 t}$ for all $t \geq \frac{\log 2}{c_1}$, so for all $t \geq \frac{\log 2}{c_1}$ it holds that

$$\int_{t_0}^t \eta(s) q^2(s) ds \leq \int_{t_0}^{\frac{\log 2}{c_1}} \eta(s) q^2(s) ds + \int_{\frac{\log 2}{c_1}}^t \eta(s) ds + 6 \int_{\frac{\log 2}{c_1}}^t \eta(s) e^{-c_1 s} ds.$$

The first term is fixed and finite since $\eta(s) q^2(s)$ is bounded on the fixed interval $[t_0, \frac{\log 2}{c_1}]$. The third term is $\mathcal{O}(1)$ since $\eta(t)$ is bounded, so $\int_{\frac{\log 2}{c_1}}^\infty \eta(s) e^{-c_1 s} ds$ converges. This finishes for $\star$.

For $\star\star$, by Lemma B.4,

$$m_t^2 = A(g, c_1)^2 \leq A(g^2, c_1),$$

so

$$\int_{t_0}^t \eta(s) \frac{m_s^2}{v_s} ds \leq \int_{t_0}^t \frac{\eta(s)}{v_s} \int_0^s c_1 e^{-c_1(s-r)} g_r^2 dr ds. \tag{21}$$

It suffices to show that (21) is at most $\int_{t_0}^t \eta(s) ds + \mathcal{O}(1)$.

By Lemma B.3, $g_r^2 = \frac{1}{c_2} \frac{dv_r}{dr} + v_r$, and integration by parts gives

$$\int_0^s \frac{c_1}{c_2} e^{-c_1(s-r)} \frac{dv_r}{dr} dr = \frac{c_1}{c_2} \left( v_r e^{-c_1(s-r)} \Big|_0^s - \int_0^s c_1 e^{-c_1(s-r)} v_r dr \right)$$

$$\overset{v_0 = 0}{=} \frac{c_1}{c_2} \left( v_s - \int_0^s c_1 e^{-c_1(s-r)} v_r dr \right).$$

So,

$$\int_0^s c_1 e^{-c_1(s-r)} g_r^2 dr = \int_0^s \left( \frac{c_1}{c_2} e^{-c_1(s-r)} \frac{dv_r}{dr} + c_1 e^{-c_1(s-r)} v_r \right) dr = \frac{c_1}{c_2} v_s - \left( \frac{c_1}{c_2} - 1 \right) \int_0^s c_1 e^{-c_1(s-r)} v_r dr,$$

and

$$\text{RHS of (21)} = \frac{c_1}{c_2} \int_{t_0}^t \eta(s) ds - \left(\frac{c_1}{c_2} - 1\right) \int_{t_0}^t \eta(s) \int_0^s c_1 e^{-c_1(s-r)} \frac{v_r}{v_s} dr ds$$

$$= \int_{t_0}^t \eta(s) ds + \left(\frac{c_1}{c_2} - 1\right) \int_{t_0}^t \eta(s) ds - \left(\frac{c_1}{c_2} - 1\right) \int_{t_0}^t \eta(s) \int_0^s c_1 e^{-c_1(s-r)} \frac{v_r}{v_s} dr ds$$

$$= \int_{t_0}^t \eta(s) ds + \frac{c_1 - c_2}{c_2} \left( \int_{t_0}^t \eta(s) ds - \int_{t_0}^t \eta(s) \int_0^s c_1 e^{-c_1(s-r)} \frac{v_r}{v_s} dr ds \right)$$

$$= \int_{t_0}^t \eta(s) ds + \frac{c_1 - c_2}{c_2} \left( \int_{t_0}^t \eta(s) \left( 1 - \int_0^s c_1 e^{-c_1(s-r)} \frac{v_r}{v_s} dr \right) ds \right)$$

$$= \int_{t_0}^t \eta(s) ds + \frac{c_1 - c_2}{c_2} \left( \int_{t_0}^t \eta(s) \left( e^{-c_1 s} + \int_0^s c_1 e^{-c_1(s-r)} dr - \int_0^s c_1 e^{-c_1(s-r)} \frac{v_r}{v_s} dr \right) ds \right)$$

$$= \int_{t_0}^t \eta(s) ds + \frac{c_1 - c_2}{c_2} \left( \int_{t_0}^t \eta(s) e^{-c_1 s} ds + \int_{t_0}^t \eta(s) \int_0^s c_1 e^{-c_1(s-r)} \left( 1 - \frac{v_r}{v_s} \right) dr ds \right) .$$

Since $\eta$ is bounded, $\int_{t_0}^t \eta(s) e^{-c_1 s} ds \leq \mathcal{O}(1)$. Therefore it suffices to show

$$\int_{t_0}^t \eta(s) \int_0^s c_1 e^{-c_1(s-r)} \left( 1 - \frac{v_r}{v_s} \right) dr ds \leq \mathcal{O}(1) \tag{22}$$

Indeed, it holds for any positive $x, y$ that $1 - \frac{x}{y} \leq \log \frac{y}{x}$. Applying this and a crucial change of summation order $\{t_0 \leq s \leq t, 0 \leq r \leq s\} \mapsto \{t_0 \leq r \leq t, r \leq s \leq t\}$,

$$\text{LHS of (22)} \leq \int_{t_0}^t \eta(s) \int_0^s c_1 e^{-c_1(s-r)} \left( 1 - \frac{v_r}{v_s} \right) dr ds \leq \int_{t_0}^t \eta(s) \int_0^s c_1 e^{-c_1(s-r)} \left( \log v_s - \log v_r \right) dr ds$$

$$= \int_{t_0}^t \eta(s) \log v_s (1 - e^{-c_1 s}) ds - \int_{t_0}^t \eta(s) \int_0^s c_1 e^{-c_1(s-r)} \log v_r dr ds$$

$$\overset{\text{sum order}}{=} \int_{t_0}^t \eta(s) \log v_s (1 - e^{-c_1 s}) ds - \int_{t_0}^t \log v_r \int_r^t \eta(s) c_1 e^{-c_1(s-r)} ds dr$$

$$\overset{r \leftrightarrow s}{=} \int_{t_0}^t \eta(s) \log v_s (1 - e^{-c_1 s}) ds - \int_{t_0}^t \log v_s \int_s^t \eta(r) c_1 e^{-c_1(r-s)} dr ds$$

$$= \int_{t_0}^t \log v_s \left( \eta(s)(1 - e^{-c_1 s}) - \int_s^t \eta(r) c_1 e^{-c_1(r-s)} dr \right) ds .$$

Since $\eta$ is non-increasing,

$$\eta(s)(1 - e^{-c_1 s}) - \int_s^t \eta(r) c_1 e^{-c_1(r-s)} dr \geq \eta(s)(1 - e^{-c_1 s}) - \eta(s) \int_s^t c_1 e^{-c_1(r-s)} dr$$

$$= \eta(s)(1 - e^{-c_1 s}) - \eta(s)(1 - e^{-c_1(t-s)})$$

$$= \eta(s) \left( e^{-c_1(t-s)} - e^{-c_1 s} \right)$$

$$\geq -\eta(s) e^{-c_1 s} .$$

Since eventually $g_t^2 < 1 - \delta$, there exists $t_1$ with $\forall t \geq t_1 : v_t \leq 1$ (Corollary B.6), hence $\log v_t \leq 0$. So,

$$\int_{t_1}^t \log v_s \left( \eta(s)(1 - e^{-c_1 s}) - \int_s^t \eta(r) c_1 e^{-c_1(r-s)} dr \right) ds \leq \int_{t_1}^t -\log v_s \eta(s) e^{-c_1 s} ds .$$

Since $g_t \neq 0$ on an initial interval, $v_t \geq e^{-c_2 t} \int_0^t c_2 e^{c_2 s} g_s^2 ds \geq \Omega(e^{-c_2 t})$, so there exists $C > 0$ with $\log v_t \geq -c_2 t - C$, therefore,

$$\int_{t_1}^t -\log v_s \eta(s) e^{-c_1 s} ds \leq \int_{t_1}^\infty (c_2 s + C) \eta(s) e^{-c_1 s} ds < \infty .$$

And since on $[t_0, t_1]$ the integral is also finite (the integrand is bounded), we are finished showing (22) and therefore finished altogether. $\qquad\square$

## C. Proof Details

### C.1. Losses

In this work we consider *log-concave, exponentially-tailed losses*. Namely, we consider a loss of the form

$$\mathcal{L}(\boldsymbol{\theta}) = \sum_{i=1}^{m} \ell(y_i f(\mathbf{x}_i; \boldsymbol{\theta})) ,$$

where $\ell(u) = e^{-\varphi(u)}$, and $\varphi \in C^2(\mathbb{R})$ with $\exists \Phi'_M, \Phi''_M > 0 : \forall u \in \mathbb{R} : 0 < \varphi'(u) \leq \Phi'_M, 0 \leq \varphi''(u) \leq \Phi''_M$.

Note that $\ell \in (\ell_{\exp}, \ell_{\log})$ satisfy this:

$$\ell_{\exp}(u) = e^{-u} = e^{-\varphi_{\exp}(u)}, \quad \ell_{\log}(u) = \log(1 + e^{-u}) = e^{-\varphi_{\log}(u)} ,$$

for

$$\varphi_{\exp}(u) = u, \quad \varphi_{\log}(u) = -\log\log(1 + e^{-u}) .$$

The conditions on $\varphi$ imply the following:

1. $\varphi$ is strictly monotone increasing and $\ell$ is strictly monotone decreasing. Also $u \mapsto \varphi'(u)u$ is strictly monotone increasing on $[0, \infty)$:

$$\frac{d\varphi'(u)u}{du} = \varphi''(u)u + \varphi'(u) \geq \varphi'(u) > 0 .$$

2. For any $u_0 \in \mathbb{R}$, $\varphi$ has at least a linear growth rate on $[u_0, \infty)$:

$$\varphi(u) - \varphi(u_0) = \int_{u_0}^{u} \varphi'(w)dw \geq \varphi'(u_0)(u - u_0) . \tag{23}$$

   In particular $\varphi(u) \overset{u \to \infty}{\longrightarrow} \infty$.

3. $\varphi$ has a strictly monotone increasing inverse $\varphi^{-1}$ defined on $\varphi(\mathbb{R})$. In particular $\varphi^{-1}$ is defined on $[\varphi(0), \infty)$, $\varphi^{-1}(u) \overset{u \to \infty}{\longrightarrow} \infty$, and $(\varphi^{-1})'$ is *nonincreasing* since $(\varphi^{-1})'(u) = \frac{1}{\varphi'(\varphi^{-1}(u))}$ and $\varphi^{-1}, \varphi'$ are increasing. This implies also that $(\varphi^{-1})'$ is bounded on $[\varphi(0), \infty)$.

4. For any $a \in \mathbb{R} \cup \{\pm\infty\}$, $\lim_{u \to a} \ell(u) = 0$ implies $\lim_{u \to a} \varphi(u) = \infty$, implying $a = \infty$, since $\varphi$ is bounded on any compact interval and monotone increasing. Note that since the same is true of $\varphi^{-1}$, this implies also that if $\lim_{u \to a} \varphi^{-1}(u) = \infty$ then $a = \infty$.

### C.2. Algorithm-Independent KKT Stationarity

In this section we provide general insight into properties of trajectories of homogeneous models, culminating in Theorem C.8 which shows that KKT stationarity of a limit point of $\frac{\boldsymbol{\theta}_t}{\|\boldsymbol{\theta}_t\|}$ with respect to Problem (11) follows from decay of the loss and alignment between gradients and parameters.

**Definition C.1** (Hard and Soft Margins). Let $f(\mathbf{x}; \boldsymbol{\theta})$ be a model satisfying (M1-Weak), (M2). We denote the "hard" and "soft" margins $\gamma(\boldsymbol{\theta}), \widetilde{\gamma}(\boldsymbol{\theta})$ for $\boldsymbol{\theta} \neq 0$ as follows:

$$\gamma(\boldsymbol{\theta}) := \min_{i \in [m]} y_i f\left(x_i; \frac{\boldsymbol{\theta}}{\|\boldsymbol{\theta}\|}\right), \quad \widetilde{\gamma}(\boldsymbol{\theta}) := \frac{\varphi^{-1}\left(\log \frac{1}{\mathcal{L}(\boldsymbol{\theta})}\right)}{\|\boldsymbol{\theta}\|^L} .$$

Note that $\widetilde{\gamma}$ is well defined whenever $\log \frac{1}{\mathcal{L}(\boldsymbol{\theta})} \in \varphi(\mathbb{R})$, and in particular whenever $\log \frac{1}{\mathcal{L}(\boldsymbol{\theta})} > \varphi(0)$.

Recall the notations from Subsection 2.1; in particular $q_{\min}^t = \min_{i \in [m]} z_i^t = \min_{i \in [m]} y_i f(\mathbf{x}_i; \boldsymbol{\theta}_t) = \gamma(\boldsymbol{\theta}_t) \|\boldsymbol{\theta}_t\|^L$.

**Lemma C.2** (Properties of Any Trajectory ). *Let $\boldsymbol{\theta}_t$ be any arc of parameters of a model $f(\mathbf{x}; \boldsymbol{\theta})$ assuming (M1-Weak), (M2). Then the following hold for any norm $\|\cdot\|$:*

1. *For any $t \geq 0$:*
$$\ell\left(q_{\min}^t\right) \leq \mathcal{L}(\boldsymbol{\theta}_t) \leq m \cdot \ell\left(q_{\min}^t\right) .$$

2. *For any $t \geq 0$ with $\boldsymbol{\theta}_t \neq \mathbf{0}$ and $\log \frac{1}{\mathcal{L}(\boldsymbol{\theta}_t)} > \varphi(0)$:*
$$\widetilde{\gamma}(\boldsymbol{\theta}_t) \leq \gamma(\boldsymbol{\theta}_t) .$$

   *For any $t \geq 0$ with $\boldsymbol{\theta}_t \neq \mathbf{0}$ and $\varphi(q_{\min}^t) - \log m > \varphi(0)$:*
$$\gamma(\boldsymbol{\theta}_t) - \frac{(\varphi^{-1})'(\varphi(q_{\min}^t) - \log m) \cdot \log m}{\|\boldsymbol{\theta}_t\|^L} \leq \widetilde{\gamma}(\boldsymbol{\theta}_t) ,$$

   *Finally, if $\mathcal{L}(\boldsymbol{\theta}_t) \overset{t \to \infty}{\longrightarrow} 0$, then $\|\boldsymbol{\theta}_t\| \overset{t \to \infty}{\longrightarrow} \infty$ and $|\widetilde{\gamma}(\boldsymbol{\theta}_t) - \gamma(\boldsymbol{\theta}_t)| \overset{t \to \infty}{\longrightarrow} 0$.*

3. *For almost any $t \geq 0$ with $\boldsymbol{\theta}_t \neq \mathbf{0}$:*
$$\left\| \frac{d \frac{\boldsymbol{\theta}_t}{\|\boldsymbol{\theta}_t\|}}{dt} \right\| \leq \frac{2 \left\| \frac{d\boldsymbol{\theta}_t}{dt} \right\|}{\|\boldsymbol{\theta}_t\|} . \tag{24}$$

4. *There exists $M > 0$ so that for any $t \geq 0$ with $\boldsymbol{\theta}_t \neq \mathbf{0}$,*
$$\|\mathbf{g}_t\|_\star \leq M \|\boldsymbol{\theta}_t\|^{L-1} \mathcal{L}(\boldsymbol{\theta}_t) . \tag{25}$$

   *Also, for any $t \geq 0$,*
$$L\mathcal{L}(\boldsymbol{\theta}_t) \cdot \min_{i \in [m]} \varphi'(z_i^t) z_i^t \leq \langle -\mathbf{g}_t, \boldsymbol{\theta}_t \rangle \leq \|\mathbf{g}_t\|_\star \|\boldsymbol{\theta}_t\| .$$

   *If $\log \frac{1}{\mathcal{L}(\boldsymbol{\theta}_t)} > \varphi(0)$ then*
$$L\mathcal{L}(\boldsymbol{\theta}_t) \frac{\varphi^{-1}(\log \frac{1}{\mathcal{L}(\boldsymbol{\theta}_t)})}{(\varphi^{-1})'(\log \frac{1}{\mathcal{L}(\boldsymbol{\theta}_t)})} \leq L\mathcal{L}(\boldsymbol{\theta}_t) \cdot \min_{i \in [m]} \varphi'(z_i^t) z_i^t \leq \langle -\mathbf{g}_t, \boldsymbol{\theta}_t \rangle . \tag{26}$$

   *If $q_{\min}^t > q > 0$ and $\boldsymbol{\theta}_t \neq \mathbf{0}$ then*
$$\varphi'(q) L\mathcal{L}(\boldsymbol{\theta}_t) \gamma(\boldsymbol{\theta}_t) \|\boldsymbol{\theta}_t\|^L \leq L\mathcal{L}(\boldsymbol{\theta}_t) \cdot \min_{i \in [m]} \varphi'(z_i^t) z_i^t \leq \langle -\mathbf{g}_t, \boldsymbol{\theta}_t \rangle . \tag{27}$$

5. *If for all large enough $t$, $\|\boldsymbol{\theta}_t\| \geq N_{min} > 0$ and $\gamma(\boldsymbol{\theta}_t) > \gamma_{min} > 0$ then*
$$\|\mathbf{g}_t\|_\star = \Theta(\|\boldsymbol{\theta}_t\|^{L-1} \mathcal{L}(\boldsymbol{\theta}_t)) = \Theta(\|\boldsymbol{\theta}_t\|^{L-1} \ell\left(q_{\min}^t\right)) .$$

   *And in particular $\|\mathbf{g}_t\|_\star$ is bounded.*

*Proof.* 1. This follows directly from the definition of the loss,
$$\mathcal{L}(\boldsymbol{\theta}_t) = \sum_{i=1}^m \ell\left(y_i f(\mathbf{x}_i; \boldsymbol{\theta}_t)\right) ,$$

noting that $q_{\min}^t = \min_{i \in [m]} y_i f(\mathbf{x}_i; \boldsymbol{\theta}_t)$ by definition, and $\ell$ is monotone decreasing, giving
$$\ell\left(q_{\min}^t\right) \leq \mathcal{L}(\boldsymbol{\theta}_t) \leq m \cdot \ell\left(q_{\min}^t\right)$$

2. From item 1, when $\boldsymbol{\theta}_t \neq \mathbf{0}$,
$$\frac{1}{m} e^{\varphi(q_{\min}^t)} \leq \frac{1}{\mathcal{L}(\boldsymbol{\theta}_t)} \leq e^{\varphi(q_{\min}^t)} ,$$
$$-\log m + \varphi\left(q_{\min}^t\right) \leq \log \frac{1}{\mathcal{L}(\boldsymbol{\theta}_t)} \leq \varphi\left(q_{\min}^t\right) . \tag{28}$$

Whenever $\log \frac{1}{\mathcal{L}(\boldsymbol{\theta}_t)} > \varphi(0)$, we may apply $\varphi^{-1}$ to the right inequality (recall $\varphi^{-1}$ is increasing) and divide by $\|\boldsymbol{\theta}_t\|^L$, getting
$$\widetilde{\gamma}(\boldsymbol{\theta}_t) \leq \gamma(\boldsymbol{\theta}_t) .$$

Whenever $-\log m + \varphi(q_{\min}^t) \in \varphi(\mathbb{R})$, and in particular whenever $-\log m + \varphi(q_{\min}^t) > \varphi(0)$, we may also apply $\varphi^{-1}$ to the left inequality of Equation (28):

$$\frac{\varphi^{-1}(\varphi(q_{\min}^t) - \log m)}{\|\boldsymbol{\theta}_t\|^L} \leq \widetilde{\gamma}(\boldsymbol{\theta}_t) \, .$$

Applying the mean value theorem, for some $u \in [\varphi(q_{\min}^t) - \log m, \varphi(q_{\min}^t)]$ it holds that

$$q_{\min}^t - \varphi^{-1}\left(\varphi\left(q_{\min}^t\right) - \log m\right) = \varphi^{-1}(\varphi(q_{\min}^t)) - \varphi^{-1}\left(\varphi\left(q_{\min}^t\right) - \log m\right) = (\varphi^{-1})'(u) \cdot (\log m) \, .$$

And since $(\varphi^{-1})'$ is monotone decreasing,

$$q_{\min}^t - \varphi^{-1}\left(\varphi\left(q_{\min}^t\right) - \log m\right) \leq (\varphi^{-1})'(\varphi(q_{\min}^t) - \log m) \cdot \log m \, .$$

Altogether

$$\gamma(\boldsymbol{\theta}_t) - \frac{(\varphi^{-1})'(\varphi(q_{\min}^t) - \log m) \cdot \log m}{\|\boldsymbol{\theta}_t\|^L} \leq \widetilde{\gamma}(\boldsymbol{\theta}_t) \, . \tag{29}$$

Now assume $\mathcal{L}(\boldsymbol{\theta}_t) \xrightarrow{t \to \infty} 0$. In particular $\ell(q_{\min}^t) = \ell(\gamma(\boldsymbol{\theta}_t)\|\boldsymbol{\theta}_t\|^L) \xrightarrow{t \to \infty} 0$, implying $\varphi(q_{\min}^t) = \varphi(\gamma(\boldsymbol{\theta}_t)\|\boldsymbol{\theta}_t\|^L) \xrightarrow{t \to \infty} \infty$ and therefore $\gamma(\boldsymbol{\theta}_t)\|\boldsymbol{\theta}_t\|^L \xrightarrow{t \to \infty} \infty$. In particular this means $\gamma(\boldsymbol{\theta}_t) > 0$ for all large enough $t$, and since $\gamma(\boldsymbol{\theta}_t)$ is bounded, also $\|\boldsymbol{\theta}_t\| \xrightarrow{t \to \infty} \infty$. Since $\varphi(q_{\min}^t) - \log m \xrightarrow{t \to \infty} \infty$ we have Equation (29) for all large enough $t$. Finally, since $\varphi(q_{\min}^t) \xrightarrow{t \to \infty} \infty$, $\|\boldsymbol{\theta}_t\| \xrightarrow{t \to \infty} \infty$ and $(\varphi^{-1})'$ is bounded on $[\varphi(0), \infty)$ it holds that

$$|\gamma(\boldsymbol{\theta}_t) - \widetilde{\gamma}(\boldsymbol{\theta}_t)| \xrightarrow{t \to \infty} 0 \, .$$

3. Let $\mathbf{n}_t \in \partial \|\boldsymbol{\theta}_t\|$, so by the chain rule for arcs (Theorem A.6), and using $\|\mathbf{n}_t\|_\star \leq 1$ from Equation (18):

$$\left|\frac{d\|\boldsymbol{\theta}_t\|}{dt}\right| \leq \left|\left\langle \mathbf{n}_t, \frac{d\boldsymbol{\theta}_t}{dt} \right\rangle\right| \leq \left\|\frac{d\boldsymbol{\theta}_t}{dt}\right\| \|\mathbf{n}_t\|_\star \leq \left\|\frac{d\boldsymbol{\theta}_t}{dt}\right\| \, .$$

Therefore

$$\left\|\frac{d\frac{\boldsymbol{\theta}_t}{\|\boldsymbol{\theta}_t\|}}{dt}\right\| = \left\|\frac{1}{\|\boldsymbol{\theta}_t\|}\frac{d\boldsymbol{\theta}_t}{dt} + \boldsymbol{\theta}_t\left(-\frac{1}{\|\boldsymbol{\theta}_t\|^2}\frac{d\|\boldsymbol{\theta}_t\|}{dt}\right)\right\| \leq \frac{\left\|\frac{d\boldsymbol{\theta}_t}{dt}\right\|}{\|\boldsymbol{\theta}_t\|} + \frac{1}{\|\boldsymbol{\theta}_t\|}\left|\frac{d\|\boldsymbol{\theta}_t\|}{dt}\right| \leq 2 \cdot \frac{\left\|\frac{d\boldsymbol{\theta}_t}{dt}\right\|}{\|\boldsymbol{\theta}_t\|} \, .$$

4. By the chain rule (Corollary A.2), let $\mathbf{h}_i^t \in \partial f(\mathbf{x}_i; \boldsymbol{\theta}_t)$ with

$$\mathbf{g}_t = -\sum_{i=1}^{m} \ell(z_i^t)\varphi'(z_i^t)y_i\mathbf{h}_i^t \, .$$

For the upper bound, using Theorem B.2(a) of (Lyu and Li, 2019),

$$\mathbf{g}_t = -\sum_{i=1}^{m} \ell\left(z_i^t\right)\varphi'(z_i^t)y_i\mathbf{h}_i^t = -\|\boldsymbol{\theta}_t\|^{L-1}\sum_{i=1}^{m} \ell\left(z_i^t\right)\varphi'(z_i^t)y_i\bar{\mathbf{h}}_i^t \, ,$$

where $\bar{\mathbf{h}}_i^t \in \partial f(\mathbf{x}_i; \frac{\boldsymbol{\theta}_t}{\|\boldsymbol{\theta}_t\|})$. On the unit sphere, which is a compact set, $f$ is Lipschitz, hence $\exists M' > 0 : \|\bar{\mathbf{h}}_i^t\|_\star \leq M'$ for any $t$. So, using $|\varphi'(z_i^t)| \leq \Phi'_M$,

$$\|\mathbf{g}_t\|_\star \leq M'\Phi'_M \|\boldsymbol{\theta}_t\|^{L-1} \mathcal{L}(\boldsymbol{\theta}_t) =: M \|\boldsymbol{\theta}_t\|^{L-1} \mathcal{L}(\boldsymbol{\theta}_t) \, .$$

For the lower bounds,

$$\|\boldsymbol{\theta}_t\| \|\mathbf{g}_t\|_\star \geq \langle \boldsymbol{\theta}_t, -\mathbf{g}_t \rangle = \left\langle \boldsymbol{\theta}_t, \sum_{i=1}^{m} \ell\left(z_i^t\right)\varphi'(z_i^t)y_i\mathbf{h}_i^t \right\rangle$$

$$= \sum_{i=1}^{m} \ell\left(z_i^t\right)\varphi'(z_i^t)y_i \langle \boldsymbol{\theta}_t, \mathbf{h}_i^t \rangle$$

$$= L \sum_{i=1}^{m} \ell\left(z_i^t\right)\varphi'(z_i^t)z_i^t \, .$$

Where the last equality uses Theorem B.2 from (Lyu and Li, 2019) (Euler's Theorem for homogeneous functions). Continuing, we use that fact that $u \mapsto \varphi'(u)u$ is increasing on $[0, \infty)$, that $\log \frac{1}{\mathcal{L}} > \varphi(0)$, that $z_i^t \geq q_{\min}^t \geq \varphi^{-1}(\log \frac{1}{\mathcal{L}})$ by previous items, and finally that $\varphi'(\varphi^{-1}(u)) = \frac{1}{(\varphi^{-1})'(u)}$. Altogether, this gives $\varphi'(z_i^t)z_i^t \geq \varphi'(\varphi^{-1}(\log \frac{1}{\mathcal{L}})) \cdot \varphi^{-1}(\log \frac{1}{\mathcal{L}}) = \frac{\varphi^{-1}(\log \frac{1}{\mathcal{L}})}{(\varphi^{-1})'(\log \frac{1}{\mathcal{L}})}$, so we continue with

$$\geq L\mathcal{L} \min_{i \in [m]} \varphi'(z_i^t)z_i^t$$

$$\geq L\mathcal{L} \frac{\varphi^{-1}(\log \frac{1}{\mathcal{L}})}{(\varphi^{-1})'(\log \frac{1}{\mathcal{L}})} \ .$$

Also note that if $q_{\min}(\boldsymbol{\theta}_t) > q > 0$ then $\forall i \in [m] : \varphi'(z_i^t) \geq \varphi'(q) > 0$, and recall $z_i^t \geq q_{\min}^t = \gamma(\boldsymbol{\theta}_t)\|\boldsymbol{\theta}_t\|^L$, so

$$\varphi'(q)L\mathcal{L}(\boldsymbol{\theta}_t)\gamma(\boldsymbol{\theta}_t)\|\boldsymbol{\theta}_t\|^L \leq L\mathcal{L} \min_{i \in [m]} \varphi'(z_i^t)z_i^t \leq \langle -\mathbf{g}_t, \boldsymbol{\theta}_t \rangle \ ,$$

finishing the lower bounds.

5. Note that $q_{\min}^t \geq N_{min}^L \gamma_{min} =: q > 0$, so we use Equations (25), (27) to conclude

$$\varphi'(q)\gamma_{min}L\mathcal{L}(\boldsymbol{\theta}_t)\|\boldsymbol{\theta}_t\|^{L-1} \leq \|\mathbf{g}_t\|_\star \leq M\|\boldsymbol{\theta}_t\|^{L-1}\mathcal{L}(\boldsymbol{\theta}_t) \ .$$

We claim the RHS is bounded. Since it is strictly positive, so we show that its logarithm is bounded from above. Indeed, $\mathcal{L}(\boldsymbol{\theta}_t) \leq me^{-\varphi(q_{\min}(\boldsymbol{\theta}_t))} \leq me^{-\varphi(\gamma_{min}\|\boldsymbol{\theta}_t\|^L)}$, so

$$\log \|\boldsymbol{\theta}_t\|^{L-1}\mathcal{L}(\boldsymbol{\theta}_t) \leq (L-1)\log \|\boldsymbol{\theta}_t\| + \log m - \varphi(\gamma_{min}\|\boldsymbol{\theta}_t\|^L) \ .$$

On any compact interval $\|\boldsymbol{\theta}_t\| \in [N_{min}, N]$ this expression is bounded from above by continuity, and by Equation (23), choosing $0 < u < q, \varphi(\gamma_{min}\|\boldsymbol{\theta}_t\|^L) \geq \varphi(u) + \varphi'(u)(\gamma_{min}\|\boldsymbol{\theta}_t\|^L - u)$, so the expression tends to $-\infty$ as $\|\boldsymbol{\theta}_t\| \to \infty$.

$\square$

In this work we prove KKT stationarity using the notion of *approximate* KKT points.

**Definition C.3.** We say $\boldsymbol{\theta}$ is a feasible point of Problem (11) if $\forall i \in [m] : y_i f(\mathbf{x}_i; \boldsymbol{\theta}) \geq 1$, or equivalently if $q_{\min}(\boldsymbol{\theta}) \geq 1$.

**Definition C.4.** A feasible point $\boldsymbol{\theta}$ of Problem (11) is called an $(\varepsilon, \delta)$-approximate KKT point if there exist $\lambda_1, ..., \lambda_m \geq 0$, $\mathbf{h}_i \in \partial f(\mathbf{x}_i; \boldsymbol{\theta})$ and $\mathbf{k} \in \partial \frac{1}{2}\|\boldsymbol{\theta}\|^2$ with

1. $\|\sum_{i=1}^m \lambda_i y_i \mathbf{h}_i - \mathbf{k}\|_2 \leq \varepsilon$.

2. $\sum_{i=1}^m \lambda_i(y_i f(\mathbf{x}_i; \boldsymbol{\theta}) - 1) \leq \delta$.

**Definition C.5.** We say that a feasible point of Problem (11) satisfies the *Mangasarian-Fromovitz Constraint Qualifications* if there exists $\mathbf{v} \in \mathbb{R}^p$ such that for all $i \in [m]$ with $1 - y_i f(\mathbf{x}_i; \boldsymbol{\theta}) = 0$ and for all $\mathbf{h} \in \partial(1 - y_i f(\mathbf{x}_i; \boldsymbol{\theta}))$, it holds:

$$\langle \mathbf{v}, \mathbf{h} \rangle > 0. \tag{30}$$

**Lemma C.6** (Lemma A.11 in Tsilivis et al. (2025)). *The MFCQ constraints are satisfied by any feasible point $\boldsymbol{\theta}$ of Problem* (11) *for $f$ satisfying, (M1-Weak), (M2).*

**Theorem C.7** (Theorem C.4 in Lyu and Li (2019) + Lemma C.6). *Let $\boldsymbol{\theta}_n \to \boldsymbol{\theta}$ be a converging sequence so that $\boldsymbol{\theta}_n$ is a $(\varepsilon_n, \delta_n)$-approximate KKT point of Problem* (11)*, where $\varepsilon_n \to 0, \delta_n \to 0$. Then $\boldsymbol{\theta}$ is a KKT point of Problem* (11)*.*

**Theorem C.8** (KKT Stationarity Derived from Alignment). *Let $\boldsymbol{\theta}_t$ be any arc of parameters of a model $f(\mathbf{x}; \boldsymbol{\theta})$ assuming (M1-Weak), (M2), and denote $\mathbf{g}_t \in \partial \mathcal{L}(\boldsymbol{\theta}_t)$ any choice of subgradient along the trajectory. Let $\bar{\boldsymbol{\theta}}$ with $\|\bar{\boldsymbol{\theta}}\| = 1, \gamma(\bar{\boldsymbol{\theta}}) > 0$. Assume that there exists a sequence $t_n$ with $\boldsymbol{\theta}_{t_n} \neq \mathbf{0}, \mathbf{g}_{t_n} \neq \mathbf{0}$ and:*

1. $\mathcal{L}(\boldsymbol{\theta}_{t_n}) \xrightarrow{n \to \infty} 0$

2. $\frac{\boldsymbol{\theta}_{t_n}}{\|\boldsymbol{\theta}_{t_n}\|} \xrightarrow{n \to \infty} \bar{\boldsymbol{\theta}}$

3. $\left\langle \frac{\boldsymbol{\theta}_{t_n}}{\|\boldsymbol{\theta}_{t_n}\|}, -\frac{\mathbf{g}_{t_n}}{\|\mathbf{g}_{t_n}\|_\star} \right\rangle \xrightarrow{n \to \infty} 1.$

*Then $\bar{\boldsymbol{\theta}}$ is the direction of a KKT point of Equation* (11).

*Proof.* First, note that since $\mathcal{L}(\boldsymbol{\theta}_{t_n}) \overset{n\to\infty}{\longrightarrow} 0$, it holds that $\|\boldsymbol{\theta}_{t_n}\| \overset{n\to\infty}{\longrightarrow} \infty, q_{\min}^{t_n} \overset{n\to\infty}{\longrightarrow} \infty$. Since $\gamma(\boldsymbol{\theta}_{t_n})$ converges to $\gamma(\bar{\boldsymbol{\theta}}) > 0$, in particular for large enough $n$, $\gamma(\boldsymbol{\theta}_{t_n}) > \frac{1}{2}\gamma(\bar{\boldsymbol{\theta}}) > 0$. Therefore it holds by Lemma C.2 that for large enough $n$, there exists $C > 0$ with

$$\|\mathbf{g}_{t_n}\|_{\star} \geq C\mathcal{L}(\boldsymbol{\theta}_{t_n}) \|\boldsymbol{\theta}_{t_n}\|^{L-1} \,.$$

Now, any $\boldsymbol{\theta}$ with $\gamma(\boldsymbol{\theta}) > 0$ has for all $i \in [m]$ that $y_i f(\mathbf{x}_i; \boldsymbol{\theta}) > 0$ and therefore there exists a feasible point $\widetilde{\boldsymbol{\theta}}$ of Equation (11) along its direction with $q_{\min}(\widetilde{\boldsymbol{\theta}}) = 1$, given by

$$\widetilde{\boldsymbol{\theta}} := \frac{\boldsymbol{\theta}}{q_{\min}(\boldsymbol{\theta})^{1/L}} = \frac{\boldsymbol{\theta}}{\min_{i\in[m]}\left(y_i f(\mathbf{x}_i; \boldsymbol{\theta})\right)^{1/L}} = \frac{\boldsymbol{\theta}}{\gamma(\boldsymbol{\theta})^{1/L}\|\boldsymbol{\theta}\|} \,.$$

Denote $\hat{\boldsymbol{\theta}} := \widetilde{\bar{\boldsymbol{\theta}}} = \frac{\bar{\boldsymbol{\theta}}}{\gamma(\bar{\boldsymbol{\theta}})^{1/L}}$. We claim $\hat{\boldsymbol{\theta}}$ is the desired KKT point. We show that in fact $\hat{\boldsymbol{\theta}}$ is itself a $(\varepsilon_s, \delta_s)$-approximate KKT point for arbitrarily small $(\varepsilon_s, \delta_s)$, allowing us to use Theorem C.7 with the constant sequence $\hat{\boldsymbol{\theta}}$.

First note that by continuity and scale-invariance of $\gamma(\boldsymbol{\theta})$, it holds that

$$\widetilde{\boldsymbol{\theta}}_{t_n} = \frac{1}{\gamma(\boldsymbol{\theta}_{t_n})^{1/L}} \frac{\boldsymbol{\theta}_{t_n}}{\|\boldsymbol{\theta}_{t_n}\|} \overset{n\to\infty}{\longrightarrow} \frac{1}{\gamma(\bar{\boldsymbol{\theta}})^{1/L}}\bar{\boldsymbol{\theta}} = \hat{\boldsymbol{\theta}} \,.$$

Take subgradients $\mathbf{h}_i^t \in \partial f(\mathbf{x}_i, \boldsymbol{\theta}_t)$ defining $\mathbf{g}_t$ by the chain rule as in Lemma C.2. Since $\left(q_{\min}(\boldsymbol{\theta}_t)^{-\frac{1}{L}}\right)^{L-1} = q_{\min}(\boldsymbol{\theta}_t)^{\frac{1}{L}-1}$, it holds by Theorem B.2(a) in Lyu and Li (2019) that $\widetilde{\mathbf{h}}_i^t := q_{\min}(\boldsymbol{\theta}_t)^{\frac{1}{L}-1}\mathbf{h}_i^t \in \partial f(\mathbf{x}_i; \widetilde{\boldsymbol{\theta}}_t)$. Since $f$ is locally Lipschitz, it holds that $\partial f(\mathbf{x}_i; \cdot)$ is bounded and has a closed graph around $\hat{\boldsymbol{\theta}}$, so for every $i \in [m]$ there exists a subsequence $t_s$ of $t_n$ with $\widetilde{\mathbf{h}}_i^{(t_s)} \to \hat{\mathbf{h}}_i$ for some $\hat{\mathbf{h}}_i \in \partial f(\mathbf{x}_i; \hat{\boldsymbol{\theta}})$. By iteratively choosing subsequences for $i = 1, ..., m$ we may assume that $\widetilde{\mathbf{h}}_i^{(t_s)} \to \hat{\mathbf{h}}_i$ for all $i \in [m]$. Take a Bolzano-Weierstrass limit $\mathbf{u}^{\star}$ of $-\frac{\mathbf{g}_{t_s}}{\|\mathbf{g}_{t_s}\|_{\star}}$, and again abuse notation by assuming w.l.o.g $-\frac{\mathbf{g}_{t_s}}{\|\mathbf{g}_{t_s}\|_{\star}} \overset{s\to\infty}{\longrightarrow} \mathbf{u}^{\star}$. It therefore holds that $\|\mathbf{u}^{\star}\|_{\star} = 1$ and

$$\langle\bar{\boldsymbol{\theta}}, \mathbf{u}^{\star}\rangle = \lim_{s\to\infty}\left\langle\bar{\boldsymbol{\theta}}, -\frac{\mathbf{g}_{t_s}}{\|\mathbf{g}_{t_s}\|_{\star}}\right\rangle = 1 \,.$$

Denote

$$\mathbf{k} := \left\|\hat{\boldsymbol{\theta}}\right\|\mathbf{u}^{\star}, \quad \forall i \in [m], s \in \mathbb{N} : \lambda_{s,i} := \frac{\left\|\hat{\boldsymbol{\theta}}\right\| q_{\min}(\boldsymbol{\theta}_{t_s})^{1-\frac{1}{L}}\ell\left(z_i^{t_s}\right)\varphi'(z_i^{t_s})}{\|\mathbf{g}_{t_s}\|_{\star}} \geq 0 \,.$$

Note that $\mathbf{k} \in \partial\frac{1}{2}\left\|\hat{\boldsymbol{\theta}}\right\|^2$ since by the chain rule $\partial\frac{1}{2}\left\|\hat{\boldsymbol{\theta}}\right\|^2 = \left\|\hat{\boldsymbol{\theta}}\right\| \cdot \partial\left\|\hat{\boldsymbol{\theta}}\right\| = \left\{\left\|\hat{\boldsymbol{\theta}}\right\|\mathbf{v} \mid \langle\mathbf{v}, \hat{\boldsymbol{\theta}}\rangle = \left\|\hat{\boldsymbol{\theta}}\right\|, \|\mathbf{v}\|_{\star} \leq 1\right\}$ and indeed $\langle\mathbf{u}^{\star}, \hat{\boldsymbol{\theta}}\rangle = \langle\mathbf{u}^{\star}, \left\|\hat{\boldsymbol{\theta}}\right\|\bar{\boldsymbol{\theta}}\rangle = \left\|\hat{\boldsymbol{\theta}}\right\|$ and $\|\mathbf{u}^{\star}\|_{\star} = 1$. Choosing $\hat{\mathbf{h}}_i, i \in [m]$ as the subgradients from Definition C.4, it remains to show that the errors are bounded by $\varepsilon_s, \delta_s \to 0$. Beginning from $\varepsilon_s$, first note that

$$\sum_{i=1}^m \lambda_{s,i} y_i \widetilde{\mathbf{h}}_i^{(t_s)} = \sum_{i=1}^m \lambda_{s,i} q_{\min}^{\frac{1}{L}-1} y_i \mathbf{h}_i^{(t_s)} = \frac{\left\|\hat{\boldsymbol{\theta}}\right\|}{\|\mathbf{g}_{t_s}\|_{\star}}\sum_{i=1}^m \ell\left(z_i^{t_s}\right)\varphi'(z_i^{t_s})y_i\mathbf{h}_i^{(t_s)} = -\left\|\hat{\boldsymbol{\theta}}\right\|\frac{\mathbf{g}_{t_s}}{\|\mathbf{g}_{t_s}\|_{\star}} \,.$$

So,

$$\varepsilon_s := \left\|\sum_{i=1}^m \lambda_{s,i} y_i \hat{\mathbf{h}}_i - \left\|\hat{\boldsymbol{\theta}}\right\|\mathbf{u}^{\star}\right\|_2 \leq \left\|\sum_{i=1}^m \lambda_{s,i} y_i\left(\hat{\mathbf{h}}_i - \widetilde{\mathbf{h}}_i^{(t_s)}\right)\right\|_2 + \left\|\sum_{i=1}^m \lambda_{s,i} y_i \widetilde{\mathbf{h}}_i^{(t_s)} - \left\|\hat{\boldsymbol{\theta}}\right\|\mathbf{u}^{\star}\right\|_2 \,. \tag{31}$$

The second term goes to 0 since $-\frac{\mathbf{g}_{t_s}}{\|\mathbf{g}_{t_s}\|_{\star}} \to \mathbf{u}^{\star}$. To show the first term goes to 0, we use $\widetilde{\mathbf{h}}_i^{(t_s)} \to \hat{\mathbf{h}}_i$ and show $\lambda_{s,i}$ are

bounded. We use $\|\mathbf{g}_{t_s}\|_\star \geq C\mathcal{L}(\boldsymbol{\theta}_{t_s})\|\boldsymbol{\theta}_{t_s}\|^{L-1}$:

$$\lambda_{s,i} = \frac{\left\|\hat{\boldsymbol{\theta}}\right\| q_{\min}(\boldsymbol{\theta}_{t_s})^{1-\frac{1}{L}}\ell(z_i^t)\,\varphi'(z_i^t)}{\|\mathbf{g}_{t_s}\|_\star} \leq \frac{\Phi_M'\left\|\hat{\boldsymbol{\theta}}\right\|\gamma(\boldsymbol{\theta}_{t_s})^{1-\frac{1}{L}}\|\boldsymbol{\theta}_{t_s}\|^{L-1}\mathcal{L}}{\|\mathbf{g}_{t_s}\|_\star}$$

$$\leq \frac{\Phi_M'\left\|\hat{\boldsymbol{\theta}}\right\|\gamma(\boldsymbol{\theta}_{t_s})^{1-\frac{1}{L}}}{C} < \infty\,,$$

since $\gamma$ is bounded. For $\delta_s$,

$$\delta_s = \left|\sum_{i=1}^m \lambda_{s,i}(y_i f(\mathbf{x}_i;\hat{\boldsymbol{\theta}})-1)\right| \leq \left|\sum_{i=1}^m \lambda_{s,i}(y_i f(\mathbf{x}_i;\hat{\boldsymbol{\theta}}) - y_i f(\mathbf{x}_i;\widetilde{\boldsymbol{\theta}}_{t_s}))\right| + \left|\sum_{i=1}^m \lambda_{s,i}(y_i f(\mathbf{x}_i;\widetilde{\boldsymbol{\theta}}_{t_s})-1)\right| \quad (32)$$

The first term goes to 0 from continuity of $f$ and boundedness of $\lambda_{s,i}$. As for the second term, note that each summand is non-negative. Plugging in $\lambda_{s,i}$ (here $z_i = z_i^{t_s}$, and similarly for $q_{\min}, \mathcal{L}$),

$$\sum_{i=1}^m \lambda_{s,i}\big(y_i f(\mathbf{x}_i;\widetilde{\boldsymbol{\theta}}_{t_s})-1\big) = \frac{\left\|\hat{\boldsymbol{\theta}}\right\|}{\|\mathbf{g}_{t_s}\|_\star}\sum_{i=1}^m q_{\min}^{1-\frac{1}{L}}\ell(z_i)\,\varphi'(z_i)\left(\frac{z_i}{q_{\min}}-1\right)$$

$$= \frac{\left\|\hat{\boldsymbol{\theta}}\right\|}{q_{\min}^{1/L}\|\mathbf{g}_{t_s}\|_\star}\sum_{i=1}^m \varphi'(z_i)\ell(z_i)(z_i-q_{\min})$$

$$\leq \frac{\left\|\hat{\boldsymbol{\theta}}\right\|}{\|\boldsymbol{\theta}_{t_s}\|\gamma(\boldsymbol{\theta}_{t_s})^{1/L}C\mathcal{L}\|\boldsymbol{\theta}_{t_s}\|^{L-1}}\sum_{i=1}^m \varphi'(z_i)e^{-\varphi(z_i)}(z_i-q_{\min})$$

$$= \frac{\left\|\hat{\boldsymbol{\theta}}\right\|e^{-\varphi(q_{\min})}}{C\gamma(\boldsymbol{\theta}_{t_s})^{1/L}\mathcal{L}\|\boldsymbol{\theta}_{t_s}\|^L}\sum_{i=1}^m \varphi'(z_i)e^{-(\varphi(z_i)-\varphi(q_{\min}))}(z_i-q_{\min})$$

$$\leq \frac{\left\|\hat{\boldsymbol{\theta}}\right\|}{C\gamma(\boldsymbol{\theta}_{t_s})^{1/L}\|\boldsymbol{\theta}_{t_s}\|^L}\sum_{i=1}^m \varphi'(z_i)e^{-(\varphi(z_i)-\varphi(q_{\min}))}(z_i-q_{\min})$$

$$= \frac{\left\|\hat{\boldsymbol{\theta}}\right\|}{C\gamma(\boldsymbol{\theta}_{t_s})^{1/L}\|\boldsymbol{\theta}_{t_s}\|^L}\sum_{i:z_i>q_{\min}} \varphi'(z_i)e^{-(\varphi(z_i)-\varphi(q_{\min}))}(\varphi(z_i)-\varphi(q_{\min}))\frac{z_i-q_{\min}}{\varphi(z_i)-\varphi(q_{\min})}$$

$$\leq \frac{\left\|\hat{\boldsymbol{\theta}}\right\|}{C\gamma(\boldsymbol{\theta}_{t_s})^{1/L}\|\boldsymbol{\theta}_{t_s}\|^L}\sum_{i:z_i>q_{\min}} \frac{\varphi'(z_i)}{e}\frac{z_i-q_{\min}}{\varphi(z_i)-\varphi(q_{\min})}$$

$$\xrightarrow{s\to\infty} 0$$

The first inequality is again by $\|\mathbf{g}_t\|_\star \geq C\mathcal{L}\|\boldsymbol{\theta}_t\|^{L-1}$, and the second by $e^{-\varphi(q_{\min})} \leq \mathcal{L}$. In the last inequality we used the fact $\forall z \geq 0: ze^{-z} \leq 1/e$. The limit holds since $\|\boldsymbol{\theta}_{t_s}\| \xrightarrow{s\to\infty} \infty$, $\gamma(\boldsymbol{\theta}_{t_s}) \xrightarrow{s\to\infty} \gamma(\bar{\boldsymbol{\theta}}) > 0$, $\varphi'$ is bounded, and since for all large enough $s$, $q_{\min}^{t_s} > 1$, so by Equation (23):

$$\forall i \in [m], z_i^{t_s} > q_{\min}^{t_s}: \frac{z_i^{t_s}-q_{\min}^{t_s}}{\varphi(z_i^{t_s})-\varphi(q_{\min}^{t_s})} \leq \frac{z_i^{t_s}-q_{\min}^{t_s}}{\varphi'(1)(z_i^{t_s}-q_{\min}^{t_s})} = \frac{1}{\varphi'(1)} < \infty$$

$\square$

## C.3. Normalized Steepest Descent

In this section we follow the proof ideas of Tsilivis et al. (2025) to extend both main results: monotonicity of the soft margin and convergence to KKT points, to normalized steepest descent with a learning rate schedule $\eta(t)$. As mentioned in Section 3.1, the results may also be derived from a reparameterization argument; we put forth the proofs for completeness and for the rates in Lemma C.10. For convenience, we introduce notations of two additional assumptions used only in this section:

(LR-NSD) $\eta(t)$ satisfies $\int_0^\infty \eta(t)dt = \infty$.

(R1) $\exists t_0 \geq 0 : \mathcal{L}(\boldsymbol{\theta}_{t_0}) < 1$.

**Theorem C.9** (Monotonicity of the Soft Margin). *Let $\boldsymbol{\theta}_t$ follow a trajectory of Equation* (3), *under Assumptions (M1-Weak), (M2), (R1), (LR-NSD). Then $\widetilde{\gamma}(\boldsymbol{\theta}_t)$ is non-decreasing on $[t_0, \infty)$.*

*Proof.* By the chain rule, for almost any $t \geq t_0$ and for any $\mathbf{g}_t \in \partial \mathcal{L}(\boldsymbol{\theta}_t)$:

$$\frac{d\mathcal{L}(\boldsymbol{\theta}_t)}{dt} = \left\langle \mathbf{g}_t, \frac{d\boldsymbol{\theta}_t}{dt} \right\rangle .$$

In particular this holds for $\mathbf{g}_t \in \partial \mathcal{L}(\boldsymbol{\theta}_t)$ from the definition of normalized steepest descent for which

$$-\|\mathbf{g}_t\|_\star = \left\langle \frac{1}{\eta(t)} \frac{d\boldsymbol{\theta}_t}{dt}, \mathbf{g}_t \right\rangle = \frac{1}{\eta(t)} \frac{d\mathcal{L}(\boldsymbol{\theta}_t)}{dt} . \tag{33}$$

Notice that (R1) is equivalent to $\log \frac{1}{\mathcal{L}(\boldsymbol{\theta}_t)} > \varphi(0)$, or $\varphi^{-1}\left( \log \frac{1}{\mathcal{L}(\boldsymbol{\theta}_t)} \right) > 0$, so for $t = t_0$ we have $\widetilde{\gamma}(\boldsymbol{\theta}_t) > 0$, and $\log \widetilde{\gamma}$ is well defined. For any $t \geq t_0$ let $\mathbf{n}_t \in \partial \|\boldsymbol{\theta}_t\|$ (with $\|\mathbf{n}_t\|_\star \leq 1$) and any $\mathbf{g}_t \in \partial \mathcal{L}(\boldsymbol{\theta}_t)$. For almost any $t \geq t_0$ with $\widetilde{\gamma}(\boldsymbol{\theta}_t) > 0$ it holds that

$$\frac{d \log \widetilde{\gamma}(\boldsymbol{\theta}_t)}{dt} = \frac{d}{dt} \log \varphi^{-1}\left( \log \frac{1}{\mathcal{L}(\boldsymbol{\theta}_t)} \right) - L \frac{d}{dt} \log \|\boldsymbol{\theta}_t\|$$

$$= -\frac{d\mathcal{L}(\boldsymbol{\theta}_t)}{dt} \frac{(\varphi^{-1})'(\log \frac{1}{\mathcal{L}(\boldsymbol{\theta}_t)})}{\mathcal{L}(\boldsymbol{\theta}_t)\varphi^{-1}(\log \frac{1}{\mathcal{L}(\boldsymbol{\theta}_t)})} - L \left\langle \frac{\mathbf{n}_t}{\|\boldsymbol{\theta}_t\|}, \frac{d\boldsymbol{\theta}_t}{dt} \right\rangle \qquad \text{Chain Rule}$$

$$\geq -\frac{d\mathcal{L}(\boldsymbol{\theta}_t)}{dt} \frac{(\varphi^{-1})'(\log \frac{1}{\mathcal{L}(\boldsymbol{\theta}_t)})}{\mathcal{L}(\boldsymbol{\theta}_t)\varphi^{-1}(\log \frac{1}{\mathcal{L}(\boldsymbol{\theta}_t)})} - L \frac{\|\frac{d\boldsymbol{\theta}_t}{dt}\|}{\|\boldsymbol{\theta}_t\|} \qquad \text{Def. of dual norm}$$

$$= \|\mathbf{g}_t\|_\star \eta(t) \frac{(\varphi^{-1})'(\log \frac{1}{\mathcal{L}(\boldsymbol{\theta}_t)})}{\mathcal{L}(\boldsymbol{\theta}_t)\varphi^{-1}(\log \frac{1}{\mathcal{L}(\boldsymbol{\theta}_t)})} - \frac{L\eta(t)}{\|\boldsymbol{\theta}_t\|} \qquad \text{Equation (33)}$$

$$\geq \frac{L\eta(t)\|\mathbf{g}_t\|_\star}{\langle \boldsymbol{\theta}_t, -\mathbf{g}_t \rangle} - \frac{L\eta(t)}{\|\boldsymbol{\theta}_t\|} \qquad \text{Equation (26)}$$

$$\geq 0 . \qquad \text{Def. of Dual Norm}$$

This shows that $\log \widetilde{\gamma}(\boldsymbol{\theta}_t)$ is non-decreasing whenever $\widetilde{\gamma}(\boldsymbol{\theta}_t) > 0$. Since $\widetilde{\gamma}(\boldsymbol{\theta}_{t_0}) > 0$, we get that $\log \widetilde{\gamma}(\boldsymbol{\theta}_t)$ and hence $\widetilde{\gamma}(\boldsymbol{\theta}_t)$ are non-decreasing on $[t_0, \infty)$.

$\square$

**Lemma C.10** (Descent Lemma for NSD). *Let $\boldsymbol{\theta}_t$ follow a trajectory of Equation* (3) *under Assumptions (M1-Weak), (M2), (R1), (LR-NSD). Then for almost any $t \geq t_0$ : $\frac{d\mathcal{L}(\boldsymbol{\theta}_t)}{dt} < 0$, and it holds that $\mathcal{L}(\boldsymbol{\theta}_t) \overset{t\to\infty}{\longrightarrow} 0$, $\|\boldsymbol{\theta}_t\| \overset{t\to\infty}{\longrightarrow} \infty$, that $\gamma, \widetilde{\gamma}$ both converge to some $\gamma_\infty > 0$ and $\frac{\|\boldsymbol{\theta}_t\|}{\int_0^t \eta} \overset{t\to\infty}{\longrightarrow} 1$, $\frac{\varphi^{-1}\left( \log \frac{1}{\mathcal{L}(\boldsymbol{\theta}_t)} \right)}{\gamma_\infty \|\boldsymbol{\theta}_t\|^L} \overset{t\to\infty}{\longrightarrow} 1$.*

*Proof.* By Equation (33) and Equation (26), for almost any $t \geq t_0$, there exists $\mathbf{g}_t \in \partial \mathcal{L}(\boldsymbol{\theta}_t)$ with

$$\frac{d\mathcal{L}(\boldsymbol{\theta}_t)}{dt} = -\eta(t)\|\mathbf{g}_t\|_\star \leq -\frac{1}{\|\boldsymbol{\theta}_t\|} \eta(t) L \mathcal{L} \frac{\varphi^{-1}(\log \frac{1}{\mathcal{L}})}{(\varphi^{-1})'(\log \frac{1}{\mathcal{L}})} < 0 . \tag{34}$$

Note that $\varphi^{-1}(\log \frac{1}{\mathcal{L}}) > 0$ since by Theorem C.9, $\widetilde{\gamma}(\boldsymbol{\theta}_t) \geq \widetilde{\gamma}(\boldsymbol{\theta}_{t_0}) > 0$. Rearranging,

$$-\frac{1}{L} \frac{d\mathcal{L}(\boldsymbol{\theta}_t)}{dt} \frac{1}{\mathcal{L}} \left( \varphi^{-1}\left( \log \frac{1}{\mathcal{L}} \right) \right)^{\frac{1}{L}-1} (\varphi^{-1})'\left( \log \frac{1}{\mathcal{L}} \right) \geq \eta(t) \frac{(\varphi^{-1}\left( \log \frac{1}{\mathcal{L}} \right))^{1/L}}{\|\boldsymbol{\theta}_t\|} = \eta(t)\widetilde{\gamma}(\boldsymbol{\theta}_t)^{1/L} .$$

Since $\widetilde{\gamma}$ is non-decreasing, positive and bounded by above (from Lemma C.2 item 2, $\widetilde{\gamma} \leq \gamma$, and $\gamma$ is bounded by continuity of $f$ and compactness of the unit $\|\cdot\|$-sphere), it converges to some $\gamma_\infty > 0$. Therefore, for every $\varepsilon > 0$ there exists some $t_\varepsilon$

with $\forall t \geq t_\varepsilon : \widetilde{\gamma} \geq \gamma_\infty - \varepsilon$. Integrating on $[t_\varepsilon, t]$:

$$\left(\varphi^{-1}\left(\log \frac{1}{\mathcal{L}(\boldsymbol{\theta}_t)}\right)\right)^{1/L} - \left(\varphi^{-1}\left(\log \frac{1}{\mathcal{L}(\boldsymbol{\theta}_{t_\varepsilon})}\right)\right)^{1/L} \geq (\gamma_\infty - \varepsilon)^{1/L} \int_{t_\varepsilon}^t \eta \,. \tag{35}$$

By Assumption (LR-NSD), the RHS tends to $\infty$ as $t \to \infty$, showing that $\varphi^{-1}(\log \frac{1}{\mathcal{L}(\boldsymbol{\theta}_t)}) \to \infty$, and therefore $\frac{1}{\mathcal{L}(\boldsymbol{\theta}_t)} \overset{t\to\infty}{\longrightarrow} \infty$, $\mathcal{L}(\boldsymbol{\theta}_t) \overset{t\to\infty}{\longrightarrow} 0$ and in particular $\|\boldsymbol{\theta}_t\| \to \infty$. Note that by Lemma C.2, it holds since $\|\boldsymbol{\theta}_t\| \to \infty$ that $|\gamma - \widetilde{\gamma}| \to 0$, so also $\gamma \to \gamma_\infty$. Moreover, we get that $\liminf_{t\to\infty} \frac{\varphi^{-1}\left(\log \frac{1}{\mathcal{L}(\boldsymbol{\theta}_t)}\right)}{\left(\int_0^t \eta\right)^L} \geq \gamma_\infty - \varepsilon$, but since this holds for any $\varepsilon > 0$ it holds that

$$\liminf_{t\to\infty} \frac{\varphi^{-1}\left(\log \frac{1}{\mathcal{L}(\boldsymbol{\theta}_t)}\right)}{\left(\int_0^t \eta\right)^L} \geq \gamma_\infty \,. \tag{36}$$

Note that by definition of $\widetilde{\gamma}$,

$$\frac{\varphi^{-1}\left(\log \frac{1}{\mathcal{L}(\boldsymbol{\theta}_t)}\right)}{\|\boldsymbol{\theta}_t\|^L} = \widetilde{\gamma}(\boldsymbol{\theta}_t) \overset{t\to\infty}{\longrightarrow} \gamma_\infty \,. \tag{37}$$

Putting together Equations (36), (37) implies

$$\liminf_{t\to\infty} \frac{\|\boldsymbol{\theta}_t\|}{\int_0^t \eta} = \liminf_{t\to\infty} \left( \frac{\|\boldsymbol{\theta}_t\|^L}{\varphi^{-1}\left(\log \frac{1}{\mathcal{L}(\boldsymbol{\theta}_t)}\right)} \frac{\varphi^{-1}\left(\log \frac{1}{\mathcal{L}(\boldsymbol{\theta}_t)}\right)}{\left(\int_0^t \eta\right)^L} \right)^{1/L} \geq \left(\frac{1}{\gamma_\infty}\gamma_\infty\right)^{1/L} = 1 \,.$$

It also holds by the triangle inequality and by Assumption (LR-NSD) that

$$\limsup_{t\to\infty} \frac{\|\boldsymbol{\theta}_t\|}{\int_0^t \eta} \leq \limsup_{t\to\infty} \frac{\|\boldsymbol{\theta}_0\| + \int_0^t \left\|\frac{d\boldsymbol{\theta}_s}{ds}\right\| ds}{\int_0^t \eta} = \limsup_{t\to\infty} \frac{\|\boldsymbol{\theta}_0\| + \int_0^t \eta}{\int_0^t \eta} = 1 \,.$$

Proving

$$\lim_{t\to\infty} \frac{\|\boldsymbol{\theta}_t\|}{\int_0^t \eta} = 1 \,.$$

$\square$

**Theorem C.11** (KKT Stationarity for NSD - Theorem 3.1)**.** *Let $\boldsymbol{\theta}_t$ be a trajectory of normalized steepest descent with respect to a norm $\|\cdot\|$ (Equation (3)) under Assumptions (M1-Weak), (M2), (R1), (LR-NSD). Then any limit point $\bar{\boldsymbol{\theta}}$ of $\frac{\boldsymbol{\theta}_t}{\|\boldsymbol{\theta}_t\|}$ is the direction of a KKT point of Problem (11) with the same norm $\|\cdot\|$.*

*Proof.* Let $\bar{\boldsymbol{\theta}}$ be a limit point of $\frac{\boldsymbol{\theta}_t}{\|\boldsymbol{\theta}_t\|}$. By Theorem C.8, and since by Lemma C.10, $\mathcal{L}(\boldsymbol{\theta}_t) \overset{t\to\infty}{\longrightarrow} 0$ and $\gamma(\boldsymbol{\theta}_t) \overset{t\to\infty}{\longrightarrow} \gamma_\infty > 0$, it suffices to find a sequence $t_n \overset{n\to\infty}{\longrightarrow} \infty$ for which $\frac{\boldsymbol{\theta}_{t_n}}{\|\boldsymbol{\theta}_{t_n}\|} \overset{n\to\infty}{\longrightarrow} \bar{\boldsymbol{\theta}}$ and $\left\langle \frac{\boldsymbol{\theta}_{t_n}}{\|\boldsymbol{\theta}_{t_n}\|}, -\frac{\mathbf{g}_{t_n}}{\|\mathbf{g}_{t_n}\|_\star} \right\rangle \to 1$.

We construct a sequence $t_n$ by induction, taking $t_0$ as the base. Suppose $t_0 < t_1 < ... < t_{n-1}$ have already been constructed. By Lemma C.10, $\widetilde{\gamma}(\boldsymbol{\theta}_t) \to \gamma_\infty > 0$. Using this together with the fact that $\bar{\boldsymbol{\theta}}$ is a limit point of $\frac{\boldsymbol{\theta}_t}{\|\boldsymbol{\theta}_t\|}$, there exists $s_n > t_{n-1}+1$ with the following:

$$\left\| \frac{\boldsymbol{\theta}_{s_n}}{\|\boldsymbol{\theta}_{s_n}\|} - \bar{\boldsymbol{\theta}} \right\| \leq \frac{1}{n} \quad \text{and} \quad \frac{1}{L}\log \frac{\gamma_\infty}{\widetilde{\gamma}(\boldsymbol{\theta}_{s_n})} \leq \frac{1}{n^2} \,.$$

Notice that since $\|\boldsymbol{\theta}_t\| \to \infty$,

$$\int_{s_n}^\infty \frac{\left\|\frac{d\boldsymbol{\theta}_t}{dt}\right\|}{\|\boldsymbol{\theta}_t\|} dt \geq \int_{s_n}^\infty \frac{\frac{d\|\boldsymbol{\theta}_t\|}{dt}}{\|\boldsymbol{\theta}_t\|} dt = \int_{s_n}^\infty \frac{d\log \|\boldsymbol{\theta}_t\|}{dt} dt = \lim_{t\to\infty} \log \frac{\|\boldsymbol{\theta}_t\|}{\|\boldsymbol{\theta}_{s_n}\|} = \infty \,.$$

Therefore, there exists $s'_n > s_n$ with $\int_{s_n}^{s'_n} \frac{\left\|\frac{d\boldsymbol{\theta}_t}{dt}\right\|}{\|\boldsymbol{\theta}_t\|} dt = \frac{1}{n}$. Now, from the proof of Theorem C.9, it holds that

$$\frac{d\log\widetilde{\gamma}(\boldsymbol{\theta}_t)}{dt} \geq \frac{L\eta(t)\|\mathbf{g}_t\|_\star}{\langle\boldsymbol{\theta}_t, -\mathbf{g}_t\rangle} - \frac{L\eta(t)}{\|\boldsymbol{\theta}_t\|} = \frac{L\left\|\frac{d\boldsymbol{\theta}_t}{dt}\right\|}{\|\boldsymbol{\theta}_t\|}\left(\frac{1}{\left\langle -\frac{\mathbf{g}_t}{\|\mathbf{g}_t\|_\star}, \frac{\boldsymbol{\theta}_t}{\|\boldsymbol{\theta}_t\|}\right\rangle} - 1\right).$$

Therefore, rearranging terms and integrating on $[s_n, s'_n]$:

$$\frac{1}{L}\log\frac{\widetilde{\gamma}(s'_n)}{\widetilde{\gamma}(s_n)} \geq \int_{s_n}^{s'_n}\frac{\left\|\frac{d\boldsymbol{\theta}_t}{dt}\right\|}{\|\boldsymbol{\theta}_t\|}\left(\frac{1}{\left\langle -\frac{\mathbf{g}_t}{\|\mathbf{g}_t\|_\star}, \frac{\boldsymbol{\theta}_t}{\|\boldsymbol{\theta}_t\|}\right\rangle} - 1\right) dt.$$

By a standard proof by contradiction there exists a (non-zero measure set of points) $t_n \in (s_n, s'_n)$ with

$$\frac{1}{\left\langle \frac{\boldsymbol{\theta}_{t_n}}{\|\boldsymbol{\theta}_{t_n}\|}, -\frac{\mathbf{g}_{t_n}}{\|\mathbf{g}_{t_n}\|_\star}\right\rangle} - 1 \leq \frac{1}{L}\frac{\log\frac{\widetilde{\gamma}(s'_n)}{\widetilde{\gamma}(s_n)}}{\int_{s_n}^{s'_n}\frac{\left\|\frac{d\boldsymbol{\theta}_t}{dt}\right\|}{\|\boldsymbol{\theta}_t\|}dt} \leq \frac{1}{n},$$

because otherwise for almost every $t \in (s_n, s'_n)$ we have the opposite inequality, and so

$$\int_{s_n}^{s'_n}\frac{\left\|\frac{d\boldsymbol{\theta}_t}{dt}\right\|}{\|\boldsymbol{\theta}_t\|}\left(\frac{1}{\left\langle -\frac{\mathbf{g}_t}{\|\mathbf{g}_t\|_\star}, \frac{\boldsymbol{\theta}_t}{\|\boldsymbol{\theta}_t\|}\right\rangle} - 1\right) dt > \frac{1}{L}\frac{\log\frac{\widetilde{\gamma}(s'_n)}{\widetilde{\gamma}(s_n)}}{\int_{s_n}^{s'_n}\frac{\left\|\frac{d\boldsymbol{\theta}_t}{dt}\right\|}{\|\boldsymbol{\theta}_t\|}dt}\int_{s_n}^{s'_n}\frac{\left\|\frac{d\boldsymbol{\theta}_t}{dt}\right\|}{\|\boldsymbol{\theta}_t\|}dt = \frac{1}{L}\log\frac{\widetilde{\gamma}(s'_n)}{\widetilde{\gamma}(s_n)},$$

a contradiction. Thus, $\left\langle \frac{\boldsymbol{\theta}_{t_n}}{\|\boldsymbol{\theta}_{t_n}\|}, -\frac{\mathbf{g}_{t_n}}{\|\mathbf{g}_{t_n}\|_\star}\right\rangle \geq \frac{1}{1+\frac{1}{n}} \overset{n\to\infty}{\longrightarrow} 1$. The opposite inequality follows from the definition of the dual norm, so in fact $\left\langle \frac{\boldsymbol{\theta}_{t_n}}{\|\boldsymbol{\theta}_{t_n}\|}, -\frac{\mathbf{g}_{t_n}}{\|\mathbf{g}_{t_n}\|_\star}\right\rangle \overset{n\to\infty}{\longrightarrow} 1$. Furthermore, we claim the sequence $\frac{\boldsymbol{\theta}_{t_n}}{\|\boldsymbol{\theta}_{t_n}\|}$ converges to $\bar{\boldsymbol{\theta}}$:

$$\left\|\frac{\boldsymbol{\theta}_{t_n}}{\|\boldsymbol{\theta}_{t_n}\|} - \bar{\boldsymbol{\theta}}\right\| \leq \left\|\frac{\boldsymbol{\theta}_{t_n}}{\|\boldsymbol{\theta}_{t_n}\|} - \frac{\boldsymbol{\theta}_{s_n}}{\|\boldsymbol{\theta}_{s_n}\|}\right\| + \left\|\frac{\boldsymbol{\theta}_{s_n}}{\|\boldsymbol{\theta}_{s_n}\|} - \bar{\boldsymbol{\theta}}\right\| \leq \left\|\frac{\boldsymbol{\theta}_{t_n}}{\|\boldsymbol{\theta}_{t_n}\|} - \frac{\boldsymbol{\theta}_{s_n}}{\|\boldsymbol{\theta}_{s_n}\|}\right\| + \frac{1}{n}$$

$$\leq \frac{1}{n} + \int_{s_n}^{t_n}\left\|\frac{d\frac{\boldsymbol{\theta}_t}{\|\boldsymbol{\theta}_t\|}}{dt}\right\| dt \leq \frac{1}{n} + 2\int_{s_n}^{t_n}\frac{\left\|\frac{d\boldsymbol{\theta}_t}{dt}\right\|}{\|\boldsymbol{\theta}_t\|}dt \leq \frac{3}{n} \to 0,$$

where we used Equation (24) in the second to last inequality.

$\square$

## C.4. Approximate Steepest Descent

In this section we prove a KKT result for trajectories $\boldsymbol{\theta}_t$ approximating steepest descent. We provide the following definition:

**Definition C.12** (Approximate Steepest Descent). We say an arc $\boldsymbol{\theta}_t$ is a trajectory of Approximate Steepest Descent with respect to a norm $\|\cdot\|$ if eventually $\exists \mathbf{g}_t \neq \mathbf{0} \in \partial\mathcal{L}(\boldsymbol{\theta}_t)$ for almost any $t$, and there exist $\nu(t) > 0, R_{\max} > 0$ with:

1. $\lim_{t\to\infty} N(t) := \lim_{t\to\infty}\int_0^t \nu = \infty$

2. $\limsup_{t\to\infty}\frac{\|\boldsymbol{\theta}_t\|}{N(t)} \leq R_{\max}$

3. $\operatorname{ess\,liminf}_{t\to\infty} r(t) \geq 1$, where

$$r(t) \overset{a.e.}{=} \sup_{\mathbf{g}_t \in \partial\mathcal{L}(\boldsymbol{\theta}_t)\backslash\{\mathbf{0}\}}\left\langle \frac{1}{\nu(t)}\frac{d\boldsymbol{\theta}_t}{dt}, -\frac{\mathbf{g}_t}{\|\mathbf{g}_t\|_\star}\right\rangle.$$

Note that as $\partial\mathcal{L}(\boldsymbol{\theta}_t)$ is a compact set for any locally Lipschitz $\mathcal{L}$, the supremum in the definition of $r(t)$ is attained by some $\mathbf{g}_t \in \partial\mathcal{L}(\boldsymbol{\theta}_t)$.

Some motivation is in order. A possible and natural choice of $\nu(t)$ is $\nu(t) = \left\|\frac{d\boldsymbol{\theta}_t}{dt}\right\|$. This choice is motivated by the fact that for exact Steepest Flow, we have by definition for almost any $t$

$$r(t) = \left\langle \frac{1}{\nu(t)}\frac{d\boldsymbol{\theta}_t}{dt}, -\frac{\mathbf{g}_t}{\|\mathbf{g}_t\|_\star}\right\rangle = 1,$$

and

$$\|\boldsymbol{\theta}_t\| \leq \|\boldsymbol{\theta}_0\| + \int_0^t \nu .$$

Also, the following type of trajectory, which is a simpler and more "natural" definition of Approximate Steepest Descent, is also covered by our definition when choosing $\nu(t) = \left\|\frac{d\boldsymbol{\theta}_t}{dt}\right\|$:

$$\left\langle \frac{\frac{d\boldsymbol{\theta}_t}{dt}}{\left\|\frac{d\boldsymbol{\theta}_t}{dt}\right\|}, -\frac{\mathbf{g}_t}{\|\mathbf{g}_t\|_\star} \right\rangle \overset{t\to\infty}{\longrightarrow} 1 .$$

It is crucial in our analysis for an inner product of this type to tend to 1. However, we can allow $\nu(t)$ to be momentarily larger than $\left\|\frac{d\boldsymbol{\theta}_t}{dt}\right\|$, as long as on average it is not so (condition 2, when $R_{\max} \leq 1$). Thus the definition allows more flexibility for adaptive algorithms such as Adam, since we can choose $\nu(t)$ as an external learning rate, with $\frac{d\boldsymbol{\theta}_t}{dt}$ affected both by $\nu(t)$ and by the dynamics. In particular this flexibility will be invaluable for the main result of KKT convergence for Adam.

**Lemma C.13.** *If $\boldsymbol{\theta}_t$ is an arc with eventually $\exists \mathbf{g}_t \neq \mathbf{0} \in \partial\mathcal{L}(\boldsymbol{\theta}_t)$ for almost any $t$,* ess $\liminf r(t) \geq 1$ *for* $r(t) \overset{a.e.}{=}$ $\sup_{\mathbf{g}_t \in \partial\mathcal{L}(\boldsymbol{\theta}_t)\backslash\{\mathbf{0}\}} \left\langle \frac{1}{\nu(t)}\frac{d\boldsymbol{\theta}_t}{dt}, -\frac{\mathbf{g}_t}{\|\mathbf{g}_t\|_\star} \right\rangle, \nu(t) = \left\|\frac{d\boldsymbol{\theta}_t}{dt}\right\|,$ *and* $\int_0^t \nu \overset{t\to\infty}{\longrightarrow} \infty$, *then $\boldsymbol{\theta}_t$ is a trajectory of Approximate Steepest Descent with $R_{max} \leq 1$.*

*Proof.* According to Definition C.12, it remains to show that $\limsup_{t\to\infty} \frac{\|\boldsymbol{\theta}_t\|}{N(t)} \leq 1$ where $N(t) = \int_0^t \nu = \int_0^t \left\|\frac{d\boldsymbol{\theta}_s}{ds}\right\| ds$. By the triangle inequalities, $\|\boldsymbol{\theta}_t\| \leq \|\boldsymbol{\theta}_0\| + N(t)$, and since $N(t) \overset{t\to\infty}{\longrightarrow} \infty$,

$$\limsup_{t\to\infty} \frac{\|\boldsymbol{\theta}_t\|}{N(t)} \leq \limsup_{t\to\infty}\left(1 + \frac{\|\boldsymbol{\theta}_0\|}{N(t)}\right) = 1$$

$\square$

**Lemma C.14.** *If $\boldsymbol{\theta}_t$ is a trajectory of Approximate Steepest Descent, then for almost all $t \geq 0$ there exists $\mathbf{g}_t \in \partial\mathcal{L}(\boldsymbol{\theta}_t)$ with:*

$$r(t)\nu(t) = \frac{\frac{-d\mathcal{L}}{dt}}{\|\mathbf{g}_t\|_\star} \leq \left\|\frac{d\boldsymbol{\theta}_t}{dt}\right\| ,$$

*and in particular, $\frac{d\mathcal{L}}{dt} < 0$ for almost all large enough $t$.*

*Proof.* For almost any $t$, let $\mathbf{g}_t \neq \mathbf{0} \in \partial\mathcal{L}(\boldsymbol{\theta}_t)$ be such that $r(t) = \left\langle \frac{1}{\nu(t)}\frac{d\boldsymbol{\theta}_t}{dt}, -\frac{\mathbf{g}_t}{\|\mathbf{g}_t\|_\star} \right\rangle$. By the chain rule, for almost any $t$,

$$\frac{d\mathcal{L}}{dt} = \left\langle \mathbf{g}_t, \frac{d\boldsymbol{\theta}_t}{dt} \right\rangle ,$$

so

$$-\frac{d\mathcal{L}}{dt} \leq \left|\frac{d\mathcal{L}}{dt}\right| = \left|\left\langle \mathbf{g}_t, \frac{d\boldsymbol{\theta}_t}{dt} \right\rangle\right| \leq \|\mathbf{g}_t\|_\star \left\|\frac{d\boldsymbol{\theta}_t}{dt}\right\| ,$$

and

$$-\frac{d\mathcal{L}}{dt} = \left\langle -\mathbf{g}_t, \frac{d\boldsymbol{\theta}_t}{dt} \right\rangle = r(t)\nu(t)\|\mathbf{g}_t\|_\star , \tag{38}$$

so $\frac{d\mathcal{L}}{dt} < 0$ for all $t$ with $r(t) > 0$, which occurs for almost all large enough $t$ since ess $\liminf_{t\to\infty} r(t) \geq 1$. $\square$

**Lemma C.15** (Descent Lemma for Approximate SD). *Assume $\boldsymbol{\theta}_t$ is a trajectory of Approximate Steepest Descent, and assume that there exists $t_0 \geq 0, \gamma_{min} > 0$ with $\forall t \geq t_0 : \boldsymbol{\theta}_t \neq \mathbf{0}$ and $\gamma(\boldsymbol{\theta}_t) > \gamma_{min} > 0$. Then $\mathcal{L}(\boldsymbol{\theta}_t) \overset{t\to\infty}{\longrightarrow} 0, \|\boldsymbol{\theta}_t\| \overset{t\to\infty}{\longrightarrow} \infty$.*

*Proof.* Take $t_1 \geq t_0$ with:

$$\forall t \geq t_1 : r(t) \geq \frac{1}{2} \text{ a.e. }, \quad \gamma(\boldsymbol{\theta}_t) > \gamma_{min}$$

By Lemma C.14, for almost any $t \geq t_1$ we have $\frac{d\mathcal{L}}{dt} < 0$, so

$$\mathcal{L}(\boldsymbol{\theta}_t) = \mathcal{L}(\boldsymbol{\theta}_{t_1}) + \int_{t_1}^{t} \frac{d\mathcal{L}}{ds} ds < \mathcal{L}(\boldsymbol{\theta}_{t_1}) \leq m \cdot \ell\left(\gamma_{min} \|\boldsymbol{\theta}_{t_1}\|^L\right) < m \cdot \ell(0) .$$

In particular, this implies that $\|\boldsymbol{\theta}_t\| \geq \Omega(1)$, because if towards a contradiction there existed a sequence $t_n \stackrel{n\to\infty}{\longrightarrow} \infty$ with $\boldsymbol{\theta}_{t_n} \stackrel{n\to\infty}{\longrightarrow} \mathbf{0}$, this would imply by continuity of $f$ that $\forall i \in [m] : f(\mathbf{x}_i; \boldsymbol{\theta}_{t_n}) \to f(\mathbf{x}_i; \mathbf{0}) = 0$ and therefore $\mathcal{L}(\boldsymbol{\theta}_{t_n}) \stackrel{n\to\infty}{\longrightarrow} m \cdot \ell(0)$, a contradiction. Therefore denote $N_{min} = \inf_{t \geq t_1} \|\boldsymbol{\theta}_t\| > 0$. By Lemma C.2, there exists $C > 0$ with $\|\mathbf{g}_t\|_\star \geq C\mathcal{L}(\boldsymbol{\theta}_t)\|\boldsymbol{\theta}_t\|^{L-1}$.

So, by Lemma C.14, for almost any $t \geq t_1$:

$$-\frac{d\mathcal{L}}{dt} = r(t)\nu(t)\|\mathbf{g}_t\|_\star \geq r(t)\nu(t)C\|\boldsymbol{\theta}_t\|^{L-1}\mathcal{L}$$

$$\geq \frac{N_{min}^{L-1}C}{2}\nu(t)\mathcal{L} .$$

Dividing by $\mathcal{L}$ and integrating both sides on $[t_1, t]$ we get

$$-\log\mathcal{L}(\boldsymbol{\theta}_t) - (-\log\mathcal{L}(\boldsymbol{\theta}_{t_1})) \geq \frac{N_{min}^{L-1}C}{2}(N(t) - N(t_1)) \stackrel{t\to\infty}{\longrightarrow} \infty ,$$

implying $\mathcal{L} \to 0$ and therefore $\|\boldsymbol{\theta}_t\| \to \infty$. $\qquad\square$

**Lemma C.16.** *Assume $\boldsymbol{\theta}_t$ is a trajectory of Approximate Steepest Descent with $R_{max} \leq 1$, and assume that eventually $\boldsymbol{\theta}_t \neq 0$ and $\lim_{t\to\infty} \frac{\boldsymbol{\theta}_t}{\|\boldsymbol{\theta}_t\|} = \bar{\boldsymbol{\theta}}$ with $\gamma(\bar{\boldsymbol{\theta}}) > 0$. Then there exists a sequence $t_n \to \infty$ with $\left\langle \frac{\boldsymbol{\theta}_{t_n}}{\|\boldsymbol{\theta}_{t_n}\|}, -\frac{\mathbf{g}_{t_n}}{\|\mathbf{g}_{t_n}\|_\star} \right\rangle \to 1$.*

*Proof.* Since $\lim_{t\to\infty} \frac{\boldsymbol{\theta}_t}{\|\boldsymbol{\theta}_t\|} = \bar{\boldsymbol{\theta}}$ with $\gamma(\bar{\boldsymbol{\theta}}) > 0$, eventually $\gamma(\boldsymbol{\theta}_t) > \frac{1}{2}\gamma(\bar{\boldsymbol{\theta}}) > 0$, so from Lemma C.15, $\mathcal{L}(\boldsymbol{\theta}_t) \to 0, \|\boldsymbol{\theta}_t\| \to \infty$. By Lemma C.2 we conclude $|\gamma(\boldsymbol{\theta}_t) - \widetilde{\gamma}(\boldsymbol{\theta}_t)| \to 0$, so $\gamma(\boldsymbol{\theta}_t), \widetilde{\gamma}(\boldsymbol{\theta}_t)$ both converge to $\gamma(\bar{\boldsymbol{\theta}}) > 0$, hence also $\log\widetilde{\gamma}(\boldsymbol{\theta}_t)$ converges. We build a sequence $t_n$ by induction. Choose $t_0$ with $\forall t \geq t_0 : \mathcal{L}(\boldsymbol{\theta}_t) < \ell(0)$, and suppose $t_0 < ... < t_{n-1}$ have been chosen. Choose $\tau_1 > t_{n-1} + 1$ with

$$\sup_{\tau_2 \geq \tau_1} \log\left(\frac{\widetilde{\gamma}(\boldsymbol{\theta}_{\tau_2})}{\widetilde{\gamma}(\boldsymbol{\theta}_{\tau_1})}\right) \leq \frac{L}{n}$$

$$\forall t > \tau_1 : r(t) > 1 - \frac{1}{n} \text{ a.e.}$$

For almost any $t \geq t_0$, $\exists \mathbf{g}_t \neq \mathbf{0} \in \partial\mathcal{L}(\boldsymbol{\theta}_t)$ with:

$$\frac{d\log\widetilde{\gamma}}{dt} = \frac{d}{dt}\log\varphi^{-1}\left(\log\frac{1}{\mathcal{L}(\boldsymbol{\theta}_t)}\right) - L\frac{d}{dt}\log\|\boldsymbol{\theta}_t\|$$

$$= -\frac{d\mathcal{L}(\boldsymbol{\theta}_t)}{dt}\frac{(\varphi^{-1})'(\log\frac{1}{\mathcal{L}(\boldsymbol{\theta}_t)})}{\mathcal{L}(\boldsymbol{\theta}_t)\varphi^{-1}(\log\frac{1}{\mathcal{L}(\boldsymbol{\theta}_t)})} - L\frac{d}{dt}\log\|\boldsymbol{\theta}_t\|$$

$$= \nu(t)\|\mathbf{g}_t\|_\star r(t)\frac{(\varphi^{-1})'(\log\frac{1}{\mathcal{L}(\boldsymbol{\theta}_t)})}{\mathcal{L}(\boldsymbol{\theta}_t)\varphi^{-1}(\log\frac{1}{\mathcal{L}(\boldsymbol{\theta}_t)})} - L\frac{d}{dt}\log\|\boldsymbol{\theta}_t\|$$

$$\geq \frac{L\nu(t)}{\|\boldsymbol{\theta}_t\|}\frac{r(t)}{\left\langle\frac{\boldsymbol{\theta}_t}{\|\boldsymbol{\theta}_t\|}, -\frac{\mathbf{g}_t}{\|\mathbf{g}_t\|_\star}\right\rangle} - L\frac{d}{dt}\log\|\boldsymbol{\theta}_t\| .$$

In the last inequality we used Equation (26), namely $\langle\boldsymbol{\theta}_t, -\mathbf{g}_t\rangle \geq L\mathcal{L}(\boldsymbol{\theta}_t)\frac{\varphi^{-1}(\log\frac{1}{\mathcal{L}(\boldsymbol{\theta}_t)})}{(\varphi^{-1})'(\log\frac{1}{\mathcal{L}(\boldsymbol{\theta}_t)})}$, implying $\frac{(\varphi^{-1})'(\log\frac{1}{\mathcal{L}(\boldsymbol{\theta}_t)})}{\mathcal{L}(\boldsymbol{\theta}_t)\varphi^{-1}(\log\frac{1}{\mathcal{L}(\boldsymbol{\theta}_t)})} \geq \frac{L}{\langle\boldsymbol{\theta}_t, -\mathbf{g}_t\rangle}$.

So, rearranging terms and integrating on $[\tau_1, \tau_2]$ for any $\tau_2 > \tau_1$,

$$\frac{1}{L}\log\left(\frac{\widetilde{\gamma}(\boldsymbol{\theta}_{\tau_2})}{\widetilde{\gamma}(\boldsymbol{\theta}_{\tau_1})}\right) + \log\frac{\|\boldsymbol{\theta}_{\tau_2}\|}{\|\boldsymbol{\theta}_{\tau_1}\|} \geq \int_{\tau_1}^{\tau_2} \frac{\nu(t)}{\|\boldsymbol{\theta}_t\|}\frac{r(t)}{\left\langle\frac{\boldsymbol{\theta}_t}{\|\boldsymbol{\theta}_t\|}, -\frac{\mathbf{g}_t}{\|\mathbf{g}_t\|_\star}\right\rangle}dt .$$

In particular, by a standard proof-by-contradiction argument (see Theorem C.11), for every $\tau_2$ there exists a non-zero measure set of points $t_n \in (\tau_1, \tau_2)$ with

$$\frac{\frac{1}{L} \log \left( \frac{\widetilde{\gamma}(\boldsymbol{\theta}_{\tau_2})}{\widetilde{\gamma}(\boldsymbol{\theta}_{\tau_1})} \right) + \log \frac{\|\boldsymbol{\theta}_{\tau_2}\|}{\|\boldsymbol{\theta}_{\tau_1}\|}}{\int_{\tau_1}^{\tau_2} \frac{\nu(t)}{\|\boldsymbol{\theta}_t\|} dt} \geq \frac{r(t_n)}{\left\langle \frac{\boldsymbol{\theta}_{t_n}}{\|\boldsymbol{\theta}_{t_n}\|}, -\frac{\mathbf{g}_{t_n}}{\|\mathbf{g}_{t_n}\|_\star} \right\rangle} \ .$$

In particular $t_n$ can be chosen with $r(t_n) > 1 - \frac{1}{n}$.

Note that for any $F(t), G(t)$ with $\limsup \frac{F}{G} \leq 1$ and $\int_0^\infty G = \infty$ it holds that $\limsup \frac{\int_0^t F}{\int_0^t G} \leq 1$. Indeed, for any $\varepsilon > 0$ take $t_\varepsilon$ with $\forall t \geq t_\varepsilon : F(t) \leq (1+\varepsilon)G(t)$, so $\frac{\int_0^t F}{\int_0^t G} = \frac{\int_0^{t_\varepsilon} F + \int_{t_\varepsilon}^t F}{\int_0^{t_\varepsilon} G + \int_{t_\varepsilon}^t G} \leq \frac{\int_0^{t_\varepsilon} F + (1+\varepsilon)\int_{t_\varepsilon}^t G}{\int_0^{t_\varepsilon} G + \int_{t_\varepsilon}^t G} \overset{t \to \infty}{\longrightarrow} 1 + \varepsilon$. Also, note that if $\limsup \frac{F}{G} \leq 1$ and $\int_0^\infty F = \infty$ then in particular $\int_0^\infty G = \infty$.

In our case, since $\limsup_{t \to \infty} \frac{\nu(t)}{N(t)} / \frac{\nu(t)}{\|\boldsymbol{\theta}_t\|} = \limsup_{t \to \infty} \frac{\|\boldsymbol{\theta}_t\|}{N(t)} \leq 1$, and $\int_{\tau_1}^\infty \frac{\nu(t)}{N(t)} dt = \lim_{t \to \infty} \log \frac{N(t)}{N(\tau_1)} = \infty$ (implying also $\int_{\tau_1}^\infty \frac{\nu(t)}{\|\boldsymbol{\theta}_t\|} dt = \infty$), there exists $q(\tau_2)$ with $q(\tau_2) \overset{\tau_2 \to \infty}{\longrightarrow} 1$ and

$$\int_{\tau_1}^{\tau_2} \frac{\nu(t)}{\|\boldsymbol{\theta}_t\|} dt \geq q(\tau_2) \int_{\tau_1}^{\tau_2} \frac{\nu(t)}{N(t)} dt = q(\tau_2) \log \left( \frac{N(\tau_2)}{N(\tau_1)} \right) \ .$$

Since $\|\boldsymbol{\theta}_t\|, N(t) \to \infty$ and again since $\limsup_{t \to \infty} \frac{\|\boldsymbol{\theta}_t\|}{N(t)} \leq 1$ there exists large enough $\tau_2$ with $\int_{\tau_1}^{\tau_2} \frac{\nu(t)}{\|\boldsymbol{\theta}_t\|} dt \geq 1$ and

$$1 + \frac{1}{n} \geq \frac{\log \frac{\|\boldsymbol{\theta}_{\tau_2}\|}{\|\boldsymbol{\theta}_{\tau_1}\|}}{q(\tau_2) \log \frac{N(\tau_2)}{N(\tau_1)}} \ .$$

Altogether for such $\tau_2$ and $t_n \in (\tau_1, \tau_2)$ as above,

$$\frac{\frac{1}{L} \log \left( \frac{\widetilde{\gamma}(\boldsymbol{\theta}_{\tau_2})}{\widetilde{\gamma}(\boldsymbol{\theta}_{\tau_1})} \right) + \log \frac{\|\boldsymbol{\theta}_{\tau_2}\|}{\|\boldsymbol{\theta}_{\tau_1}\|}}{\int_{\tau_1}^{\tau_2} \frac{\nu(t)}{\|\boldsymbol{\theta}_t\|} dt} \leq \frac{1}{n} + \left( 1 + \frac{1}{n} \right) = 1 + \frac{2}{n} \ .$$

So,

$$\left\langle \frac{\boldsymbol{\theta}_{t_n}}{\|\boldsymbol{\theta}_{t_n}\|}, -\frac{\mathbf{g}_{t_n}}{\|\mathbf{g}_{t_n}\|_\star} \right\rangle \geq \frac{1 - \frac{1}{n}}{1 + \frac{2}{n}} \overset{n \to \infty}{\longrightarrow} 1 \ .$$

And the other direction is immediate by the definition of the dual norm, implying $\left\langle \frac{\boldsymbol{\theta}_{t_n}}{\|\boldsymbol{\theta}_{t_n}\|}, -\frac{\mathbf{g}_{t_n}}{\|\mathbf{g}_{t_n}\|_\star} \right\rangle \overset{n \to \infty}{\longrightarrow} 1$. $\qquad \square$

**Theorem C.17** (KKT Stationarity for Approximate SD ). *Assume $\boldsymbol{\theta}_t$ is a trajectory of Approximate Steepest Descent with $R_{max} \leq 1$, and assume that eventually $\boldsymbol{\theta}_t \neq \mathbf{0}$ and $\boldsymbol{\theta}_t$ converges in direction to some $\bar{\boldsymbol{\theta}}$ with $\gamma(\bar{\boldsymbol{\theta}}) > 0$. Then $\bar{\boldsymbol{\theta}}$ is along the direction of a KKT point of Equation* (11).

*Proof.* Follows from Theorem C.8 using Lemma C.16 and Lemma C.15. $\qquad \square$

### C.5. Momentum Steepest Descent

In this section we show that under Assumptions (M1-Weak), (M2), (LR-MSD), (T1), (T2), (T3), normalized and unnormalized momentum steepest descent are approximate steepest descent algorithms, which allows us to infer Theorem C.21, namely that the directional limit point is a KKT point of the $\|\cdot\|$-max-margin problem. Recall that (T3) is implied if strengthening (M1-Weak) to (M1).

**Definition C.18.** We denote for a choice $\mathbf{h}(\mathbf{x}_i; \boldsymbol{\theta}_t) \in \partial f(\mathbf{x}_i; \boldsymbol{\theta}_t)$ and $\forall i, i' \in [m], j \in [p]$ (see Definition B.2 for the definition of $A$):

$$\bar{\mathbf{h}}_t[i, j] := \frac{1}{\|\boldsymbol{\theta}_t\|^{L-1}} \mathbf{h}(x_i; \boldsymbol{\theta}_t)[j], \quad \bar{\mathbf{h}}_\infty[i, j] := \lim_{t \to \infty} \bar{\mathbf{h}}_t[i, j]$$

$$\mathbf{g}_t[i, j] = -\|\boldsymbol{\theta}_t\|^{L-1} \ell(z_i^t) \varphi'(z_i^t) y_i \bar{\mathbf{h}}_t[i, j], \quad \mathbf{g}_t^\infty[i, j] = -\|\boldsymbol{\theta}_t\|^{L-1} \ell(z_i^t) \varphi'(z_i^t) y_i \bar{\mathbf{h}}_\infty[i, j]$$

$$\mathbf{m}_t[i,j] = A(\mathbf{g}_t[i,j], c_1), \quad \mathbf{m}_t^\infty[i,j] = A(\mathbf{g}_t^\infty[i,j], c_1)$$

$$\mathbf{v}_t[i,i',j] = A(\mathbf{g}_t[i,j]\mathbf{g}_t[i',j], c_2), \quad \mathbf{v}_t^\infty[i,i',j] = A(\mathbf{g}_t^\infty[i,j]\mathbf{g}_t^\infty[i',j], c_2)$$

The notations derived from $\bar{\mathbf{h}}_\infty$ are well-defined for every $i \in [m]$ under (T3). Note that

$$\mathbf{g}_t[j] = \sum_{i\in[m]} \mathbf{g}_t[i,j], \quad \mathbf{m}_t[j] = \sum_{i\in[m]} \mathbf{m}_t[i,j], \quad \mathbf{v}_t[j] = \sum_{i,i'\in[m]} \mathbf{v}_t[i,i',j] .$$

The following is a key lemma showing that under certain conditions, the momentum estimates follow the asymptotically significant gradients with ratio approaching 1. Note that the lemma is independent of a specific optimization algorithm.

**Lemma C.19** (Asymptotic Momentum-Gradient Relations Under a Decaying Update). *Assume $\boldsymbol{\theta}_t$ is an arc of parameters of a model $f(\mathbf{x}; \boldsymbol{\theta})$ and let $\|\cdot\|$ be a norm. Let $\mathbf{g}_t \in \partial\mathcal{L}(\boldsymbol{\theta}_t)$ be a choice of subgradients and $\mathbf{m}_t, \mathbf{v}_t$ the appropriate momentum estimates for $\mathbf{g}_t, \mathbf{g}_t^2$ with momentum rates $c_1, c_2 > 0$. Assume that (M1-Weak), (M2), (T1), (T2), (T3) are satisfied. If $\left\|\frac{d\boldsymbol{\theta}_t}{dt}\right\| \le o\left(t^{\frac{1}{L}-1}\right)$, then:*

1. *For every $\varepsilon > 0$, denoting $J_\varepsilon(t) = \left\{j \in [p] : \frac{|\mathbf{g}_t[j]|}{\|\mathbf{g}_t\|_\star} > \varepsilon\right\}$ there exist for every $j \in [p]$ vanishing error terms $e_j(t), e_j'(t) \overset{t\to\infty}{\longrightarrow} 0$ with*

$$\forall t \ge 0, j \in J_\varepsilon(t) : \quad \mathbf{m}_t[j] = \mathbf{g}_t[j](1 + e_j(t)), \quad \sqrt{\mathbf{v}_t[j]} = |\mathbf{g}_t[j]|\left(1 + e_j'(t)\right) .$$

2. *It holds that*

$$\|\mathbf{m}_t - \mathbf{g}_t\|_\star \le o\left(\|\mathbf{g}_t\|_\star\right), \quad \frac{\|\mathbf{m}_t\|_\star}{\|\mathbf{g}_t\|_\star} \overset{t\to\infty}{\longrightarrow} 1, \quad \frac{\mathbf{m}_t}{\|\mathbf{m}_t\|_\star} - \frac{\mathbf{g}_t}{\|\mathbf{g}_t\|_\star} \overset{t\to\infty}{\longrightarrow} 0 ,$$

$$\left\|\mathbf{v}_t - \mathbf{g}_t^2\right\|_\star \le o\left(\left\|\mathbf{g}_t^2\right\|_\star\right), \quad \frac{\|\mathbf{v}_t\|_\star}{\|\mathbf{g}_t^2\|_\star} \overset{t\to\infty}{\longrightarrow} 1, \quad \frac{\mathbf{v}_t}{\|\mathbf{v}_t\|_\star} - \frac{\mathbf{g}_t^2}{\|\mathbf{g}_t^2\|_\star} \overset{t\to\infty}{\longrightarrow} 0 .$$

We first prove an auxiliary lemma.

**Lemma C.20.** *Let $\mathbf{u}, \mathbf{w} \in \mathbb{R}^p$ be nonzero vectors and let $\|\cdot\|_\star$ a norm. Let $\xi > 0$ with*

$$\|\mathbf{u} - \mathbf{w}\|_\star \le \xi \|\mathbf{w}\|_\star$$

*Then*

$$\left|\frac{\|\mathbf{u}\|_\star}{\|\mathbf{w}\|_\star} - 1\right| \le \xi, \quad \left\|\frac{\mathbf{u}}{\|\mathbf{u}\|_\star} - \frac{\mathbf{w}}{\|\mathbf{w}\|_\star}\right\|_\star \le 2\xi$$

*Proof.* Since by the triangle inequalities

$$\|\mathbf{w}\|_\star - \|\mathbf{u} - \mathbf{w}\|_\star \le \|\mathbf{u}\|_\star \le \|\mathbf{w}\|_\star + \|\mathbf{u} - \mathbf{w}\|_\star$$

This gives

$$\left|\frac{\|\mathbf{u}\|_\star}{\|\mathbf{w}\|_\star} - 1\right| \le \frac{\|\mathbf{u} - \mathbf{w}\|_\star}{\|\mathbf{w}\|_\star} \le \xi$$

Now,

$$\begin{aligned}
\left\|\frac{\mathbf{u}}{\|\mathbf{u}\|_\star} - \frac{\mathbf{w}}{\|\mathbf{w}\|_\star}\right\|_\star &\le \left\|\frac{\mathbf{u}}{\|\mathbf{u}\|_\star} - \frac{\mathbf{u}}{\|\mathbf{w}\|_\star}\right\|_\star + \left\|\frac{\mathbf{u}}{\|\mathbf{w}\|_\star} - \frac{\mathbf{w}}{\|\mathbf{w}\|_\star}\right\|_\star \\
&= \|\mathbf{u}\|_\star \left|\frac{1}{\|\mathbf{u}\|_\star} - \frac{1}{\|\mathbf{w}\|_\star}\right| + \frac{\|\mathbf{u} - \mathbf{w}\|_\star}{\|\mathbf{w}\|_\star} \\
&= \frac{\|\mathbf{u}\|_\star}{\|\mathbf{u}\|_\star}\left|1 - \frac{\|\mathbf{u}\|_\star}{\|\mathbf{w}\|_\star}\right| + \frac{\|\mathbf{u} - \mathbf{w}\|_\star}{\|\mathbf{w}\|_\star} \\
&\le 2\xi
\end{aligned}$$

$\square$

*Proof of Lemma C.19.* Denote $\forall j \in [p], I_j := \{i \in [m] : \bar{\mathbf{h}}_\infty[i,j] \neq 0\}$. For every $i,j$ with $i \in I_j$, it holds that $\mathbf{g}_t^\infty[i,j], \mathbf{g}_t[i,j]$ eventually have a constant (nonzero) sign and

$$\frac{\mathbf{g}_t^\infty[i,j]}{\mathbf{g}_t[i,j]} \overset{t \to \infty}{\longrightarrow} 1 . \tag{39}$$

We analyze $\frac{d \log |\mathbf{g}_t^\infty[i,j]|}{dt}$. One has for almost any $t$ (using the chain rule theorems A.4, A.6) that $\|\boldsymbol{\theta}_t\|, \ell(z_i^t), \varphi'(z_i^t)$ and therefore $\mathbf{g}_t^\infty[i,j]$ are differentiable w.r.t $t$. Note

$$\frac{d \log \ell(z_i^t)}{dt} = -\frac{d\varphi(z_i^t)}{dt} = -\varphi'(z_i^t) \cdot \frac{dz_i^t}{dt} ,$$

and for almost any $t$,

$$\left| \frac{dz_i^t}{dt} \right| = \left| \frac{dy_i f(\mathbf{x}_i; \boldsymbol{\theta}_t)}{dt} \right| = \left| \left\langle \frac{d\boldsymbol{\theta}_t}{dt}, y_i \mathbf{h}(\mathbf{x}_i; \boldsymbol{\theta}_t) \right\rangle \right| \leq \|\boldsymbol{\theta}_t\|^{L-1} \left\| \frac{d\boldsymbol{\theta}_t}{dt} \right\| \frac{\|\mathbf{h}(\mathbf{x}_i; \boldsymbol{\theta}_t)\|_\star}{\|\boldsymbol{\theta}_t\|^{L-1}} \overset{t \to \infty}{\longrightarrow} 0 .$$

The limit holds since $\frac{\|\mathbf{h}(\mathbf{x}_i; \boldsymbol{\theta}_t)\|_\star}{\|\boldsymbol{\theta}_t\|^{L-1}} \leq \mathcal{O}(1)$ ($f$ is locally Lipschitz, see the proof of Equation (25)) and $\left\| \frac{d\boldsymbol{\theta}_t}{dt} \right\| \leq \mathrm{o}\left(t^{\frac{1}{L}-1}\right)$, implying $\|\boldsymbol{\theta}_t\| \leq \mathrm{o}\left(t^{1/L}\right)$, therefore

$$\|\boldsymbol{\theta}_t\|^{L-1} \left\| \frac{d\boldsymbol{\theta}_t}{dt} \right\| \leq \mathrm{o}\left(t^{(1/L)(L-1)}t^{\frac{1}{L}-1}\right) = \mathrm{o}(1) .$$

Since $\varphi'$ is bounded, it also holds that $\mathrm{ess} \lim_{t \to \infty} \frac{d \log \ell(z_i^t)}{dt} = 0$. Also, since for large enough $t$, $\gamma(\boldsymbol{\theta}_t) \geq \frac{1}{2}\gamma(\bar{\boldsymbol{\theta}}) > 0$ and $\|\boldsymbol{\theta}_t\| \geq N_{min} > 0$ (Assumptions (T1), (T2)), and since $\varphi'$ is non-decreasing, $\varphi'$ is bounded from below by $\varphi'(N_{min}^L \gamma_{min}) > 0$. $\varphi''$ too is bounded, so

$$\mathrm{ess} \lim_{t \to \infty} \frac{d \log \varphi'(z_i^t)}{dt} = \mathrm{ess} \lim_{t \to \infty} \frac{\varphi''(z_i^t)\frac{dz_i^t}{dt}}{\varphi'(z_i^t)} = 0 .$$

Altogether

$$\begin{aligned} \left| \frac{d \log |\mathbf{g}_t^\infty[i,j]|}{dt} \right| &= \left| (L-1)\frac{d \log \|\boldsymbol{\theta}_t\|}{dt} + \frac{d \log \ell(z_i^t)}{dt} + \frac{d \log \varphi'(z_i^t)}{dt} \right| \\ &\leq (L-1)\frac{\left\| \frac{d\boldsymbol{\theta}_t}{dt} \right\|}{\|\boldsymbol{\theta}_t\|} + \left| \frac{d \log \ell(z_i^t)}{dt} \right| + \left| \frac{d \log \varphi'(z_i^t)}{dt} \right| \overset{t \to \infty}{\longrightarrow} 0 . \end{aligned} \tag{40}$$

The first term goes to 0 since $\left\| \frac{d\boldsymbol{\theta}_t}{dt} \right\| = \mathrm{o}(1)$ and $\|\boldsymbol{\theta}_t\| = \Omega(1)$ by Assumption (T1), and the rest of the terms have already been shown to go to 0.

Therefore, for every $i, i', j$ with $i, i' \in I_j$, it holds that $\mathrm{ess} \lim_{t \to \infty} \frac{d \log |\mathbf{g}_t^\infty[i,j]|}{dt} = 0$ and $\mathrm{ess} \lim_{t \to \infty} \frac{d \log |\mathbf{g}_t^\infty[i,j]\mathbf{g}_t^\infty[i',j]|}{dt} = 0$. Also, since $z_i^t$ is locally Lipschitz w.r.t $\boldsymbol{\theta}_t$ and $\ell, \varphi'$ are $C^1$, all of $\|\boldsymbol{\theta}_t\|, \ell(z_i^t), \varphi'(z_i^t)$ are locally Lipschitz w.r.t $\boldsymbol{\theta}_t$. Furthermore $\boldsymbol{\theta}_t$ is eventually Lipschitz w.r.t $t$ because $\left| \frac{d\|\boldsymbol{\theta}_t\|}{dt} \right| \leq \left\| \frac{d\boldsymbol{\theta}_t}{dt} \right\| \leq \mathrm{o}(1)$. Altogether $\mathbf{g}_t^\infty[i,j]$ is locally Lipschitz w.r.t $t$ and in particular locally absolutely continuous with regard to $t$. Therefore by Corollary B.8,

$$\frac{\mathbf{m}_t^\infty[i,j]}{\mathbf{g}_t^\infty[i,j]} \to 1, \quad \frac{\mathbf{v}_t^\infty[i,i',j]}{\mathbf{g}_t^\infty[i,j]\mathbf{g}_t^\infty[i',j]} \to 1 . \tag{41}$$

Also, the fact that $\frac{d \log |\mathbf{g}_t^\infty[i,j]|}{dt} \to 0$ implies that $\forall c > 0$ and $\forall i, i' \in I_j$,

$$\int_0^\infty e^{cs} |g_s^\infty[i,j]| \, ds = \infty, \quad \int_0^\infty e^{cs} |g_s^\infty[i',j]g_s^\infty[i',j]| \, ds = \infty .$$

So by Lemma B.5 2(c) and Equation (39),

$$\frac{\mathbf{m}_t[i,j]}{\mathbf{m}_t^\infty[i,j]} \to 1, \quad \frac{\mathbf{v}_t[i,i',j]}{\mathbf{v}_t^\infty[i,i',j]} \to 1 . \tag{42}$$

Putting together (39), (41), (42), we get $\forall i, i' \in I_j$:

$$\frac{\mathbf{m}_t[i,j]}{\mathbf{g}_t[i,j]} \xrightarrow{t\to\infty} 1, \quad \frac{\mathbf{v}_t[i,i',j]}{\mathbf{g}_t[i,j]\mathbf{g}_t[i',j]} \xrightarrow{t\to\infty} 1 . \tag{43}$$

Denote $G(t) = \|\boldsymbol{\theta}_t\|^{L-1} \ell(q_{\min}^t) > 0$, and notice that since $\forall t \geq 0, \exists i \in [m] : q_{\min}^t = \gamma(\boldsymbol{\theta}_t)\|\boldsymbol{\theta}_t\|^L = y_i f(\mathbf{x}_i; \boldsymbol{\theta}_t)$ it holds as in Equation (40) that $G$ is differentiable almost everywhere, locally absolutely continuous and $\mathrm{ess}\lim_{t\to\infty}\frac{d\log G}{dt} = 0$, implying by Corollary B.8 that $\frac{A(G,c)}{G} \to 1$ for any $c > 0$.

Since $\gamma(\bar{\boldsymbol{\theta}}) > 0$ and by Assumption (T1), by Lemma C.2 we have $\|\mathbf{g}_t\|_\star = \Theta(\|\boldsymbol{\theta}_t\|^{L-1}\mathcal{L}) = \Theta(G(t))$. Also note that for every $j \in [p]$ and $\bar{i} \notin I_j$ it holds that $\bar{\mathbf{h}}_\infty[\bar{i},j] = 0$, so $|\mathbf{g}_t[\bar{i},j]| \leq \mathrm{o}\left(\|\boldsymbol{\theta}_t\|^{L-1}\mathcal{L}(\boldsymbol{\theta}_t)\right) \leq \mathrm{o}(G(t))$. By Lemma B.5, item 2(c) (applied with $g = G, F = \mathbf{g}_t[\bar{i},j]$), it follows also that $|\mathbf{m}_t[\bar{i},j]| \leq \mathrm{o}(A(G,c_1)(t)) \leq \mathrm{o}(G(t)) \leq \mathrm{o}(\|\mathbf{g}_t\|_\star)$.

Therefore, for any $j \in [p]$,

$$\forall \bar{i} \notin I_j : |\mathbf{m}_t[\bar{i},j] - \mathbf{g}_t[\bar{i},j]| \leq |\mathbf{m}_t[\bar{i},j]| + |\mathbf{g}_t[\bar{i},j]| = \mathrm{o}(\|\mathbf{g}_t\|_\star) .$$

Also, for any $i \in [m], j \in [p] : |\mathbf{g}_t[i,j]| \leq \mathcal{O}\left(\|\boldsymbol{\theta}_t\|^{L-1}\mathcal{L}\right) = \mathcal{O}(\|\mathbf{g}_t\|_\star)$. Therefore by Equation (43),

$$\forall i \in I_j : |\mathbf{m}_t[i,j] - \mathbf{g}_t[i,j]| = \mathrm{o}(\mathbf{g}_t[i,j]) = \mathrm{o}(\|\mathbf{g}_t\|_\star) .$$

So altogether, for any $j \in [p]$

$$|\mathbf{m}_t[j] - \mathbf{g}_t[j]| \leq \sum_{i\in[m]} |\mathbf{m}_t[i,j] - \mathbf{g}_t[i,j]| = \mathrm{o}(\|\mathbf{g}_t\|_\star) . \tag{44}$$

Therefore, denoting $J_\varepsilon(t) = \{j \in [p] : \frac{|\mathbf{g}_t[j]|}{\|\mathbf{g}_t\|_\star} > \varepsilon\}$ we have:

$$\forall j \in J_\varepsilon(t) : \left|\frac{\mathbf{m}_t[j]}{\mathbf{g}_t[j]} - 1\right| = \frac{|\mathbf{m}_t[j] - \mathbf{g}_t[j]|}{|\mathbf{g}_t[j]|} \leq \frac{\mathrm{o}(\|\mathbf{g}_t\|_\star)}{\varepsilon\|\mathbf{g}_t\|_\star} \xrightarrow{t\to\infty} 0 ,$$

which gives item 1 for $\mathbf{m}_t$. We repeat the argument with $\mathbf{v}_t$ and $\mathbf{g}_t^2$. Whenever $i \in [m], \bar{i} \notin I_j$ we have $\mathbf{g}_t[i,j]\mathbf{g}_t[\bar{i},j] = \mathrm{o}\left(\|\mathbf{g}_t\|_\star^2\right)$ and $\mathbf{v}_t[i,\bar{i},j] = \mathrm{o}\left(\|\mathbf{g}_t\|_\star^2\right)$ by applying Lemma B.5 2(c) (with $g = G^2, F = \mathbf{g}_t[i,j]\mathbf{g}_t[\bar{i},j]$), so

$$\forall i \in [m], \bar{i} \notin I_j : |\mathbf{v}_t[i,\bar{i},j] - \mathbf{g}_t[i,j]\mathbf{g}_t[\bar{i},j]| \leq \mathrm{o}\left(\|\mathbf{g}_t\|_\star^2\right) + \mathrm{o}\left(\|\mathbf{g}_t\|_\star^2\right) ,$$

and by Equation (43)

$$\forall i, i' \in I_j : |\mathbf{v}_t[i,i',j] - \mathbf{g}_t[i,j]\mathbf{g}_t[i',j]| = \mathrm{o}(\mathbf{g}_t[i,j]\mathbf{g}_t[i',j]) = \mathrm{o}\left(\|\mathbf{g}_t\|_\star^2\right) ,$$

so

$$\forall j \in [p] : \left|\mathbf{v}_t[j] - \mathbf{g}_t^2[j]\right| \leq \mathrm{o}\left(\|\mathbf{g}_t\|_\star^2\right) . \tag{45}$$

Hence,

$$\forall j \in J_\varepsilon(t) : \left|\frac{\mathbf{v}_t[j]}{\mathbf{g}_t^2[j]} - 1\right| = \frac{\left|\mathbf{v}_t[j] - \mathbf{g}_t^2[j]\right|}{|\mathbf{g}_t^2[j]|} \leq \frac{\mathrm{o}\left(\|\mathbf{g}_t\|_\star^2\right)}{\varepsilon^2\|\mathbf{g}_t\|_\star^2} \xrightarrow{t\to\infty} 0$$

This gives the result of item 1 for $\mathbf{v}_t$.

Notice that from norm equivalence, $\|\mathbf{g}_t\|_\star^2 \leq \mathcal{O}\left(\|\mathbf{g}_t\|_\infty^2\right) = \mathcal{O}(\|\mathbf{g}_t^2\|_\infty) \leq \mathcal{O}(\|\mathbf{g}_t^2\|_\star)$. Therefore Equations (44), (45) imply that there exists a vanishing $e(t) \xrightarrow{t\to\infty} 0$ with

$$\|\mathbf{m}_t - \mathbf{g}_t\|_\star \leq e(t)\|\mathbf{g}_t\|_\star, \quad \|\mathbf{v}_t - \mathbf{g}_t^2\|_\star \leq e(t)\|\mathbf{g}_t^2\|_\star .$$

Thus item 2 follows from Lemma C.20 (note that $\mathbf{g}_t \neq 0$ since $\|\mathbf{g}_t\|_\star \geq \Omega(\mathcal{L}\|\boldsymbol{\theta}_t\|^{L-1}) > 0$ by Assumptions (T1),(T2) and Lemma C.2, and therefore $\mathbf{m}_t, \mathbf{v}_t \neq 0$ by Equations (44), (45)).

$\square$

**Theorem C.21** (KKT Stationarity for Momentum SD - Implies Theorem 3.2). *Assume $\boldsymbol{\theta}_t$ follows a trajectory of normalized or unnormalized momentum steepest descent under Assumptions (M1-Weak), (M2), (LR-MSD), (T1), (T2), (T3). Then $\bar{\boldsymbol{\theta}} = \lim_{t\to\infty} \frac{\boldsymbol{\theta}_t}{\|\boldsymbol{\theta}_t\|}$ is along the direction of a KKT point of the $\|\cdot\|$-max-margin problem (Equation (11)).*

*Proof.* We claim that in both the normalized and unnormalized cases it holds that $\left\|\frac{d\boldsymbol{\theta}_t}{dt}\right\| \leq o\left(t^{\frac{1}{L}-1}\right)$. In the normalized case this is straightforward since $\left\|\frac{d\boldsymbol{\theta}_t}{dt}\right\| = \eta(t) \leq o\left(t^{\frac{1}{L}-1}\right)$. In the unnormalized case, by (T1),(T2), it holds that $\|\mathbf{g}_t\|_\star = \Theta(\|\boldsymbol{\theta}_t\|^{L-1}\mathcal{L})$ is bounded (Lemma C.2), so $\|\mathbf{m}_t\|_\star$ is bounded (Corollary B.6), implying that also in this case $\left\|\frac{d\boldsymbol{\theta}_t}{dt}\right\| = \eta(t)\|\mathbf{m}_t\|_\star \leq \mathcal{O}\left(\eta(t)\right) \leq o\left(t^{\frac{1}{L}-1}\right)$.

Note that for any $\alpha > 0$ it holds that

$$\arg\min_{\|\mathbf{u}\|=\alpha} \langle \mathbf{u}, \mathbf{m}_t \rangle = \arg\min_{\|\mathbf{u}\|=1} \langle \alpha\mathbf{u}, \mathbf{m}_t \rangle = \alpha \arg\min_{\|\mathbf{u}\|=1} \langle \mathbf{u}, \mathbf{m}_t \rangle ,$$

so in both the normalized and unnormalized cases, it holds that

$$\frac{\frac{d\boldsymbol{\theta}_t}{dt}}{\left\|\frac{d\boldsymbol{\theta}_t}{dt}\right\|} = \arg\min_{\|\mathbf{u}\|=1} \langle \mathbf{u}, \mathbf{m}_t \rangle .$$

Thus

$$\left\langle \frac{\frac{d\boldsymbol{\theta}_t}{dt}}{\left\|\frac{d\boldsymbol{\theta}_t}{dt}\right\|}, -\frac{\mathbf{g}_t}{\|\mathbf{g}_t\|_\star} \right\rangle = \left\langle \arg\min_{\|\mathbf{u}\|=1} \langle \mathbf{u}, \mathbf{m}_t \rangle, -\frac{\mathbf{g}_t}{\|\mathbf{g}_t\|_\star} \right\rangle = \left\langle \arg\min_{\|\mathbf{u}\|=1} \langle \mathbf{u}, \mathbf{m}_t \rangle, -\frac{\mathbf{m}_t}{\|\mathbf{m}_t\|_\star} + \frac{\mathbf{m}_t}{\|\mathbf{m}_t\|_\star} - \frac{\mathbf{g}_t}{\|\mathbf{g}_t\|_\star} \right\rangle$$

$$= 1 + \left\langle \arg\min_{\|\mathbf{u}\|=1} \langle \mathbf{u}, \mathbf{m}_t \rangle, \frac{\mathbf{m}_t}{\|\mathbf{m}_t\|_\star} - \frac{\mathbf{g}_t}{\|\mathbf{g}_t\|_\star} \right\rangle .$$

We claim that $\boldsymbol{\theta}_t$ is an approximate steepest descent trajectory (Definition C.12) with $\nu(t) = \left\|\frac{d\boldsymbol{\theta}_t}{dt}\right\|$ and $R_{max} \leq 1$. We have already seen $\|\mathbf{g}_t\|_\star \neq 0$. By Lemma C.13, it suffices to show that $\text{ess}\lim_{t\to\infty} \left\langle \frac{\frac{d\boldsymbol{\theta}_t}{dt}}{\left\|\frac{d\boldsymbol{\theta}_t}{dt}\right\|}, -\frac{\mathbf{g}_t}{\|\mathbf{g}_t\|_\star} \right\rangle = 1$. It therefore suffices to show that

$$\text{ess}\lim_{t\to\infty} \left\langle \arg\min_{\|\mathbf{u}\|=1} \langle \mathbf{u}, \mathbf{m}_t \rangle, \frac{\mathbf{m}_t}{\|\mathbf{m}_t\|_\star} - \frac{\mathbf{g}_t}{\|\mathbf{g}_t\|_\star} \right\rangle = 0 .$$

And since $\left\|\arg\min_{\|\mathbf{u}\|=1} \langle \mathbf{u}, \mathbf{m}_t \rangle\right\| = 1$, it suffices to show that

$$\frac{\mathbf{m}_t}{\|\mathbf{m}_t\|_\star} - \frac{\mathbf{g}_t}{\|\mathbf{g}_t\|_\star} \xrightarrow{t\to\infty} 0 .$$

And this is a result of Lemma C.19. Now the KKT result follows from Theorem C.17. $\qquad\square$

### C.6. Composite MSD Algorithms, Muon and Muon-Signum

In this section we prove a simple lemma showing that by partitioning the parameters of a model and training each part with a different normalized SD/MSD algorithm, the resulting algorithm is itself normalized SD/MSD with respect to the maximum among the norms. This allows formulating Muon (on multi-layer networks) and composite algorithms (such as running Muon on matrices and Signum on non-matrix parameters) as normalized momentum steepest descent algorithms.

**Lemma C.22** (Dual of Max Norm). *Let $\boldsymbol{\theta} = (\mathbf{u}_1, ..., \mathbf{u}_K)$ be a product representation of $\mathbb{R}^p$, with $\mathbf{u}_k \in \mathbb{R}^{p_k}$, $\sum_k p_k = p$. Let $\{\|\cdot\|_{(k)}\}_{k=1}^K$ be norms on $\mathbb{R}^{p_k}$ with dual norms $\{\|\cdot\|_{(k),\star}\}_{k=1}^K$. Denote $\|\boldsymbol{\theta}\| = \max_k \|\mathbf{u}_k\|$. Then $\|\cdot\|$ is a norm, and its dual norm is $\|\boldsymbol{\theta}\|_\star = \sum_k \|\mathbf{u}_k\|_{(k),\star}$.*

*Proof.* First, it is a standard result that $\|\cdot\|$ is a norm (one can easily verify that $\|\cdot\|$ satisfies positive definiteness, homogeneity

and the triangle inequality). Note that

$$\|\boldsymbol{\theta}\|_{\star} := \max\left\{\langle \mathbf{x}, \boldsymbol{\theta}\rangle \mid \|\mathbf{x}\| = 1\right\}$$

$$= \max\left\{\sum_k \langle \mathbf{x}_k, \mathbf{u}_k\rangle \mid \max_k \|\mathbf{x}_k\|_{(k)} = 1\right\}$$

$$= \max\left\{\sum_k \|\mathbf{x}_k\|_{(k)} \|\mathbf{u}_k\|_{(k),\star} \mid \max_k \|\mathbf{x}_k\|_{(k)} = 1\right\}$$

$$= \sum_k \|\mathbf{u}_k\|_{(k),\star} \ .$$

The third equality holds since for each $k$, by definition of the dual norm $\langle \mathbf{x}_k, \mathbf{u}_k\rangle$ is upper bounded by $\|\mathbf{x}_k\|_{(k)} \|\mathbf{u}_k\|_{(k),\star}$ and this bound is attained by choosing $\mathbf{x}_k$ in the direction that defines $\|\mathbf{u}_k\|_{(k),\star}$. The last equality holds by choosing $\forall k : \|\mathbf{x}_k\|_{(k)} = 1$. $\qquad\square$

**Lemma C.23** (Composing SD and MSD Algorithms). *Let $\boldsymbol{\theta}_t = (\mathbf{u}_1(t), ..., \mathbf{u}_K(t))$ be an arc of parameters of a model $f(\mathbf{x}_i; \boldsymbol{\theta}_t)$ with $\mathbf{u}_k \in \mathbb{R}^{p_k}$, $\sum_k p_k = p$. Denote a choice of subgradients $\mathbf{g}_t = (\mathbf{g}_1(t), ..., \mathbf{g}_K(t)) \in \partial \mathcal{L}(\boldsymbol{\theta}_t)$, and momentum estimates $\mathbf{m}_t = (\mathbf{m}_1(t), ..., \mathbf{m}_K(t))$. Let $\{\|\cdot\|_{(k)}\}_{k=1}^K$ be norms on $\mathbb{R}^{p_k}$ with dual norms $\{\|\cdot\|_{(k),\star}\}_{k=1}^K$. Assume there exists $\eta(t) > 0$ so that for all $k \in [K]$ one of the following hold:*

1. *(Normalized SD) For almost any $t$,*

$$\frac{d\mathbf{u}_k(t)}{dt} \in \eta(t) \arg\min_{\|\mathbf{u}\|_{(k)} = 1} \langle \mathbf{u}, \mathbf{g}_k(t)\rangle \ .$$

2. *(Normalized MSD) For almost any $t$,*

$$\frac{d\mathbf{u}_k(t)}{dt} \in \eta(t) \arg\min_{\|\mathbf{u}\|_{(k)} = 1} \langle \mathbf{u}, \mathbf{m}_k(t)\rangle \ .$$

*Then, $\boldsymbol{\theta}_t$ is a trajectory of normalized SD / normalized MSD respectively, with respect to the norm $\|\boldsymbol{\theta}\| = \max_k \|\mathbf{u}_k\|$.*

*Proof.* By definition of the dual norm it holds that

$$\min_{\|\mathbf{u}\|_{(k)} = 1} \langle \mathbf{u}, \mathbf{g}_k(t)\rangle = -\|\mathbf{g}_k(t)\|_{(k),\star}, \qquad \min_{\|\mathbf{u}\|_{(k)} = 1} \langle \mathbf{u}, \mathbf{m}_k(t)\rangle = -\|\mathbf{m}_k(t)\|_{(k),\star} \ .$$

Therefore in the case of SD, for almost any $t$,

$$\left\langle \frac{1}{\eta(t)} \frac{d\boldsymbol{\theta}_t}{dt}, \mathbf{g}_t\right\rangle = \sum_k \left\langle \frac{1}{\eta(t)} \frac{d\mathbf{u}_k(t)}{dt}, \mathbf{g}_k(t)\right\rangle = -\sum_k \|\mathbf{g}_k(t)\|_{(k),\star} \stackrel{\text{Lemma C.22}}{=} -\|\mathbf{g}_t\|_{\star} \ .$$

Similarly in the case of MSD,

$$\left\langle \frac{1}{\eta(t)} \frac{d\boldsymbol{\theta}_t}{dt}, \mathbf{m}_t\right\rangle = \sum_k \left\langle \frac{1}{\eta(t)} \frac{d\mathbf{u}_k(t)}{dt}, \mathbf{m}_k(t)\right\rangle = -\sum_k \|\mathbf{m}_k(t)\|_{(k),\star} \stackrel{\text{Lemma C.22}}{=} -\|\mathbf{m}_t\|_{\star} \ .$$

$\qquad\square$

The following is a more general lemma pertaining to approximate SD algorithms.

**Lemma C.24** (Composing Approximate SD Algorithms). *Let $\boldsymbol{\theta}_t = (\mathbf{u}_1(t), ..., \mathbf{u}_K(t))$ be an arc of parameters of a model $f(\mathbf{x}_i; \boldsymbol{\theta}_t)$ with $\mathbf{u}_k \in \mathbb{R}^{p_k}$, $\sum_k p_k = p$. Denote a choice of subgradients $\mathbf{g}_t = (\mathbf{g}_1(t), ..., \mathbf{g}_K(t)) \in \partial \mathcal{L}(\boldsymbol{\theta}_t)$, with eventually $\mathbf{g}_i(t) \neq \mathbf{0}$ for almost any $t$. Assume there exists $\nu(t) > 0, R_1, .., R_K > 0$ and norms $\{\|\cdot\|_{(k)}\}_{k=1}^K$ (with dual norms $\{\|\cdot\|_{(k),\star}\}_{k=1}^K$) so that:*

1. $\int_0^t \nu \stackrel{t\to\infty}{\Longrightarrow} \infty$

2. $\forall k \in [K] : \limsup_{t\to\infty} \frac{\|\mathbf{u}_k(t)\|_{(k)}}{\int_0^t \nu} \le R_k$

3. $\forall k \in [K] : \operatorname{ess\,liminf}_{t\to\infty} r_k(t) \ge 1$ where $r_k(t) \overset{a.e.}{=} \left\langle \frac{1}{\nu(t)} \frac{d\mathbf{u}_k(t)}{dt}, -\frac{\mathbf{g}_k(t)}{\|\mathbf{g}_k(t)\|_{(k),\star}} \right\rangle$

*Then $\boldsymbol{\theta}_t$ is a trajectory of Approximate Steepest Descent with respect to the norm $\|\cdot\| = \max_k \|\cdot\|_{(k)}$, $\nu(t)$ and $R_{max} \le \max_k R_k$.*

*Proof.* It clearly holds by definition that $\limsup_{t\to\infty} \frac{\|\boldsymbol{\theta}_t\|}{\int_0^t \nu} \le \max_k R_k = R_{\max}$. Therefore it remains to show $\operatorname{ess\,liminf}_{t\to\infty} r(t) \ge 1$. Indeed, with the same choice of subgradients $\mathbf{g}_t$ we have

$$r(t) = \left\langle \frac{1}{\nu(t)} \frac{d\boldsymbol{\theta}_t}{dt}, -\frac{\mathbf{g}_t}{\|\mathbf{g}_t\|_\star} \right\rangle = \frac{\sum_k \left\langle \frac{1}{\nu(t)} \frac{d\mathbf{u}_k(t)}{dt}, -\mathbf{g}_k(t) \right\rangle}{\sum_k \|\mathbf{g}_k(t)\|_{(k),\star}} .$$

Now note the following general claim: for any set of functions $q_1(t), ..., q_K(t)$ and $\widetilde{q}_1(t), ..., \widetilde{q}_K(t) > 0$ with $\forall k \in [K]$ : $\operatorname{ess\,liminf}_{t\to\infty} \frac{q_k}{\widetilde{q}_k} \ge 1$ it holds that $\operatorname{ess\,liminf}_{t\to\infty} \frac{\sum_k q_k}{\sum_k \widetilde{q}_k} \ge 1$. Indeed, for any $\varepsilon > 0$ it holds eventually for almost any $t$ that $q_k(t) \ge \widetilde{q}_k(t)(1-\varepsilon)$ for all $k$, so $\sum_k q_k(t) \ge \sum_k \widetilde{q}_k(t)(1-\varepsilon) = (1-\varepsilon)\sum_k \widetilde{q}_k(t)$.

Applying the claim with $q_k(t) = \left\langle \frac{1}{\nu(t)} \frac{d\mathbf{u}_k(t)}{dt}, -\mathbf{g}_k(t) \right\rangle$ and $\widetilde{q}_k(t) = \|\mathbf{g}_k(t)\|_{(k),\star}$ finishes. $\qquad\square$

Below is our definition of Muon in the exact orthogonalization setting:

**Definition C.25** (Spectral and Nuclear Norms). For a real-valued matrix $W$ let $W = U\Sigma V^T$ the SVD of $W$, for $\Sigma = \operatorname{diag}(\sigma_1, ..., \sigma_r)$, $\sigma_i > 0$. We denote the spectral and nuclear norms of $W$ respectively as

$$\|W\|_{\mathrm{sp}} = \max_{i\in[r]} \sigma_i, \quad \|W\|_{\mathrm{nuc}} = \sum_{i\in[r]} \sigma_i .$$

For a collection of matrices $\mathbf{W} = (W_1, ..., W_K)$ we denote (msp short for max-sp, snuc short for sum-nuc)

$$\|\mathbf{W}\|_{\mathrm{msp}} = \max_{k\in[K]} \|W_k\|_{\mathrm{sp}}, \quad \|\mathbf{W}\|_{\mathrm{snuc}} = \sum_{k\in[K]} \|W_k\|_{\mathrm{nuc}} .$$

It is a standard fact that $\|\cdot\|_{\mathrm{nuc}}$ is the dual norm of $\|\cdot\|_{\mathrm{sp}}$. By Lemma C.22 this implies that $\|\cdot\|_{\mathrm{snuc}}$ is the dual norm of $\|\cdot\|_{\mathrm{msp}}$.

**Definition C.26** (Muon). Let $\boldsymbol{\theta}_t = (W_1(t), ..., W_K(t)) \in \mathbb{R}^p$ be a trajectory representing a collection of matrices, $\mathbf{g}_t = (G_1(t), ..., G_K(t)) \in \partial\mathcal{L}(\boldsymbol{\theta}_t)$ a choice of subgradients and $\mathbf{m}_t = (M_1(t), ..., M_K(t))$ a momentum estimate of $\mathbf{g}_t$ with parameter $c_1 > 0$. The Muon update is defined as $\frac{d\boldsymbol{\theta}_t}{dt} = (\frac{dW_1}{dt}, ..., \frac{dW_K}{dt})$ for:

$$\forall k \in [K] : \frac{dW_k}{dt} \in \left\{ -\eta(t) \cdot U_k(t)V_k^T(t) \mid M_k(t) \overset{\mathrm{SVD}}{=} U_k(t)\Sigma_k(t)V_k^T(t) \right\} ,$$

where SVD is the reduced SVD operator, i.e. $\Sigma_k(t)$ is a square diagonal matrix with strictly positive diagonal entries and $U_k(t), V_k(t)$ have orthonormal columns.

**Lemma C.27.** *Muon is a normalized momentum steepest descent algorithm with $\|\cdot\| = \|\cdot\|_{\mathrm{msp}}$.*

*Proof.* By the definition of normalized momentum steepest descent, we need to prove that

$$\left\| \frac{1}{\eta(t)} \frac{d\boldsymbol{\theta}_t}{dt} \right\|_{\mathrm{msp}} = 1, \quad \left\langle \frac{1}{\eta(t)} \frac{d\boldsymbol{\theta}_t}{dt}, \mathbf{m}_t \right\rangle = -\|\mathbf{m}_t\|_{\mathrm{snuc}} .$$

By definition of these norms, it suffices to show for every $k$ that (omitting $t$ for brevity) $\|U_k V_k^\top\|_{\mathrm{sp}} = 1$ and $\langle U_k V_k^\top, M_k \rangle = \|M_k\|_{\mathrm{nuc}}$. Indeed, $\|U_k V_k^\top\|_{\mathrm{sp}} = 1$ by definition ($U_k V_k^\top$ is an SVD of itself with singular values in $\{1, 0\}$). Denote $\Sigma_k = \operatorname{diag}(\sigma_1, ..., \sigma_r)$ then we must show

$$\langle U_k V_k^\top, M_k \rangle = \sum_{i=1}^r \sigma_i .$$

And indeed, since the elementwise dot product between two matrices $X, Y$ is equal to $\text{Tr}[X^\top Y]$, using $U_k^\top U_k = I$, $V_k^\top V_k = I$ we get

$$\left\langle U_k V_k^\top, M_k \right\rangle = \text{Tr}[M_k^\top U_k V_k^\top] = \text{Tr}[V_k \Sigma_k^\top U_k^\top U_k V_k^\top] = \text{Tr}[V_k \Sigma_k^\top V_k^\top] = \text{Tr}[V_k^\top V_k \Sigma_k^\top] = \text{Tr}[\Sigma_k^\top] = \sum_{i=1}^{r} \sigma_i$$

$\square$

**Corollary C.28** (KKT Stationarity for Muon - Implies Corollary 3.3). *Assume $\boldsymbol{\theta} = (W_1, ..., W_K) \in \mathbb{R}^p$ is a parameter vector representing a collection of matrices, following a trajectory of Muon with a shared learning rate schedule $\eta(t)$ under Assumptions (M1-Weak), (M2), (LR-MSD), (T1), (T2), (T3). Then $\bar{\boldsymbol{\theta}} = \lim_{t\to\infty} \frac{\boldsymbol{\theta}_t}{\|\boldsymbol{\theta}_t\|}$ is along the direction of a KKT point of the $\|\cdot\|$-max-margin problem (Equation (11)), with $\|\cdot\| = \|\cdot\|_{\text{msp}}$.*

*Proof.* Follows from Theorem C.21 since $\boldsymbol{\theta}_t$ is a trajectory of normalized momentum steepest descent with $\|\cdot\| = \|\cdot\|_{\text{msp}}$ (by Lemmas C.23, C.27). $\square$

Notice that Lemma C.23 reveals that any collection of momentum steepest descent algorithms may be run in parallel, resulting in a new momentum steepest descent algorithm with respect to the maximal norm. In particular, Muon-Signum, i.e. running Muon on weight matrices and Signum on non-matrix parameters, answers this definition.

**Corollary C.29** (KKT Stationarity for Muon-Signum - Implies Corollary 3.4). *Assume $\boldsymbol{\theta} = (W_1, ..., W_K, \mathbf{u}) = (\mathbf{W}, \mathbf{u}) \in \mathbb{R}^p$ is a parameter vector representing a collection of matrices and additional parameters $\mathbf{u}$. Assume $W_1, ..., W_K$ follow a trajectory of Muon and $\mathbf{u}$ follows a trajectory of Signum, with a shared scheduled learning rate $\eta(t)$. Assume (M1-Weak), (M2), (LR-MSD), (T1), (T2), (T3). Then $\bar{\boldsymbol{\theta}} = \lim_{t\to\infty} \frac{\boldsymbol{\theta}_t}{\|\boldsymbol{\theta}_t\|}$ is along the direction of a KKT point of the $\|\cdot\|$-max-margin problem (Equation (11)), with $\|\boldsymbol{\theta}\| = \max\{\|\mathbf{W}\|_{\text{msp}}, \|\mathbf{u}\|_\infty\}$.*

*Proof.* Follows from Theorem C.21 since $\boldsymbol{\theta}_t$ is a trajectory of normalized momentum steepest descent with $\|\boldsymbol{\theta}\| = \max\{\|\mathbf{W}\|_{\text{msp}}, \|\mathbf{u}\|_\infty\}$ (by Lemmas C.23, C.27). $\square$

## C.7. Adam

In this section we show that Adam is an approximate steepest descent algorithm with $\|\cdot\| = \|\cdot\|_\infty$ in the regime of a non-increasing learning rate $\eta(t) \leq o\left(t^{\frac{1}{L}-1}\right)$, $\int_0^\infty \eta = \infty$ (i.e. (LR-Adam)). This allows us to infer using Theorem C.17 that the assumed directional limit point $\bar{\boldsymbol{\theta}} = \lim_{t\to\infty} \frac{\boldsymbol{\theta}_t}{\|\boldsymbol{\theta}_t\|}$ of the trajectory is the direction of a KKT point of the $\ell_\infty$-max-margin problem. Recall again that (T3) is implied by (T2) if strengthening (M1-Weak) to (M1).

**Theorem C.30** (KKT Stationarity for Adam - Implies Theorem 3.5). *Let $\boldsymbol{\theta}_t$ be a trajectory of Adam with $c_1 \geq c_2$ (Equation (8)), under Assumptions (M1-Weak), (M2), (LR-Adam), (T1), (T2), (T3). Then the limit point $\bar{\boldsymbol{\theta}}$ of $\frac{\boldsymbol{\theta}_t}{\|\boldsymbol{\theta}_t\|}$ is the direction of a KKT point of Problem (11) with $\|\cdot\| = \|\cdot\|_\infty$.*

*Proof.* First, by assumptions (T1), (T2), it holds (Lemma C.2) that $\|\mathbf{g}_t\|_1 = \Theta(\|\boldsymbol{\theta}_t\|^{L-1} \mathcal{L})$ and in particular eventually $\mathbf{g}_t \neq \mathbf{0}$. By Lemma B.4, denote $C = C(c_1, c_2) > 0$ with $\forall t : \left\|\frac{\mathbf{m}_t}{\sqrt{\mathbf{v}_t}}\right\|_\infty \leq C$. This and (LR-Adam) imply that $\left\|\frac{d\boldsymbol{\theta}_t}{dt}\right\| \leq o\left(t^{\frac{1}{L}-1}\right)$. Therefore by Lemma C.19 with $\|\cdot\| = \|\cdot\|_\infty$, fixing $\varepsilon > 0$, there exist $e_j(t), e_j'(t) \overset{t\to\infty}{\longrightarrow} 0$ with:

$$\forall j \in J_\varepsilon(t): \qquad \mathbf{m}_t[j] = \mathbf{g}_t[j](1 + e_j(t)), \quad \sqrt{\mathbf{v}_t[j]} = |\mathbf{g}_t[j]|\left(1 + e_j'(t)\right).$$

Recalling $\hat{\mathbf{m}}_t = (1 - e^{-c_1 t})^{-1}\mathbf{m}_t$, $\hat{\mathbf{v}}_t = (1 - e^{-c_2 t})^{-1}\mathbf{v}_t$, since the bias correction terms themselves tend to 1 there exist $e_j(t) \overset{t\to\infty}{\longrightarrow} 0$ with (reusing the notation $e_j$)

$$\forall j \in J_\varepsilon(t): \frac{\hat{\mathbf{m}}_t[j]}{\sqrt{\hat{\mathbf{v}}_t[j]}} = \text{sign}\left(\mathbf{g}_t[j]\right) \cdot (1 + e_j(t)).$$

Again since $\frac{\sqrt{1-e^{-c_2 t}}}{1-e^{-c_1 t}} \to 1$ there exists $t_1 \geq 0$ with $\forall t \geq t_1 : 2C \geq \left\|\frac{\hat{\mathbf{m}}_t}{\sqrt{\hat{\mathbf{v}}_t}}\right\|_\infty$. Denote $\hat{C} := \max\left\{2C, \sup_{t\in(0,t_1]} \left\|\frac{\hat{\mathbf{m}}_t}{\sqrt{\hat{\mathbf{v}}_t}}\right\|_\infty\right\}$ an upper bound on $\left\|\frac{\hat{\mathbf{m}}_t}{\sqrt{\hat{\mathbf{v}}_t}}\right\|_\infty$ for all $t > 0$ (the supremum is finite by Lemma B.9).

For any $j \notin J_\varepsilon(t)$ it holds that $\left| \frac{\hat{\mathbf{m}}_t[j]}{\sqrt{\hat{\mathbf{v}}_t[j]}} \frac{\mathbf{g}_t[j]}{\|\mathbf{g}_t\|_1} \right| \leq \varepsilon \hat{C}$. Also, $\sum_{j \in J_\varepsilon(t)} |\mathbf{g}_t[j]| \geq \sum_{j \in [p]} |\mathbf{g}_t[j]| - \varepsilon p \|\mathbf{g}_t\|_1 = \|\mathbf{g}_t\|_1 (1 - \varepsilon p)$.

Therefore,

$$\left\langle \frac{\hat{\mathbf{m}}_t}{\sqrt{\hat{\mathbf{v}}_t}}, \frac{\mathbf{g}_t}{\|\mathbf{g}_t\|_1} \right\rangle \geq \sum_{j \in J_\varepsilon(t)} \frac{\hat{\mathbf{m}}_t[j]}{\sqrt{\hat{\mathbf{v}}_t[j]}} \frac{\mathbf{g}_t[j]}{\|\mathbf{g}_t\|_1} - \varepsilon \hat{C} p = \sum_{j \in J_\varepsilon(t)} (1 + e_j(t)) \frac{\text{sign}(\mathbf{g}_t[j]) \mathbf{g}_t[j]}{\|\mathbf{g}_t\|_1} - \varepsilon \hat{C} p$$

$$\geq \min_j (1 + e_j(t)) \cdot \frac{\sum_{j \in J_\varepsilon(t)} |\mathbf{g}_t[j]|}{\|\mathbf{g}_t\|_1} - \varepsilon \hat{C} p$$

$$\geq \min_j (1 + e_j(t)) \cdot \frac{\|\mathbf{g}_t\|_1 (1 - \varepsilon p)}{\|\mathbf{g}_t\|_1} - \varepsilon \hat{C} p$$

$$\xrightarrow{t \to \infty} 1 - \varepsilon p - \varepsilon \hat{C} p .$$

Since this holds for any $\varepsilon > 0$, this implies

$$\liminf_{t \to \infty} \left\langle \frac{\hat{\mathbf{m}}_t}{\sqrt{\hat{\mathbf{v}}_t}}, \frac{\mathbf{g}_t}{\|\mathbf{g}_t\|_1} \right\rangle \geq 1 .$$

Therefore,

$$\lim \inf_{t \to \infty} \left\langle \frac{1}{\eta(t)} \frac{d\boldsymbol{\theta}_t}{dt}, -\frac{\mathbf{g}_t}{\|\mathbf{g}_t\|_1} \right\rangle \geq 1 .$$

Also, since $\|\boldsymbol{\theta}_t\|_\infty \leq \int_0^t \eta(s) \left\| \frac{\hat{\mathbf{m}}_s}{\sqrt{\hat{\mathbf{v}}_s}} \right\|_\infty ds \leq \hat{C} \int_0^t \eta(s) ds$ it holds that

$$\limsup_{t \to \infty} \frac{\|\boldsymbol{\theta}_t\|_\infty}{\int_0^t \eta} \leq \hat{C} .$$

Therefore, by Lemma C.15, $\mathcal{L}(\boldsymbol{\theta}_t) \to 0, \|\boldsymbol{\theta}_t\| \to \infty$. This in turn implies that $\forall j \in [p] : |\mathbf{g}_t[j]| \xrightarrow{t \to \infty} 0$ (by Equation (25), noting that $\|\boldsymbol{\theta}_t\|^{L-1} = o(\mathcal{L}^{-1})$), so by Lemma B.10 it follows that in fact

$$\lim \sup_{t \to \infty} \frac{\|\boldsymbol{\theta}_t\|_\infty}{\int_0^t \eta} \leq 1 .$$

Therefore by Theorem C.17 with $\nu(t) = \eta(t)$ we get the result. $\qquad \square$

## C.8. Muon-Adam

Muon-Adam differs from Muon-Signum in that it does not adhere to the definition of normalized momentum steepest descent (due to the presence of Adam). Therefore the treatment of Muon-Adam relies directly on the framework of Approximate Steepest Descent, drawing main techniques from both Theorem C.30 and Theorem C.21. In addition to those, the key technical step is dividing $[0, \infty)$ into times $t$ when the dual norm has a significant contribution from the $\|\cdot\|_1$ norm of the gradient w.r.t $\mathbf{u}$, and times when there is a significant contribution from the $\|\cdot\|_{\text{snuc}}$ norm of the gradient w.r.t $\mathbf{W}$ (these subsets may overlap).

**Theorem C.31** (KKT Stationarity for Muon-Adam - Implies Theorem 3.6). *Assume $\boldsymbol{\theta} = (W_1, ..., W_K, \mathbf{u}) = (\mathbf{W}, \mathbf{u}) \in \mathbb{R}^p$ is a parameter vector representing a collection of matrices and additional parameters $\mathbf{u}$. Assume $W_1, ..., W_K$ follow a trajectory of Muon and $\mathbf{u}$ follows a trajectory of Adam, with respective learning rates of the form $\eta_0^M \eta(t), \eta_0^A \eta(t)$ for $\eta_0^M, \eta_0^A > 0$ and momentum parameters $c_M$ for Muon and $c_1 \geq c_2$ for Adam. Assume (M1-Weak), (M2), (LR-Adam), (T1), (T2), (T3), (A1). Then $\bar{\boldsymbol{\theta}} = \lim_{t \to \infty} \frac{\boldsymbol{\theta}_t}{\|\boldsymbol{\theta}_t\|}$ is along the direction of a KKT point of the $\|\cdot\|$-max-margin problem (Equation (11)), with*

$$\|\boldsymbol{\theta}\| = \max \left\{ \frac{\eta_0^A}{\eta_0^M} \|\mathbf{W}\|_{\text{msp}}, \|\mathbf{u}\|_\infty \right\} .$$

*Proof.* Denote $\mathbf{g}_t = (\mathbf{G}_t, \mathbf{g}_t^{(u)}), \mathbf{G}_t = (G_1(t), ..., G_k(t))$ the choice of subgradients along the trajectory. Denote $\mathbf{M}_t, \mu_t, \omega_t$ the momentum estimates of $\mathbf{G}_t, \mathbf{g}_t^{(u)}, \mathbf{g}_t^{(u)^2}$ with parameters $c_M, c_1, c_2$ respectively, and $\widetilde{\mathbf{m}}_t$ the momentum estimate for the whole gradient vector $\mathbf{g}_t$ with $c_M$. We retain the notations $\mathbf{m}_t, \mathbf{v}_t$ for the momentum estimates of $\mathbf{g}_t, \mathbf{g}_t^2$ with $c_1, c_2$.

Denote $J_M \cup J_A = [p]$ a partition of $[p]$ into the indices optimized by Muon and by Adam respectively. Define $\alpha = \frac{\eta_0^A}{\eta_0^M}$, $\|\cdot\|_{\alpha \cdot \mathrm{msp}} = \alpha \|\cdot\|_{\mathrm{msp}}$, and its dual norm by $\|\cdot\|_{\frac{1}{\alpha} \cdot \mathrm{snuc}} = \frac{1}{\alpha} \|\cdot\|_{\mathrm{snuc}}$. We aim to show that $\boldsymbol{\theta}_t$ is a trajectory of Approximate Steepest Descent with respect to the norm $\|\boldsymbol{\theta}\| = \max\{\|\mathbf{W}\|_{\alpha \cdot \mathrm{msp}}, \|\mathbf{u}\|_\infty\}$, $\nu(t) = \eta_0^A \eta(t)$ and $R_{\max} \leq 1$. First, by assumptions (T1), (T2), it holds (Lemma C.2) that $\|\mathbf{g}_t\|_\star = \Theta(\|\boldsymbol{\theta}_t\|^{L-1} \mathcal{L})$ and in particular eventually $\mathbf{g}_t \neq \mathbf{0}$.

First note that we can write the update for Muon and Adam separately as

$$\frac{d\mathbf{W}_t}{dt} \in \eta_0^M \eta(t) \cdot \arg \min_{\|\mathbf{W}\|_{\mathrm{msp}}=1} \langle \mathbf{W}, \mathbf{M}_t \rangle, \quad \frac{d\mathbf{u}_t}{dt} = -\eta_0^A \eta(t) \frac{\hat{\mu}_t}{\sqrt{\hat{\omega}_t}} .$$

But since by definition $\forall \mathbf{W}, \|\mathbf{W}\|_{\alpha \cdot \mathrm{msp}} = \alpha \|\mathbf{W}\|_{\mathrm{msp}}$, we can also write

$$\frac{d\mathbf{W}_t}{dt} \in \eta_0^A \eta(t) \cdot \arg \min_{\|\mathbf{W}\|_{\mathrm{msp}}=\frac{\eta_0^M}{\eta_0^A}} \langle \mathbf{W}, \mathbf{M}_t \rangle = \eta_0^A \eta(t) \cdot \arg \min_{\|\mathbf{W}\|_{\alpha \cdot \mathrm{msp}}=1} \langle \mathbf{W}, \mathbf{M}_t \rangle .$$

So we consider the optimization of $\mathbf{W}_t$ as normalized momentum steepest descent with respect to $\|\cdot\|_{\alpha \cdot \mathrm{msp}}$ and learning rate $\nu(t) = \eta_0^A \eta(t)$. In particular this implies $\left\| \frac{1}{\nu(t)} \frac{d\mathbf{W}_t}{dt} \right\|_{\alpha \cdot \mathrm{msp}} = 1$.

We observe the quantity of interest for Definition C.12, with the choice of subgradients fixed to be the subgradients chosen by the momentum:

$$r(t) = \left\langle \frac{1}{\nu(t)} \frac{d\boldsymbol{\theta}_t}{dt}, -\frac{\mathbf{g}_t}{\|\mathbf{g}_t\|_\star} \right\rangle = \left\langle \frac{1}{\nu(t)} \frac{d\mathbf{u}_t}{dt}, \frac{-\mathbf{g}_t^{(u)}}{\|\mathbf{g}_t\|_\star} \right\rangle + \left\langle \frac{1}{\nu(t)} \frac{d\mathbf{W}_t}{dt}, -\frac{\mathbf{G}_t}{\|\mathbf{g}_t\|_\star} \right\rangle$$

$$= \frac{\left\langle \frac{\hat{\mu}_t}{\sqrt{\hat{\omega}_t}}, \mathbf{g}_t^{(u)} \right\rangle + \left\langle \frac{1}{\nu(t)} \frac{d\mathbf{W}_t}{dt}, -\mathbf{G}_t \right\rangle}{\|\mathbf{G}_t\|_{\frac{1}{\alpha} \cdot \mathrm{snuc}} + \left\| \mathbf{g}_t^{(u)} \right\|_1} . \tag{46}$$

Our main goal is showing that $\mathrm{ess} \liminf_{t \to \infty} r(t) \geq 1$.

Fix $\frac{1}{2p} > \varepsilon > 0$ and denote $J_{\varepsilon^2}(t) = \left\{ j \in [p] \mid \frac{|\mathbf{g}_t[j]|}{\|\mathbf{g}_t\|_\star} > \varepsilon^2 \right\}$. Also denote

$$T_{\varepsilon,M} = \left\{ t \geq 0 \mid \frac{\|\mathbf{G}_t\|_{\frac{1}{\alpha} \cdot \mathrm{snuc}}}{\|\mathbf{g}_t\|_\star} \geq \varepsilon p \right\}, \quad T_{\varepsilon,A} = \left\{ t \geq 0 \mid \frac{\left\| \mathbf{g}_t^{(u)} \right\|_1}{\|\mathbf{g}_t\|_\star} \geq \varepsilon p \right\} .$$

Since $\|\mathbf{g}_t\|_\star = \|\mathbf{G}_t\|_{\frac{1}{\alpha} \cdot \mathrm{snuc}} + \left\| \mathbf{g}_t^{(u)} \right\|_1$ (Lemma C.22) and $\varepsilon p < \frac{1}{2}$ it holds that $T_{\varepsilon,M} \cup T_{\varepsilon,A} = [0, \infty)$. By Lemma C.19 (applied separately with $c_M$ and with $c_1, c_2$) it holds that there exist vanishing error terms $e_j(t) \to 0$ with

$$\forall j \in J_{\varepsilon^2}(t) : \widetilde{\mathbf{m}}_t[j] = \mathbf{g}_t[j](1 + e_j(t)), \quad \frac{\hat{\mathbf{m}}_t[j]}{\sqrt{\hat{\mathbf{v}}_t[j]}} = \mathrm{sign}\left(\mathbf{g}_t[j]\right)(1 + e_j(t)) .$$

We first observe the Adam expression. Notice that by definition $\left\| \mathbf{g}_t^{(u)} \right\|_1 = \sum_{j \in J_A} |\mathbf{g}_t[j]|$, so it holds for $t \in T_{\varepsilon,A}$ that

$$\sum_{j \in J_{\varepsilon^2}(t) \cap J_A} |\mathbf{g}_t[j]| = \left\| \mathbf{g}_t^{(u)} \right\|_1 - \sum_{j \in J_A \setminus J_{\varepsilon^2}(t)} |\mathbf{g}_t[j]| \geq \left\| \mathbf{g}_t^{(u)} \right\|_1 - \varepsilon^2 p \|\mathbf{g}_t\|_\star \geq \left\| \mathbf{g}_t^{(u)} \right\|_1 - \varepsilon \left\| \mathbf{g}_t^{(u)} \right\|_1 = (1 - \varepsilon) \left\| \mathbf{g}_t^{(u)} \right\|_1 .$$

The first inequality from the definition of $J_{\varepsilon^2}(t)$ and the second from the definition of $T_{\varepsilon,A}$. Denoting $\hat{C} =$

$\sup_{t\in[0,\infty)}\left\|\frac{\hat{\mu}_t}{\sqrt{\hat{\omega}_t}}\right\|_\infty$ as in the proof of Theorem C.30,

$$\forall t\in T_{\varepsilon,A}:\left\langle\frac{1}{\nu(t)}\frac{d\mathbf{u}_t}{dt},\frac{-\mathbf{g}_t^{(u)}}{\|\mathbf{g}_t\|_\star}\right\rangle=\left\langle\frac{\hat{\mu}_t}{\sqrt{\hat{\omega}_t}},\frac{\mathbf{g}_t^{(u)}}{\left\|\mathbf{g}_t^{(u)}\right\|_1}\right\rangle$$

$$=\frac{1}{\left\|\mathbf{g}_t^{(u)}\right\|_1}\sum_{j\in J_A}\frac{\hat{\mathbf{m}}_t[j]}{\sqrt{\hat{\mathbf{v}}_t[j]}}\mathbf{g}_t[j]$$

$$\geq\frac{1}{\left\|\mathbf{g}_t^{(u)}\right\|_1}\sum_{j\in J_A\cap J_{\varepsilon^2}(t)}\frac{\hat{\mathbf{m}}_t[j]}{\sqrt{\hat{\mathbf{v}}_t[j]}}\mathbf{g}_t[j]-\varepsilon^2\hat{C}p$$

$$=\frac{1}{\left\|\mathbf{g}_t^{(u)}\right\|_1}\sum_{j\in J_A\cap J_{\varepsilon^2}(t)}(1+e_j(t))\,|\mathbf{g}_t[j]|-\varepsilon^2\hat{C}p$$

$$\geq\min_j(1+e_j(t))\frac{1}{\left\|\mathbf{g}_t^{(u)}\right\|_1}\left\|\mathbf{g}_t^{(u)}\right\|_1(1-\varepsilon)-\varepsilon^2\hat{C}p$$

$$\xrightarrow{t\to\infty}1-\mathcal{O}\left(\varepsilon\right).$$

Where $\mathcal{O}\left(\cdot\right)$ hides uniform constants independent of $t$ and $\varepsilon$. Also

$$\forall t\notin T_{\varepsilon,A}:\left|\left\langle\frac{\hat{\mu}_t}{\sqrt{\hat{\omega}_t}},\frac{\mathbf{g}_t^{(u)}}{\|\mathbf{g}_t\|_\star}\right\rangle\right|\leq\hat{C}\frac{\left\|\mathbf{g}_t^{(u)}\right\|_1}{\|\mathbf{g}_t\|_\star}\leq\hat{C}\varepsilon p=\mathcal{O}\left(\varepsilon\right).$$

Considering the Muon expression, by norm equivalence and by Lemma C.19 (and since $\mathbf{M}_t-\mathbf{G}_t$ is a subvector of $\tilde{\mathbf{m}}_t-\mathbf{g}_t$), it holds that

$$\|\mathbf{M}_t-\mathbf{G}_t\|_{\frac{1}{\alpha}\cdot\mathrm{snuc}}\leq\mathcal{O}\left(\|\mathbf{M}_t-\mathbf{G}_t\|_\infty\right)\leq\mathcal{O}\left(\|\tilde{\mathbf{m}}_t-\mathbf{g}_t\|_\infty\right)\leq\mathcal{O}\left(\|\tilde{\mathbf{m}}_t-\mathbf{g}_t\|_\star\right)\leq\mathrm{o}\left(\|\mathbf{g}_t\|_\star\right).$$

For $t\in T_{\varepsilon,M}$ it holds that $\|\mathbf{g}_t\|_\star\leq\frac{1}{\varepsilon p}\|\mathbf{G}(t)\|_{\frac{1}{\alpha}\cdot\mathrm{snuc}}$ so there exists a vanishing $e(t)\xrightarrow{t\to\infty}0$ with

$$\forall t\in T_{\varepsilon,M}:\|\mathbf{M}_t-\mathbf{G}_t\|_{\frac{1}{\alpha}\cdot\mathrm{snuc}}\leq e(t)\|\mathbf{G}_t\|_{\frac{1}{\alpha}\cdot\mathrm{snuc}}.$$

Note that in particular $\|\mathbf{g}_t\|_\star\leq\frac{1}{\varepsilon p}\|\mathbf{G}(t)\|_{\frac{1}{\alpha}\cdot\mathrm{snuc}}$ implies $\mathbf{G}_t\neq0$. Since $e(t)$ is vanishing this implies that $\mathbf{M}_t\neq0$ for all large enough $t\in T_{\varepsilon,M}$. Thus by Lemma C.20, for all large enough $t$,

$$\forall t\in T_{\varepsilon,M}:\left\|\frac{\mathbf{M}_t}{\|\mathbf{M}_t\|_{\frac{1}{\alpha}\cdot\mathrm{snuc}}}-\frac{\mathbf{G}_t}{\|\mathbf{G}_t\|_{\frac{1}{\alpha}\cdot\mathrm{snuc}}}\right\|_{\frac{1}{\alpha}\cdot\mathrm{snuc}}\leq2e(t).$$

For all $t$ large enough with $2e(t)\leq\varepsilon$, since $\left\|\frac{1}{\nu(t)}\frac{d\mathbf{W}_t}{dt}\right\|_{\alpha\cdot\mathrm{msp}}=1$,

$$\forall t\in T_{\varepsilon,M}:\left\langle\frac{1}{\nu(t)}\frac{d\mathbf{W}_t}{dt},-\frac{\mathbf{G}_t}{\|\mathbf{G}_t\|_{\frac{1}{\alpha}\cdot\mathrm{snuc}}}\right\rangle=1-\left\langle\frac{1}{\nu(t)}\frac{d\mathbf{W}_t}{dt},\frac{\mathbf{M}_t}{\|\mathbf{M}_t\|_{\frac{1}{\alpha}\cdot\mathrm{snuc}}}-\frac{\mathbf{G}_t}{\|\mathbf{G}_t\|_{\frac{1}{\alpha}\cdot\mathrm{snuc}}}\right\rangle\geq1-\varepsilon,$$

$$\forall t\notin T_{\varepsilon,M}:\left|\left\langle\frac{1}{\nu(t)}\frac{d\mathbf{W}_t}{dt},-\frac{\mathbf{G}_t}{\|\mathbf{g}_t\|_\star}\right\rangle\right|\leq\frac{\|\mathbf{G}_t\|_{\frac{1}{\alpha}\cdot\mathrm{snuc}}}{\|\mathbf{g}_t\|_\star}<\varepsilon p.$$

Now observe any such large enough $t$. From Equation (46),

$$r(t)=\left\langle\frac{1}{\nu(t)}\frac{d\boldsymbol{\theta}_t}{dt},-\frac{\mathbf{g}_t}{\|\mathbf{g}_t\|_\star}\right\rangle=\frac{\left\langle\frac{\hat{\mu}_t}{\sqrt{\hat{\omega}_t}},\mathbf{g}_t^{(u)}\right\rangle+\left\langle\frac{1}{\nu(t)}\frac{d\mathbf{W}_t}{dt},-\mathbf{G}_t\right\rangle}{\|\mathbf{G}_t\|_{\frac{1}{\alpha}\cdot\mathrm{snuc}}+\left\|\mathbf{g}_t^{(u)}\right\|_1}.$$

If $t \in T_{\varepsilon,M} \setminus T_{\varepsilon,A}$ then $\|\mathbf{G}_t\|_{\frac{1}{\alpha}\cdot\mathrm{snuc}} \geq (1-\varepsilon p)\|\mathbf{g}_t\|_\star = (1-\varepsilon p)\left(\|\mathbf{G}_t\|_{\frac{1}{\alpha}\cdot\mathrm{snuc}} + \left\|\mathbf{g}_t^{(u)}\right\|_1\right)$, so

$$r(t) \geq \frac{(1-\varepsilon)\|\mathbf{G}_t\|_{\frac{1}{\alpha}\cdot\mathrm{snuc}} - \mathcal{O}\left(\varepsilon\right)\|\mathbf{G}_t\|_{\frac{1}{\alpha}\cdot\mathrm{snuc}}}{\frac{1}{1-\varepsilon p}\|\mathbf{G}_t\|_{\frac{1}{\alpha}\cdot\mathrm{snuc}}} = 1 - \mathcal{O}\left(\varepsilon\right) .$$

And similarly if $t \in T_{\varepsilon,A} \setminus T_{\varepsilon,M}$. Also, if $t \in T_{\varepsilon,M} \cap T_{\varepsilon,A}$ then

$$r(t) \geq \frac{(1-\varepsilon)\|\mathbf{G}_t\|_{\frac{1}{\alpha}\cdot\mathrm{snuc}} + (1 - \mathcal{O}\left(\varepsilon\right))\left\|\mathbf{g}_t^{(u)}\right\|_1}{\|\mathbf{G}_t\|_{\frac{1}{\alpha}\cdot\mathrm{snuc}} + \left\|\mathbf{g}_t^{(u)}\right\|_1} \geq 1 - \mathcal{O}\left(\varepsilon\right) .$$

Altogether we have $r(t) \geq 1 - \mathcal{O}\left(\varepsilon\right)$ for almost all large enough $t$. Since $\varepsilon > 0$ was arbitrarily small, this implies

$$\operatorname{ess\,liminf}_{t\to\infty} r(t) \geq 1 .$$

Denote $R_{\max} = \limsup_{t\to\infty} \frac{\|\boldsymbol{\theta}_t\|}{\int_0^t \tilde{\eta}}$. As in the proof of Theorem C.30, we first show $R_{\max}$ is finite and then that $R_{\max} \leq 1$. Since $\|\boldsymbol{\theta}_t\| = \max\{\|\mathbf{W}\|_{\alpha\cdot\mathrm{msp}}, \|\mathbf{u}\|_\infty\}$ and

$$\limsup \frac{\|\mathbf{W}\|_{\alpha\cdot\mathrm{msp}}}{\int_0^t \nu} \leq 1, \quad \limsup \frac{\|\mathbf{u}\|_\infty}{\int_0^t \nu} \leq \hat{C} ,$$

It holds that $R_{\max} \leq \hat{C}$, so by Lemma C.15 it holds that $\mathcal{L}(\boldsymbol{\theta}_t) \stackrel{t\to\infty}{\longrightarrow} 0$, implying that $\|\mathbf{g}_t\|_\star \to 0$, so by Lemma B.10, it holds in fact that $\limsup \frac{\|\mathbf{u}\|_\infty}{\int_0^t \nu} \leq 1$, so $R_{\max} \leq 1$ as required. Therefore we get the result by Theorem C.17. $\qquad\square$

# D. Experimental Details

## D.1. 2-Layer Networks on MNIST

As mentioned in Section 6, we train two-layer (one hidden layer) homogeneous networks to classify $m = 2048$ MNIST digits (LeCun et al., 2002) as even or odd, using the logistic loss. We compare squared ReLU and ReLU, and the following optimizers: Normalized Gradient Descent (NGD) with and without momentum, Signum, Adam, Muon (treating the output layer as a matrix with a single row) and Muon-Adam. Training proceeds until the loss reaches a small target value ($10^{-8}$). The stability constant for Adam was chosen to be negligible with respect to gradient norm values ($\varepsilon = 10^{-20}$).

Adam momentum parameters are default ($\beta_1 = 0.9, \beta_2 = 0.999$) and momentum for other optimizers is also $\beta = 0.9$. Network parameters are initialized using Kaiming (He et al., 2015), with no corrections for non-linearities, and multiplied by a uniform factor $\alpha = 0.01$. Initial learning rate values $\eta_0$ are tuned per-setting to allow for gradual convergence to the target loss value, according to Table 1.

*Table 1.* Learning Rates for MNIST Task

| Activation | NGD w.o. momentum | NGD | Signum | Adam | Muon | Muon-Adam |
|---|---|---|---|---|---|---|
| ReLU | 2.0 | $8 \times 10^{-1}$ | $5 \times 10^{-3}$ | $1 \times 10^{-2}$ | $2 \times 10^{-1}$ | $5 \times 10^{-2}$ |
| Squared ReLU | 1.5 | $3 \times 10^{-1}$ | $3 \times 10^{-3}$ | $5 \times 10^{-3}$ | $8 \times 10^{-2}$ | $5 \times 10^{-2}$ |

Figure 2 presents results for Muon-Adam (the main results are shown in Figure 1). We observe that compared to Muon and Adam, Muon-Adam maximizes the norm $\max\{\|W\|_{\mathrm{msp}}, \|\mathbf{u}\|_\infty\}$ as expected, where the matrix $W$ is the first layer and the vector $\mathbf{u}$ is the output layer.

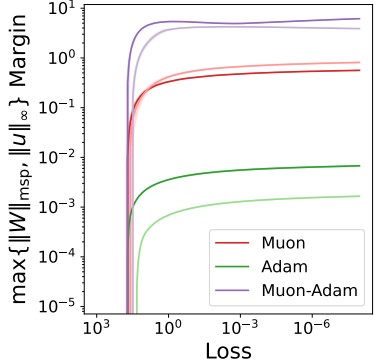

*Figure 2.* Margin values vs. loss for different optimizers. A lighter/darker color signifies the squared-ReLU / ReLU activations respectively. Lines are mean values over 10 random seeds, while filled areas represent one standard deviation.

## D.2. 4-Layer Networks on CIFAR10

Additionally, we extend the experimental setting to the CIFAR10 dataset, training 4-homogeneous ReLU networks with a single convolutional layer (16 output channels, $3 \times 3$ kernel with stride $=$ padding $= 1$), followed by ReLU and $2 \times 2$ max pooling, and 3 fully connected layers (2 hidden layers with ReLU activation) with hidden width 1024. The training task is classification of $m = 4096$ training points into two classes ("cat", "airplane"), with both classes represented equally and points chosen at random from the CIFAR10 dataset. Models were trained in full-batch with the logistic loss until a small loss value of $10^{-8}$ was reached. Initialization was done according to (He et al., 2015) with the ReLU correction, and weights later scaled by $\alpha = 0.1$. NGD with momentum, Signum, Adam, and Muon-Adam were compared, with the stability constant for Adam again chosen to be $\varepsilon = 10^{-20}$. The learning rate schedule was again chosen to align with theory $t^{-0.8} = \mathrm{o}\left(t^{1/4-1}\right) = \mathrm{o}\left(t^{-0.75}\right)$, with base learning rates shown in Table 2, chosen for convergence in comparable time between optimizers. Results shown in Figure 3a again corroborate the theory, with each optimizer achieving the highest margin value for its associated norm. Interestingly, as in the MNIST experiment, Signum achieves a larger $\ell_\infty$ margin than Adam.

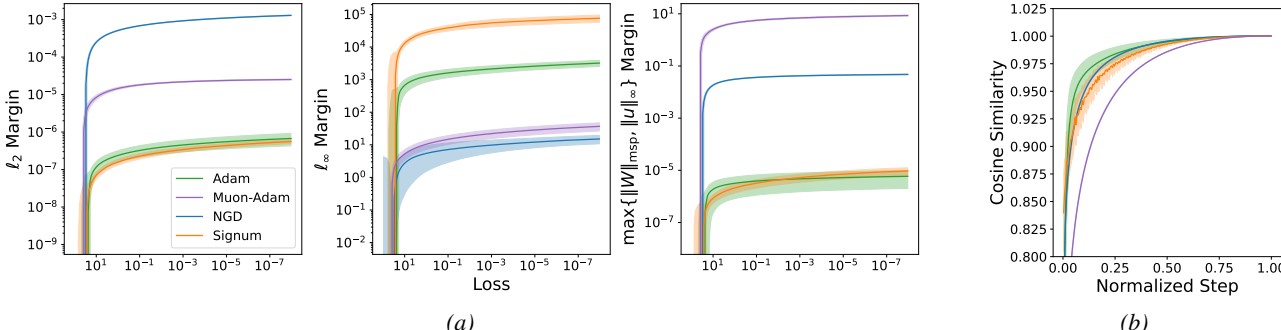

*(a)*                                                             *(b)*

*Figure 3.* (a) Margin values vs. loss for different optimizers. Lines are mean values over 10 random seeds, while filled areas represent one standard deviation. (b) Cosine similarity to last iterate $\left\langle \frac{\theta_t}{\|\theta_t\|_2}, \frac{\theta_{\text{last}}}{\|\theta_{\text{last}}\|_2} \right\rangle$, plotted on a normalized linear time scale.

*Table 2.* Learning Rates for CIFAR10 Task

| Activation | NGD | Signum | Adam | Muon-Adam |
|---|---|---|---|---|
| ReLU | $5 \times 10^{-1}$ | $3 \times 10^{-3}$ | $1 \times 10^{-2}$ | $3 \times 10^{-2}$ |

