# OpenReview forum: "The Implicit Bias of Adam and Muon on Smooth Homogeneous Neural Networks"
_ICML.cc/2026/Conference — ICML 2026 regular_

### Official Review · Reviewer_dKga · 2026-02-15

**Soundness:** 3
**Presentation:** 4
**Significance:** 3
**Originality:** 2
**Overall Recommendation:** 4
**Confidence:** 3

**Summary:**

The paper studies the concept of implicit bias in momentum-based optimization methods for binary classification with smooth homogeneous neural networks under log-concave, exponentially tailed losses. It analyzes the asymptotic behavior of normalized steepest descent, momentum steepest descent (including Muon and Signum), Adam (without the stability constant), and hybrid methods such as Muon-Adam. The authors extend prior analyses from linear models to smooth homogeneous nonlinear models and show that, under certain assumptions and decaying learning rates, these optimizers converge to KKT points of the corresponding margin maximization problems. The paper also introduces a general approximate steepest descent framework to unify the analysis and presents experiments on two-layer homogeneous neural networks trained on MNIST to validate the theoretical predictions.

**Compliance With Llm Reviewing Policy:**

Affirmed.

**Final Justification:**

The paper is overall solid, although it has some (non-fatal) weaknesses that somewhat weaken the contribution. I therefore maintain my score.

**Key Questions For Authors:**

1. The results rely on directional convergence (T2). How restrictive is this assumption? Are there conditions (beyond GD in o-minimal structures) under which such convergence might be provable for momentum-based methods?
2. How robust are the results when the assumptions are not fully satisfied in practice?
3. The analysis assumes decaying learning rates. Is it strictly necessary for the conclusions to hold?
4. The analysis is conducted in continuous time. How sensitive are the conclusions to discretization? Also, in the appendix, it is stated that "Initial learning rate values $\eta_0$ are tuned per-setting to allow for gradual convergence to the target loss value", which, as I understand it, means these learning rates are not necessarily those that yield the fastest convergence and that larger values could potentially be used. Do the same phenomena hold for discrete updates with larger, practically relevant step sizes?

**Limitations:**

Yes

**Strengths And Weaknesses:**

**Soundness**

Strengths:
- The theoretical results are accompanied by proofs.
- The theoretical framework behind the main results builds on prior implicit bias literature, so the assumptions are reasonable.
- Assumptions are explicitly stated and discussed.
- Unified treatment via Approximate Steepest Descent.

Weaknesses:
- Although most assumptions have been used in prior works, some of them can be considered quite strong. They are standard but are still nontrivial and not guaranteed for momentum-based methods. Having said that, the authors are transparent about the fact that certain assumptions may not hold and discuss these limitations.
- The decaying learning rate requirement ($\eta(t) \leq o(t^{1/(L-1)})$) may limit applicability to common practical schedules.
- The authors provide an extension to non-smooth models, but it is unclear if the additional assumptions required in this setting hold in practice. The authors themselves acknowledge this, noting that these assumptions are not satisfied in their own experiments.
- Experimental validation is limited (binary classification tasks on MNIST with shallow networks). However, I acknowledge that the paper is mainly theoretical in nature, and the experiments are not the main focus but are included for illustrative purposes.
- The claims in the experimental section that specific optimizers maximize specific margins are somewhat overstated. The experiments show that these optimizers achieve larger margins than the other algorithms tested, but they do not establish that the margins are actually maximized.

Overall, the work appears technically sound under its assumptions, but the practical validity of some of these assumptions remains open.

**Presentation**

The paper is well structured and easy to follow. Assumptions and theoretical results are presented and discussed clearly. The relation to prior literature is well discussed. I did not notice any major issues with the writing.

**Significance**

Strengths:
- Understanding implicit bias in modern optimizers is a practically relevant problem.
- Extending linear-model results to smooth homogeneous networks meaningfully advances theoretical understanding.
- The unifying viewpoint through approximate steepest descent may be reused in future research.

Weaknesses:
- The contribution can play a role within optimization theory for deep learning, but its immediate practical implications are somewhat specialized and not entirely clear.
- The scope is restricted to homogeneous models, binary classification, log-concave,
exponentially tailed loss and decaying learning rates, which narrows direct practical impact.

**Originality**
The paper builds upon the results of several prior works, e.g. Tsilivis et al. (2025) (who considered unnormalized steepest descent with a constant learning rate; here the authors consider normalized steepest descent trajectory with a learning rate schedule) and Zhang et al. (2024) (who focused on linear models; here the authors consider smooth homogeneous models). The particular setup studied in this paper does not appear to have been considered before.

Strengths:
- The work clearly distinguishes itself from prior analyses and explicitly states what is new and what was previously known.
- The formalization via Approximate Steepest Descent is new and may have broader utility.

Weaknesses:
- The results rely heavily on prior works, including proof techniques, so the contribution may be viewed as incremental.

---

> ### Author Rebuttal · Authors · 2026-03-31
>
> We thank the reviewer for their appreciation of the theoretical framework and results.
>
> Directional convergence assumption: answering the reviewer’s question, the assumption is important in our proofs. Please see our detailed answer to Reviewer ytxx, where we argue that this assumption is reasonable, common in the literature, and that implicit bias works are classically decoupled from works attempting to prove directional convergence. Establishing directional convergence under assumptions such as o-minimal structures is an interesting open question, but outside the scope of our work.
>
> Regarding the learning rate schedule $\eta(t) \leq o (t^{1/L-1})$, we agree that this does not cover all common schedules, but point out that the inverse harmonic schedule $\eta(t) = 1 / t$, common in analyses of optimizers (e.g. Shamir and Zhang (2012)), is valid for any $L > 1$. Also, we point out that many learning rate schedules are decaying in practice (e.g. Vaswani et al. (2017) uses $t^{-0.5}$). From a theoretical standpoint, the LR schedule is important; in practice, we ran preliminary experiments with a schedule violating our assumptions (but still decaying: $ t^{-0.5}$ for a 5-homogeneous model), and similar results to Figure 1 seem to hold. The initial learning rates in the experiments were indeed not chosen for fastest convergence; in theoretical settings it is common to focus on small learning rates since these often best express the theoretical properties of the optimizer.
>
> Non-smooth models: while it is true that our results hold mainly for smooth models, our empirical results suggest that the same qualitative results hold for non-smooth models. Indeed, it is not uncommon in the field of implicit bias that slight violations in assumptions in practice lead to approximate results which still contribute to our understanding of models (see discussion of homogeneity below). Also, we note that some important works in the field of implicit bias (e.g. Chizat and Bach (2020)) analyzed smooth models.
>
> Margin maximization: since the problem is non-convex, we agree that global margin maximization cannot be expected, and it is not a claim we make; rather, in the experiments we qualitatively expect the margin value to be larger than that of other optimizers, which is what we mean by e.g. “Adam maximizes the $\ell_\infty$ margin”. We accept the author’s comment that this is somewhat of a strong phrasing and will rephrase.
>
> We address additional concrete limitations mentioned by the reviewer - binary classification, log-concavity of the loss and homogeneity of the network. Indeed, our work is limited to binary classification, as are most leading works on implicit bias (Soudry et al. (2018), Lyu and Li (2019), Gunasekar (2018a;b), Nacson (2019a), to name a few). This is a minimal setting of interest which may be extended in future works. We kindly point out that, as mentioned in line 28 (right), that the logistic loss is a log-concave loss, and as this is the standard loss in practice for binary classification, we believe this is not much of a limitation. Regarding homogeneity, indeed practical models are often non-homogeneous, due to the presence of skip connections, bias terms and attention. However, some theoretical and practical works suggest that the deviation from homogeneity may not be large. In particular, Cai et al. (2025) extended implicit bias results on gradient descent to non-homogeneous models, showing that an approximate homogeneity relation holds for many architectures, such as ReLU networks with bias terms, VGG, and ResNet. From a practical viewpoint, non-homogeneous models were used in reconstruction and membership attacks based on KKT results proved for homogeneous models (Haim et al. (2022); Buzaglo et al. (2023); Oz et al. (2024);  Golbari et al. (2025)), and their success suggests an approximate homogeneity property occurs.
>
> Regarding reliance on previous works: we make a key technical contribution in the form of analysis of momentum-based algorithms. Lemma C.19 in particular combines tools proved on the momentum operator in Appendix B with a careful analysis of gradient magnitudes along the trajectory, to show an asymptotic relation between the momentum and the gradient. Additionally, we believe our introduction of approximate steepest descent (ASD) as a framework significantly generalizes previous works and constitutes a technical contribution. Finally, we also analyze composite algorithms, showing that often they are precise momentum steepest descent algorithms (e.g. Muon-Signum), while for Muon-Adam we demonstrate how one can overcome the non-trivial technical challenges of analyzing a composition of two ASD algorithms which are not MSD.
>
> Robustness of results and experimental validation: we ran preliminary experiments (CIFAR-10 with a convolutional model) which reproduce similar qualitative results, even when the model is non-smooth, and with a learning rate schedule violating our assumption.

---

> > ### Author Rebuttal · Reviewer_dKga · 2026-04-01
> >
> > Thank you for answering my questions. I maintain my view that the paper is overall solid, although it has some (non-fatal) weaknesses that somewhat weaken the contribution. I will therefore maintain my score.

---

### Official Review · Reviewer_A65r · 2026-03-04

**Soundness:** 3
**Presentation:** 4
**Significance:** 1
**Originality:** 1
**Overall Recommendation:** 3
**Confidence:** 3

**Summary:**

This paper studies the implicit bias of Adam and Muon optimizers on homogeneous networks. The analysis takes place in continuous time. The main contribution provides, under appropriate assumptions, directional convergence for the normalized steepest descent method, toward KKT points of the margin maximization problem for the underlying steepest descent norm. This extends the contribution of Tsilivis et al. (2025) that considered (unnormalized) steepest descent with a constant learning rate. The second main result extends the analysis to momentum methods, providing similar implicit bias characterization for momentum steepest descent, Adam, as well as combination of Muon and Adam, as implemented in practice. This second part relies on the analysis of a unifying mechanism: approximate steepest descent, for which a general result translate into corollaries for these algorithms. The paper is concluded with numerical experiments showing that as the loss decreases, the margin increases, and the algorithm achieving the highest margin correspond to theoretical predictions. The numerical experiments also suggest that the directional convergence assumption made for momentum methods holds in practice.

**Compliance With Llm Reviewing Policy:**

Affirmed.

**Final Justification:**

The rebuttal agree that Theorems 3.1 and 3.2 can be derived from existing work. The authors argue that "this is a small part of our paper". While this takes a small part of the appendix, it is presented as the first main contribution in the main text. The first half of the paper is dedicated to discussion on normalized versus un-normalized systems. Implementing the proposed reframing requires a large rewrite of the first half of the main text. This is the main reason why I wish to maintain my evaluation.

I agree with reviewer ytxx regarding assumption T2. I similarly still believe that this externalizes a core difficulty. I uynderstand the response of the authors. The authors cite ideas which were relevant 8 years ago. It does not make them equaly relevant today, and as mentioned by the authors, these assumptions have been proven as facts five years ago for the gradient algorithm. After thinking about the authors response, I understand that this is in line with existing literature. Although I think that this impairs the relevance of the study in a critical way, I do not want to penalize the authors too much and raised my score.

**Key Questions For Authors:**

Beyonds the main shortcomings mentioned above

"we consider Adam without the stability constant, as this setting more accurately reflects the behavior of Adam in practice". What does it mean? Do you have references for this? In practice there is a stability constant. This choice causes technical difficulties latter on, requiring the authors to make assumptions which are impossible to check a priori.

**Limitations:**

For Adams dynamics, please cite:
Barakat, A., & Bianchi, P. (2021). Convergence and dynamical behavior of the ADAM algorithm for nonconvex stochastic optimization. SIAM Journal on Optimization, 31(1), 244-274.
The first version of this paper is from 2018, more than five years before the cited references.

Remark on assumptions (M1)-(M2): for $L = 1$, the model is linear in theta. Indeed, in this case, the gradient is $0$ homogeneous and under the $C^1$ assumption, it needs to be constant around the origin, therefore constant. This makes sense mostly for $L>1$, in which case, the origin is a critical point of the loss. The authors mention it latter on, but this could be also commented around the assumptions.

Remark on (M2)-weak: $L<1$ is not compatible with local Lipschicity, the derivative needs to explode around the origin.

"Notably, networks with ReLU activations satisfy (M1-Weak) but not (M1)". Please provide a reference for this. The appendix mentions Davis et. al. 2020, which critically stands on previous works on the topic. The authors are very cautious to cite many recent ML papers related to, e.g., the continuous dynamical systems associated to ADAM, various norms used to devise algorithms, or various C1 activation functions. Why is there no credit to those who identified geometry as a key to handle nonsmoothness in applications (Bolte, Lewis ...)? This is needed to get (M1-weak), a much more critical aspect of the paper than the diversity of norms or activations recently considered. Yet, none of these works are mentioned. This double standard is a bit disapointing.


Learning rate assumption: I guess that $\eta$ needs to be non negative? Again, changing $\eta$ is equivalent to a reparameterization of time. The trajectory remains the same, it is just not traveled at the same speed. This is a specific feature of the continuous time analysis which does not really convey the practical challenges of learning rates in discrete time scenarios: lack of descent, divergence, oscillations, small progress ... This aspect of the paper is not really convincing in effectively representing learning rate issues.

The paper mentions Muon, but the algorithm is actually never explicitly described. It is not clear if the momentum dynamics fit the Muon optimizer.

Experiments: there is a very important discrepancy between the values of norms. It is not really possible to argue from the experiments that the margin is maximized, but rather that the margin increases and plateaus. This could be due to a different effect compared to margin maximization. The authors could reinforce their claims by:
- Comparing the obtained margins, with the margins that would be obtained by fixing the first layer weights to the last iterates and optimizing only for the last layer. This becomes a convex problem over fixed learned features.
- Compute approximate KKT measures. For example, when $f$ is $C^1$, for the $L_2$ norm, finding multipliers, or approximate multipliers is a QP. For other cases it should be possible to evaluate how close to feasibility is the KKT system.

Furthermore, MNIST is rather outdated. Does something similar happen on CIFAR10 for example?

Assumptions (R1) (T1) (T2) (A1) obviously do not hold for any trajectory, they are not global properties of the flow. They should be presented as trajectory, or initialization specific. The papers is limited to the set of initialization such that these hold. This needs to be properly presented.

Tsilivis et al. (2025), has been published somewhere?

**Strengths And Weaknesses:**

Understanding the implicit bias of numerical solvers used in practice is very relevant. The paper considers some of the most recent and widely used numerical solvers. The presentation is very well done, the arguments flow very well. The results sound very reasonable, the assumptions are clear. I did not check in details all the derivations in the appendix, but they seem reasonable and in line with existing literature on the topic.

Regarding practical relevance, this is mostly a theoretical paper, the main contributions being theorems characterizing the directional convergence of various continuous time dynamical systems.

The two main contributions, as announced in the introduction, are obliterated by two important shortcomings.

## Time reparameterization

If $f$ is $C^1$, the following two systems generate the same curves, for positive, nonsummable $\eta$, and real number $a$.
$$\dot{x} = - \nabla f(x), \qquad\qquad\qquad\qquad \dot{x} = - \eta(t) \frac{\nabla f(x)}{ \| \nabla f(x)\|^a}$$
This is easily seen by considering that 1/ solutions to these systems are unique on their domain of definition, and 2/ the arclength parameterization of the curves both result in the same system:
$$\frac{\dot{x}}{\|\dot{x}\|} = -\frac{\nabla f(x)}{ \| \nabla f(x)\|}$$
Qualitatively the trajectories are the same, the speed of traveling is different. In particular the limits are the same, and the directional limits also.

With this in mind, Theorems 3.1 and 3.2 only represent a very marginal extension of Tsilivis et al. (2025). Indeed, the main difference is the existence of the function $\varphi$ which is taken to be the identity in Tsilivis et al. (2025). This is a technical, incremental improvement. Otherwise, the discussions about normalized v.s. un-normalized or constant versus non constant $\eta$ do not matter so much: they are the same up to a time reparameterization. Therefore the asymptotics are the same. This considerably lowers the value of this part. In continuous time, these are relatively straightforward consequeces of Tsilivis et al. (2025): the effect of normalization or nonconstant $\eta$ do not modify the gradient curves and hence the directional limits are the same.



## Directional and margin convergence assumptions.

In order to deal with momentum methods, the authors make several assumptions about directional convergence, and margin convergence. This is super strong. Yet no such result exist in the literature. If directional convergence, or margin convergence, did not occur for Adam, the whole technical content related to this system, under this assumption, would be void. How would the authors convince the reader that their assumption do not make their result void? Is there a way to verify this assumption a priori for some models / losses?

This is very problematic: the title of the paper is on "The Implicit Bias of Adam", but the authors actually hard-code an implicit bias assumption for all momentum algorithm, the second main contribution of the paper. This is considerably undermines the relevance of this contribution.
For example, Theorem 3.3 says: if there is directional convergence, and margin convergence, it needs to be a KKT point. However, the hard part is to prove directional convergence and margin convergence or at least ensure that it holds true in some reasonable scenario.

Since the main contribution is theoretical, for both reasons, I evaluate the originality and significance of the contribution as very low.

---

> ### Author Rebuttal · Authors · 2026-03-31
>
> We thank the reviewer for acknowledging the relevance of the topic and the quality of presentation and argument flow.
>
> Regarding time reparameterization - we agree with the reviewer and thank them for this constructive feedback. Indeed, one can derive the core claims of Theorems 3.1 and 3.2 based on a reparameterization argument and Tsilivis et al. (2025), with no additional proofs required. We will reframe this section as a remark rather than a result.  We request the reviewer to keep in mind that this is a small part of our paper, (less than 3 pages of proofs out of ~30), the main contribution being the analysis of momentum-based algorithms. We acknowledge in the main text (line 320, right) that the techniques closely follow Tsilivis et al. (2025).
>
> Regarding directional convergence - please see our extensive answer to reviewer ytxx. In short we argue that this is a standard and reasonable assumption in the implicit bias literature. In light of all the literature mentioned there, we also dispute the notion that the “hard part” of implicit bias results is proving directional convergence; our proofs overcome non-trivial technical challenges and highlight insights such as the key role of gradient-parameter alignment.
>
> We emphasize that we do not assume margin convergence. Margin convergence follows from directional convergence, and our assumption is *positivity* of the limiting margin (which is always non-negative whenever the model interpolates the training points). As mentioned in line 194 (right), this too is not uncommon in foundational papers in the implicit bias literature, and strong empirical evidence is provided for it in Figures 1 and 2, where one can see that the margin is bounded away from 0 from very early on in the trajectory.
>
> Regarding presentation of assumptions, (R1) (T1) (T2) are explicitly introduced as *trajectory and realizability assumptions* (line 165, right), and (A1) is also clearly stated as an assumption, not as a universal property of the flow.
>
> Stability constant: analysis of Adam without the stability constant is standard for implicit bias results (see  Zhang et al. (2024); Fan et al. (2025); Baek et al. (2025); Xie and Li
> (2024)), as mentioned in line 133, right. Our work follows this convention and extends the viewpoint to homogenous models. The motivation for such a choice is that these analyses are asymptotic in nature, and gradients asymptotically vanish, making the stability constant dominate $\sqrt{v_t}$. Therefore including the constant leads to asymptotic behavior equivalent to momentum gradient-descent, whereas in practice most gradient values do not reach the order of the stability constant during optimization.
>
> We appreciate the perspective of the reviewer regarding the attribution of credit to previous works, including analyses of Adam and geometry-focused works. We will gladly attribute credit to additional works as the reviewer suggests.
>
> Model assumptions - we agree M2-weak adds no meaningful cases due to local Lipschitzness, and we will remove it. Also, indeed, when $L=1$, any $C^1$ model is linear in theta - this is a special case of our analysis but we will add a clarifying comment.
>
> ReLU activations satisfy M1-weak - ReLU networks are easily seen to be semialgebraic, therefore are Whitney $C^1$-stratifiable (e.g. Lojasiewicz. Ensemble semi-analytiques. IHES Lecture Notes, 1965).
>
> Regarding $\eta$ for momentum and Adam: $\eta$ is strictly positive, this is mentioned on line 114 (Section 2.2). We stress that with momentum, the choice of $\eta$ is *not* equivalent to time reparameterization, as the ODE defining the momentum term does not include $\eta$. We agree with the reviewer that continuous-time learning rate does not capture the full complexities of learning rate in discrete time, and we make no such claims in the paper. We will add a comment clarifying this.
>
> Muon: Muon is precisely defined in Definition C.26, whereafter we also prove that it is a special case of MSD. We will add a reference to this from the main text.
>
> Experiments: regarding norm discrepancies, please note that different margin values are *not* compared to one another. Rather, the same margin is compared across different optimizers, showing that each optimizer maximizes its respective margin more than other optimizers. Also, as mentioned by the reviewer, since the problem is non-convex, global margin maximization cannot be expected, and it is not a claim we make; rather, qualitatively we expect the margin value to be larger than that of other optimizers. The reviewer’s experiment suggestions are valuable: we ran preliminary experiments which show similar qualitative results for training a CNN on a two-class task from CIFAR10. We think further tuning the last layer weights would not definitively test our results, as KKT points are not necessarily local optima.
>
> Tsilivis et al. (2025) has been accepted to JMLR, to be published soon; an earlier version appeared in ICLR 2025.

---

> > ### Author Rebuttal · Reviewer_A65r · 2026-04-01
> >
> > The rebuttal agree that Theorems 3.1 and 3.2 can be derived from existing work. The authors argue that *"this is a small part of our paper"*. While this takes a small part of the appendix, it is presented as the first main contribution in the main text. The first half of the paper is dedicated to discussion on normalized versus un-normalized systems. Implementing the proposed reframing requires a large rewrite of the first half of the main text.
> >
> > I agree with reviewer ytxx regarding assumption T2. I similarly believe that this externalizes a core difficulty. I uynderstand the response of the authors. The authors cite ideas which were relevant 8 years ago. It does not make them equaly relevant today, and as mentioned by the authors, these assumptions have been proven as facts five years ago.
> >
> > For these reason, I maintain my evaluation of the manuscript.

---

### Official Review · Reviewer_uoZA · 2026-03-10

**Soundness:** 4
**Presentation:** 4
**Significance:** 3
**Originality:** 2
**Overall Recommendation:** 4
**Confidence:** 4

**Summary:**

This work characterizes the limit point of normalized steepest descent algorithms as satisfying the KKT conditions of a margin maximization problem. They consider the case where the algorithms use or do not use momentum, as well as different norms in parameter space. This study additionally generalizes previous results to the use of a decaying learning rate schedule. Finally, they perform experiments on $2$-layer ReLU squared networks to illustrate their theoretical results and verify that Muon, Adam, GD, and Signum are effectively maximizing the margin of their associated norms.

**Compliance With Llm Reviewing Policy:**

Affirmed.

**Final Justification:**

A strength of this work is technical riguor and presentation. I appreciate that the assumptions are clearly stated and the theoretical results are derived formally.

The main weakness regards originality and significance. I agree with other reviewers that the contribution is very close to prior work. The authors argued in the rebuttal that the momentum contribution and use of the approximate steepest descent framework is novel, which I agree with but remains limited.

My other technical or open questions have been adequately discussed by the authors.

Overall I recommend weak acceptance.

**Key Questions For Authors:**

1) A key assumption in this work seems to be the use of decaying learning rates. However, in practice, the learning rate often increases early in training during a short warm-up phase before being decayed. I was wondering if the authors had thought about how to circumvent this issue here and if the results could be generalized to learning rate schedules of this form?

2) At Line 217-218, the authors argue that their assumption with L=1 is weaker than $\eta_t = o(1)$ used in prior works. Could you briefly expand on why? It seems to me that replacing $L=1$ in the assumption recovers the same condition?

3) How do you think that your results could be expanded to long momentum, where the momentum parameter $\beta$ is not constant but goes to $1$ as $t\rightarrow \infty$, such as Nesterov with $\beta_t \approx 1-1/t$. This schedule is of theoretical interest and brings more important differences in the dynamics against a constant momentum scheme.

**Limitations:**

Yes

The authors highlight limitations of their work in the conclusion section, such as strong assumptions of their theoretical results and remaining open questions such as extending some results to non-smooth models.

**Strengths And Weaknesses:**

**Strengths**

*Presentation*: The results are very clearly explained and overall easy to follow. Additionally, all notations and setting descriptions are clearly described in Section 2.1. Finally, the required Assumptions required for the theoretical results are clearly and extensively explicated in Section 2.3. Finally, while the proofs and theoreticality of this work are heavy, the authors managed to explain them both extensively and clearly.

*Soundness*: The theoretical results are sound. Assumptions are precisely described in Section 2.3, and all proofs are relegated to the Appendices A, B, and C. The proof sketch plays an important role in giving a good intuition of the main ideas. Finally, the most important hyperparameters for the experiments, especially the momentum factor, are detailed in Appendix D.

**Weaknesses**

*Significance* This work is mainly theoretical and provides an interesting generalization of prior works on margin maximization to momentum algorithms, which are widely used in practice. However, it is unclear to me how to use these insights on margin maximization in practical scenarios beyond synthetic experimental settings where the assumptions of the theorems are satisfied. In particular, it is unclear how necesseary are the assumptions on log-concavity of the loss or homogeneity are, which would likely fail in the training of deep neural networks.

*Originality* The theoretical results remain close to prior works, especially [1], to which the proofs and results are similar, although generalized to a decaying learning rate and approximate steepest descent and especially momentum.

---

> ### Author Rebuttal · Authors · 2026-03-31
>
> We thank the reviewer for acknowledging the soundness and clarity of the work.
>
> Regarding significance, the field of implicit bias is mostly theoretical in nature, and all leading works in the field consider simpler setups than those used in practice. While not always directly practical, they contribute to a theoretical understanding of trained neural networks. Nevertheless, implicit bias *has* also found practical applications. Notably, Haim et al. (2022); Buzaglo et al. (2023); Oz et al. (2024) showed that it is possible to reconstruct training data from networks using the KKT conditions proved in implicit bias results, and Golbari et al. (2025)  conducted membership attacks using these same equations. Also, Vardi et al., (2022) and Frei et al. (2023) showed implications of implicit bias on robustness to adversarial data poisoning.
>
> We address the concrete limitations mentioned by the reviewer - log-concavity of the loss and homogeneity of the network. We kindly point out that, as mentioned in line 28 (right), that the logistic loss is a log-concave loss, and as this is the standard loss in practice for binary classification, we believe this is not much of a limitation. Regarding homogeneity, indeed practical models are often non-homogeneous, due to the presence of skip connections, bias terms and attention. However, some theoretical and practical works suggest that the deviation from homogeneity may not be large. In particular, Cai et al. (2025) extended implicit bias results on gradient descent to non-homogeneous models, showing that an approximate homogeneity relation holds for many architectures, such as ReLU networks with bias terms, VGG, and ResNet. From a practical viewpoint, *non-homogeneous models* were used in both the reconstruction and membership attacks mentioned above, and their success suggests an approximate homogeneity property occurs. Also, we ran preliminary experiments which show similar qualitative results for training CNNs with/without bias terms on a two-class task from CIFAR10.
>
> Regarding originality: we make a key technical contribution in the form of analysis of momentum-based algorithms. Lemma C.19 in particular combines tools proved on the momentum operator in Appendix B with a careful analysis of gradient magnitudes along the trajectory, to show an asymptotic relation between the momentum and the gradient. Additionally, we believe our introduction of approximate steepest descent (ASD) as a framework significantly generalizes previous works and constitutes a technical contribution. Finally, we also analyze composite algorithms, showing that often they are precise momentum steepest descent algorithms (e.g. Muon-Signum), while for Muon-Adam we demonstrate how one can overcome the non-trivial technical challenges of analyzing a composition of two ASD algorithms which are not MSD.
>
> Learning rate warm-up: this is easily integrated into the results.
> 1. For all results except Adam and Muon-Adam, any finite-time warm-up already satisfies the assumptions, since only the asymptotic relation $\eta (t) = o(t^{1/L-1})$ is required.
> 2. For Adam and Muon-Adam, indeed we require $\eta$ to be non-increasing, but the proofs hold almost as written if instead requiring $\eta$ to be *eventually* non-increasing (as long as $\eta$ is bounded), allowing for a finite time warm-up.
>
> Lines 217-218: in Zhang et. al (2024) and subsequent works, they added to $\eta_t = o(1)$ a technical assumption, which they show holds for polynomially decaying learning rates. We make no such assumption, i.e. our learning rate must not be inverse polynomial, only decaying.
>
> Long momentum: We thank the reviewer for this interesting question. Dynamics of such a regime would depend on the rate of convergence of $\beta$ to 1, relative to decay of the update norm. As $\beta$ approaches 1, momentum leans more towards "long-term" memory, requiring the temporal variation of the gradient to slow down to guarantee convergence of the momentum-gradient ratio. More explicitly, for a schedule $\beta(t)$ for which $1 - \beta(t) \to 0$ is non-summable, and a choice of $\eta(t)$ for which $(\int \eta )^{L-1} \cdot \eta = o(1 - \beta(t))$, we believe one could recover the same results.

---

> > ### Author Rebuttal · Reviewer_uoZA · 2026-04-04
> >
> > I thank the authors for their detailed answer. My questions have been mostly adressed, especially:
> >
> > **Learning rate warm-up**: The authors answered that for most results "only the asymptotic relation $\eta (t) = o(t^{1/L-1})$ is required"
> >
> > **Technical assumption** The authors clarified my question about the assumption on  "$\eta_t = o(1)$" compared to prior works.
> >
> > My remaining concern:
> >
> > **Originality** I appreciate the authors discussion on originality. I however agree with reviewers ytxx, A65r that the first contribution remains very close to prior works and mainly consists of a time reparametrization.
> >
> > I will maintain my score to recommend weak acceptance.

---

### Official Review · Reviewer_ytxx · 2026-03-13

**Soundness:** 3
**Presentation:** 2
**Significance:** 2
**Originality:** 2
**Overall Recommendation:** 4
**Confidence:** 4

**Summary:**

This paper studies the implicit bias of normalized steepest descent and momentum-based optimizers (including MomentumGD, Signum, Adam, and Muon) on smooth homogeneous neural networks. The main technical idea is to analyze momentum-based methods through an “approximate steepest descent” (ASD) framework and then derive convergence of the normalized parameter direction to KKT points of corresponding max-margin problems.

I found the paper technically ambitious and the unified geometric viewpoint interesting. In particular, the ASD perspective is a promising way to organize analyses across different optimizers.

**Compliance With Llm Reviewing Policy:**

Affirmed.

**Final Justification:**

The author fully resolved my concerns.

**Key Questions For Authors:**

1. For normalized steepest descent in continuous time, is the learning-rate schedule doing anything beyond a time reparameterization? If yes, please state exactly what is schedule-dependent in the resulting geometry or proofs.
2. Can the momentum-based results be reformulated under a weaker condition than full directional convergence in T2? For example,would the limiting point of the directions of $\theta_t$ be enough to derive similar results?

**Limitations:**

yes

**Strengths And Weaknesses:**

## Strengths

- The paper does not analyze each optimizer in a completely ad hoc manner, but instead introduces ASD as a common framework.
- The paper has serious math rigor. The statement of each argument is clear and accurate. The proofs are complete.
- The paper introduces the ASD framework to deal with momentum-based algorithm, which is novel and powerful.

## Weaknesses
1. The learning-rate-schedule extension for normalized steepest descent seems to have limited conceptual content. For the normalized steepest descent part, I am not convinced that introducing a learning-rate schedule adds substantial insight beyond a continuous-time time reparameterization. Unless we are considering some discrete descent algorithm, LR scheduler has no effect on the train trajectory. In other words, this analysis is nothing new but the same as the one in Tsilivis et al. (2025).

2. The directional convergence assumption (T2) is too strong and externalizes a core difficulty. This is a very strong assumption, and in my opinion it significantly limits the impact of the momentum-based conclusions. The core claim of the paper is about the implicit bias of momentum-based methods, but the analysis assumes away one of the central difficulties: whether the normalized direction actually converges. This is especially concerning from a practical standpoint. For some simple cases, the parameter direction may not even converge under Adam.

---

> ### Author Rebuttal · Authors · 2026-03-31
>
> We thank the reviewer for acknowledging the ambition of the paper and the interesting perspective provided by ASD.
>
> Regarding the novelty of the NSD (normalized steepest descent) section, we agree with the reviewer and thank them for this constructive feedback. Indeed, one can derive the core claims of Theorems 3.1, 3.2 based on a reparameterization argument and Tsilivis et al. (2025), with no additional proofs required. We will reframe this section as a remark rather than a result.  We request the reviewer to keep in mind that this is a small part of our paper, (less than 3 pages of proofs out of ~30), and we acknowledge in the main text (line 320, right) that the techniques closely follow Tsilivis et al. (2025). Our main contribution, analysis of momentum-based algorithms on smooth homogeneous models, is novel and significantly generalizes the existing results which hold for linear models only.
>
> Regarding directional convergence, we argue that this assumption is reasonable and indeed common when analyzing implicit bias in homogeneous models. First, we point out, as mentioned in line 190 (right) that the implicit bias literature classically decouples the question of directional convergence from results based on the assumption of such convergence. Indeed, some foundational, highly cited papers in the field of implicit bias (Gunasekar et al. (2018a;b); Chizat and Bach (2020); Nacson et al. (2019b)) assume directional convergence of parameters. Second, in our view the field of implicit bias concerns the characterization of solutions of optimizers, under the fundamental assumption of model convergence. As the direction of parameters in homogeneous models captures all model behavior up to scale, assuming directional convergence is akin to assuming convergence of the model. For this reason, even works which avoid assuming directional convergence, such as Lyu & Li (2019), Tsilivis et al. (2025), analyze limit points of the direction, which are mostly of interest when the direction indeed converges. Notably, all the above works were published *with no result known about directional convergence at the time*, and this was proved later for gradient descent by Ji & Telgarsky (2020), which makes us believe that work based on assumptions of directional convergence has much merit. Finally, please note that in Figure 1 we show empirical evidence of directional convergence in our experiments.
>
> The reviewer mentions that some simple cases showcase the non-convergence of the direction in Adam, which we dispute. The Adam literature indeed contains examples of non-convergence of Adam (e.g. Reddi et al. (2019); Bock and Weiss (2022)). However, crucially, such results construct optimization landscapes which do not adhere to our setting: an exponentially tailed loss and a homogeneous model (and indeed a decaying learning rate schedule). To the best of our knowledge, no counterexamples exist in this setting. Additionally, even if counterexamples could be constructed, our empirical results and the convergent behavior of Adam in practice suggest that it is reasonable to expect directional convergence on realistic datasets, and if so we provide an implicit bias result at least in some settings.
>
> While removing the assumption of directional convergence remains an important open direction, we believe our results provide meaningful progress under commonly assumed and empirically supported conditions.

---

> > ### Author Rebuttal · Reviewer_ytxx · 2026-04-06
> >
> > The authors fully resolved my concerns. I raised my score.

---

### Decision · Program_Chairs · 2026-04-30

**Decision:**

Accept (regular)

**Comment:**

This paper studies the implicit bias of normalized steepest descent and momentum-based optimizers on smooth homogeneous neural networks. The results in this work cover Adam, Muon, and their variants. It is appreciated by the majority of the reviewers that the authors have developed some new machinery to analyze the momentum-based methods via the proposed "approximate steepest descent" analysis framework, which allows them to establish the convergence of the normalized parameter direction to a KKT point of the corresponding max-margin problem (with a certain norm), under the proposed assumptions.

This paper receives one negative score from Reviewer A65r. After the rebuttal, the remaining concerns in the reviewer's latest reply are regarding:
- (1) Theorems 3.1 and 3.2 can be derived from a prior work; and
- (2) Assumption T2 (i.e., the assumption of directional convergence).

For (1), the authors' response is fine with me, since Theorems 3.1 and 3.2 are in the earlier part of this paper, and the major contributions of a paper are oftentimes in its later part, which indeed is the case here. For (2), this concern is also shared by another reviewer. While in the authors' reply they provide some historical bit on the assumption and argue that the empirical results support this assumption, whether the directional convergence assumption (T2) is strong is debatable.

Overall, I believe that this work does develop some non-trivial analysis for obtaining the theoretical results that might inspire future works. However, the majority of the reviewers point out that some of the assumptions can be considered quite strong, and some also point out that the experimental validation is limited. On the other hand, the paper is transparent about the limitation that certain assumptions may not hold and discusses these limitations, and is fairly accurate in presenting the theoretical results.

For all these reasons, the paper is recommended for a weak accept.